# Approximating Nash Equilibria in Normal-Form Games via Unbiased Stochastic Optimization

## Abstract

We propose the first, to our knowledge, loss function for approximate Nash equilibria of normal-form games that is amenable to unbiased Monte Carlo estimation. This construction allows us to deploy standard non-convex stochastic optimization techniques for approximating Nash equilibria, resulting in novel algorithms with provable guarantees. We complement our theoretical analysis with experiments demonstrating that stochastic gradient descent can outperform previous state-of-the-art approaches.

## 1  Introduction

Nash equilibrium famously encodes stable behavioral outcomes in multi-agent systems and is arguably the most influential solution concept in game theory. Formally speaking, if $n$ players independently choose $n$, possibly mixed, strategies ($x_i$ for $i \in [n]$) and their joint strategy ($\boldsymbol{x} = \prod_i x_i$) constitutes a *Nash equilibrium*, then no player has any incentive to unilaterally deviate from their strategy. This concept has sparked extensive research in various fields, ranging from economics [30] to machine learning [16], and has even inspired behavioral theory generalizations such as quantal response equilibria which allow for more realistic models of boundedly rational agents [28].

Unfortunately, when considering Nash equilibria beyond the special case of the 2-player, zero-sum scenario, two significant challenges arise. First, it becomes unclear how a group of $n$ independent players would collectively identify a Nash equilibrium when multiple equilibria are possible, giving rise to the *equilibrium selection* problem [18]. Secondly, even approximating a single Nash equilibrium is known to be computationally intractable and specifically PPAD-complete [11]. Combining both problems together, e.g., testing for the existence of equilibria with welfare greater than some fixed threshold is NP-hard and it is in fact even hard to approximate (i.e., finding a Nash equilibrium with welfare greater than $\omega$ for any $\omega > 0$, even when the best equilibrium has welfare $1 - \omega$) [2].

From a machine learning (ML) practitioner's perspective, however, such computational complexity results hardly give pause for thought as collectively we have become all too familiar with the unreasonable effectiveness of ML heuristics in circumventing such obstacles. Famously, non-convex optimization is NP-hard, even if the goal is to compute a local minimizer [31], however, stochastic gradient descent (and variants thereof) succeed in training models with billions of parameters [7].

Unfortunately, computational techniques for Nash equilibrium have so far not achieved anywhere near the same level of success. In contrast, most modern Nash equilibrium solvers for $n$-player, $m$-action, general-sum, normal-form games (NFGs) are practically restricted to a handful of players and/or actions per player except in special cases (e.g., symmetric [38] or mean-field games [34]). This is partially due to the fact that an NFG is represented by a tensor with an exponential $nm^n$ entries; even *reading* this description into memory can be computationally prohibitive. More to the point, any

35 computational technique that presumes *exact* computation of the *expectation* of any function sampled
36 according to $x$ similarly does not have any hope of scaling beyond small instances.

37 This inefficiency arguably lies at the core of the differential success between ML optimization and
38 equilibrium computation. For example, numerous techniques exist that reduce the problem of Nash
39 equilibrium computation to finding the minimum of the expectation of a random variable (see related
40 work section). Unfortunately, unlike the source of randomness in ML applications where batch
41 learning suffices to easily produce unbiased estimators, these techniques do not extend easily to game
42 theory which incorporates non-linear functions such as maximum, best-response amongst others.
43 This raises our motivating goal:

**Can we solve for Nash equilibria via unbiased stochastic optimization?**

44 **Our results.** Following in the successful steps of the interplay between ML and stochastic optimiza-
45 tion, we reformulate the approximation of Nash equilibria in an NFG as a stochastic non-convex
46 optimization problem admitting unbiased Monte-Carlo estimation. This enables the use of powerful
47 solvers and advances in parallel computing to efficiently enumerate Nash equilibria for $n$-player,
48 general-sum games. Furthermore, this re-casting allows practitioners to incorporate other desirable
49 objectives into the problem such as "find an approximate Nash equilibrium with welfare above $\omega$"
50 or "find an approximate Nash equilibrium nearest the current observed joint strategy" resolving the
51 equilibrium selection problem in effectively ad-hoc and application tailored manner. Concretely, we
52 make the following contributions by producing:

53 • A loss function $\mathcal{L}(\boldsymbol{x})$ 1) whose global minima coincide with interior Nash equilibria in normal
54   form games, 2) admits unbiased Monte-Carlo estimation, and 3) is Lipschitz and bounded.

55 • A loss function $\mathcal{L}^\tau(\boldsymbol{x})$ 1) whose global minima coincide with logit equilibria (QREs) in normal
56   form games, 2) admits unbiased Monte-Carlo estimation, and 3) is Lipschitz and bounded.

57 • An efficient randomized algorithm for approximating Nash equilibria in a novel class of games. The
58   algorithm emerges by employing a recent $\mathcal{X}$-armed bandit approach to $\mathcal{L}^\tau(\boldsymbol{x})$ and connecting its
59   stochastic optimization guarantees to approximate Nash guarantees. For large games, this enables
60   approximating equilibria *faster* than the game can even be read into memory.

61 • An empirical comparison of stochastic gradient descent against state-of-the-art baselines for
62   approximating NEs in large games. In some games, vanilla SGD actually improves upon previous
63   state-of-the-art; in others, SGD is slowed by saddle points, a familiar challenge in deep learning [12].

64 Overall, this perspective showcases a promising new route to approximating equilibria at scale in
65 practice. We conclude the paper with discussion for future work.

## 2 Preliminaries

67 In an $n$-player, normal-form game, each player $i \in \{1, \ldots, n\}$ has a strategy set $\mathcal{A}_i =$
68 $\{a_{i1}, \ldots, a_{im_i}\}$ consisting of $m_i$ pure strategies. These strategies can be naturally indexed, so
69 we redefine $\mathcal{A}_i = \{1, \ldots, m_i\}$ as an abuse of notation. Each player $i$ also has a utility function,
70 $u_i : \mathcal{A} = \prod_i \mathcal{A}_i \to [0, 1]$, (equiv. "payoff tensor") that maps joint actions to payoffs in the unit-
71 interval . Note that equilibria are invariant to payoff shift and scale [27] so we are effectively assuming
72 we know bounds on possible payoffs. We denote the average cardinality of the players' action sets
73 by $\bar{m} = \frac{1}{n} \sum_k m_k$ and maximum by $m^* = \max_k m_k$. Player $i$ may play a mixed strategy by
74 sampling from a distribution over their pure strategies. Let player $i$'s mixed strategy be represented
75 by a vector $x_i \in \Delta^{m_i - 1}$ where $\Delta^{m_i - 1}$ is the $(m_i - 1)$-dimensional probability simplex embedded
76 in $\mathbb{R}^{m_i}$. Each function $u_i$ is then extended to this domain so that $u_i(\boldsymbol{x}) = \sum_{\boldsymbol{a} \in \mathcal{A}} u_i(\boldsymbol{a}) \prod_j x_{ja_j}$
77 where $\boldsymbol{x} = (x_1, \ldots, x_n)$ and $a_j \in \mathcal{A}_j$ denotes player $j$'s component of the joint action $\boldsymbol{a} \in \mathcal{A}$. For
78 convenience, let $x_{-i}$ denote all components of $\boldsymbol{x}$ belonging to players other than player $i$.

79 The joint strategy $\boldsymbol{x} \in \prod_i \Delta^{m_i - 1}$ is a Nash equilibrium if and only if, for all $i \in \{1, \ldots, n\}$,
80 $u_i(z_i, x_{-i}) \leq u_i(\boldsymbol{x})$ for all $z_i \in \Delta^{m_i - 1}$, i.e., no player has any incentive to unilaterally deviate from
81 $\boldsymbol{x}$. Nash is typically relaxed with $\epsilon$-Nash, our focus: $u_i(z_i, x_{-i}) \leq u_i(\boldsymbol{x}) + \epsilon$ for all $z_i \in \Delta^{m_i - 1}$.

82 As an abuse of notation, let the atomic action $a_i = e_i$ also denote the $m_i$-dimensional "one-hot" vector
83 with all zeros aside from a 1 at index $a_i$; its use should be clear from the context. We also introduce

| Loss | Function | Obstacle |
|---|---|---|
| Exploitabilty | $\max_k \epsilon_k(\boldsymbol{x})$ | max of r.v. |
| Nikaido-Isoda (NI) | $\sum_k \epsilon_k(\boldsymbol{x})$ | max of r.v. |
| Fully-Diff. Exp | $\sum_k \sum_{a_k \in \mathcal{A}_k}[\max(0, u_k(a_k, x_{-i}) - u_k(\boldsymbol{x}))]^2$ | max of r.v. |
| Gradient-based NI | NI w/ $\mathtt{BR}_k \leftarrow \mathtt{aBR}_k = \Pi_\Delta\left(x_k + \eta\nabla_{x_k}u_k(\boldsymbol{x})\right)$ | $\Pi_\Delta$ of r.v. |
| Unconstrained | Loss + Simplex Deviation Penalty | sampling from $x_i \in \mathbb{R}^{m_k}$ |

Table 1: Previous loss functions for NFGs and their obstacles to unbiased estimation.

$\nabla^i_{x_i}$ as player $i$'s utility gradient. And for convenience, denote by $H^i_{il} = \mathbb{E}_{x_{-il}}[u_i(a_i, a_l, x_{-il})]$ the bimatrix game approximation [20] between players $i$ and $l$ with all other players marginalized out; $x_{-il}$ denotes all strategies belonging to players other than $i$ and $l$ and $u_i(a_i, a_l, x_{-il})$ separates out $l$'s strategy $x_l$ from the rest of the players $x_{-i}$. Similarly, denote by $T^i_{ilq} = \mathbb{E}_{x_{-ilq}}[u_i(a_i, a_l, a_q, x_{-ilq})]$ the 3-player tensor approximation to the game. Note player $i$'s utility can now be written succinctly as $u_i(x_i, x_{-i}) = x_i^\top\nabla^i_{x_i} = x_i^\top H^i_{il}x_l = x_iT^i_{ilq}x_lx_q$ for any $l, q$ where we use Einstein notation for tensor arithmetic. For convenience, define $\mathtt{diag}(z)$ as the function that places a vector $z$ on the diagonal of a square matrix, and $\mathtt{diag3} : z \in \mathbb{R}^d \to \mathbb{R}^{d \times d \times d}$ as a 3-tensor of shape $(d, d, d)$ where $\mathtt{diag3}(z)_{iii} = z_i$. Following convention from differential geometry, let $T_v\mathcal{M}$ be the tangent space of a manifold $\mathcal{M}$ at $v$. For the interior of the $d$-action simplex $\Delta^{d-1}$, the tangent space is the same at every point, so we drop the $v$ subscript, i.e., $T\Delta^{d-1}$. We denote the projection of a vector $z \in \mathbb{R}^d$ onto this tangent space as $\Pi_{T\Delta^{d-1}}(z) = z - \frac{1}{d}\mathbf{1}^\top z$. We drop $d$ when the dimensionality is clear from the context. Finally, let $\mathcal{U}(S)$ denote a discrete uniform distribution over elements from set $S$.

## 3   Related Work

Representing the problem of computing a Nash equilibrium as an optimization problem is not new. A variety of loss functions and pseudo-distance functions have been proposed. Most of them measure some function of how much each player can exploit the joint strategy by unilaterally deviating:

$$\epsilon_k(\boldsymbol{x}) \stackrel{\mathsf{def}}{=} u_k(\mathtt{BR}_k, x_{-k}) - u_k(\boldsymbol{x}) \text{ where } \mathtt{BR}_k \in \arg\max_z u_k(z, x_{-k}). \tag{1}$$

As argued in the introduction, we believe it is important to be able to subsample payoff tensors of normal-form games in order to scale to large instances. As Nash equilibria can consist of mixed strategies, it is advantageous to be able to sample from an equilibrium to estimate its exploitability $\epsilon$. However none of these losses is amenable to unbiased estimation under sampled play. Each of the functions currently explored in the literature is biased under sampled play either because 1) a random variable appears as the argument of a complex, nonlinear (non-polynomial) function or because 2) how to sample play is unclear. Exploitability, Nikaido-Isoda (NI) [32] (also known by $\mathtt{NashConv}$ [21] and ADI [15]), as well as fully-differentiable options ([36], p. 106, Eqn 4.31) introduce bias when a $\max$ over payoffs is estimated using samples from $\boldsymbol{x}$. Gradient-based NI [35] requires projecting the result of a gradient-ascent step onto the simplex; for the same reason as the $\max$, this is prohibitive because it is a nonlinear operation which introduces bias. Lastly, unconstrained optimization approaches ([36], p. 106) that instead penalize deviation from the simplex lose the ability to sample from strategies when iterates are no longer proper distributions. Table 1 summarizes these complications.

## 4   Nash Equilibrium as Stochastic Optimization

We will now develop our proposed loss function which is amenable to unbiased estimation. Our key technical insight is to pay special attention to the geometry of the simplex. To our knowledge, prior works have failed to recognize the role of the tangent space $T\Delta$. Proofs are in the appendix.

### 4.1   Stationarity on the Simplex Interior

**Lemma 1.** *Assuming player $i$'s utility, $u_i(x_i, x_{-i})$, is concave in its own strategy $x_i$, a strategy in the interior of the simplex is a best response $\mathtt{BR}_i$ if and only if it has zero projected-gradient[1] norm:*

---

[1]Not to be confused with the nonlinear (i.e., introduces bias) projected gradient operator introduced in [19].

$$BR_i \in \left( int\Delta \cap \arg\max_z u_i(z, x_{-i}) - u_i(x_i, x_{-i}) \right) \iff (BR_i \in int\Delta) \wedge (||\Pi_{T\Delta}[\nabla^i_{BR_i}]|| = 0). \tag{2}$$

121  In NFGs, each player's utility is linear in $x_i$, thereby satisfying the concavity condition of Lemma 1.

## 4.2  Projected Gradient Norm as Loss

123  An equivalent description of a Nash equilibrium is a joint strategy $\boldsymbol{x}$ where every player's strategy is
124  a best response to the equilibrium (i.e., $x_i = BR_i$ so that $\epsilon_i(\boldsymbol{x}) = 0$). Lemma 1 states that any interior
125  best response has zero projected-gradient norm, which inspires the following loss function

$$\mathcal{L}(\boldsymbol{x}) = \sum_k \eta_k ||\Pi_{T\Delta}(\nabla^k_{x_k})||^2 \tag{3}$$

126  where $\eta_k > 0$ represent scalar weights, or equivalently, step sizes to be explained next.

127  **Proposition 1.** *The loss $\mathcal{L}$ is equivalent to* `NashConv`*, but where player $k$'s best response is approxi-*
128  *mated by a single step of projected-gradient ascent with step size $\eta_k$:* $\mathtt{aBR}_k = x_k + \eta_k \Pi_{T\Delta}(\nabla^k_{x_k})$.

129  This connection was already pointed out in prior work for unconstrained problems [15, 35], but this
130  result is the first for strategies constrained to the simplex.

## 4.3  Connection to True Exploitability

132  In general, we can bound exploitability in terms of the projected-gradient norm as long as each
133  player's utility is concave (this result extends beyond gradients to subgradients of non-smooth
134  functions).

135  **Lemma 2.** *The amount a player can gain by exploiting a joint strategy $\boldsymbol{x}$ is upper bounded by a*
136  *quantity proportional to the norm of the projected-gradient:*

$$\epsilon_k(\boldsymbol{x}) \leq \sqrt{2} ||\Pi_{T\Delta}(\nabla^k_{x_k})||. \tag{4}$$

137  This bound is not tight on the boundary of the simplex, which can be seen clearly by considering $x_k$
138  to be part of a pure strategy equilibrium. In that case, this analysis assumes $x_k$ can be improved upon
139  by a projected-gradient ascent step (via the equivalence pointed out in Proposition 1). However, that
140  is false because the probability of a pure strategy cannot be increased beyond 1. We mention this to
141  provide further intuition for why $\mathcal{L}(\boldsymbol{x})$ is only valid for interior equilibria.

142  Note that $||\Pi_{T\Delta}(\nabla^k_{x_k})|| \leq ||\nabla^k_{x_k}||$ because $\Pi_{T\Delta}$ is a projection. Therefore, this improves the naive
143  bounds on exploitability and distance to best responses given using the "raw" gradient $\nabla^k_{x_k}$.

144  **Lemma 3.** *The exploitability of a joint strategy $\boldsymbol{x}$, is upper bounded by a function of $\mathcal{L}(\boldsymbol{x})$:*

$$\epsilon \leq \sqrt{\frac{2n}{\min_k \eta_k}} \sqrt{\mathcal{L}(\boldsymbol{x})} \stackrel{\mathsf{def}}{=} f(\mathcal{L}). \tag{5}$$

## 4.4  Unbiased Estimation

146  As discussed in Section 3, a primary obstacle to unbiased estimation of $\mathcal{L}(\boldsymbol{x})$ is the presence of
147  complex, nonlinear functions of random variables, with the projection of a point onto the simplex
148  being one such example (see $\Pi_\Delta$ in Table 1). However, $\Pi_{T\Delta}$, *the projection onto the tangent space*
149  *of the simplex, is linear*! This is the key that allows us to design an unbiased estimator (Lemma 5).

150  Our proposed loss requires computing the squared norm of the *expected value* of the gradient
151  under the players' mixed strategies, i.e., the $l$-th entry of player $k$'s gradient equals $\nabla^k_{x_{kl}} =$
152  $\mathbb{E}_{a_{-k} \sim x_{-k}} u_k(a_{kl}, a_{-k})$. By analogy, consider a random variable $Y$. In general, $\mathbb{E}[Y]^2 \neq \mathbb{E}[Y^2]$.
153  This means that we cannot just sample projected-gradients and then compute their average norm to
154  estimate our loss. However, consider taking two independent samples from two corresponding identi-
155  cally distributed, independent random variables $Y^{(1)}$ and $Y^{(2)}$. Then $\mathbb{E}[Y^{(1)}]^2 = \mathbb{E}[Y^{(1)}]\mathbb{E}[Y^{(2)}] =$

| | Exact | Sample Others | Sample All |
|---|---|---|---|
| Estimator of $\nabla_{x_k}^{k(p)}$ | $u_k(a_{kl}, x_{-k})$ | $u_k(a_{kl}, a_{-k} \sim x_{-k})$ | $m_k u_k(a_{kl} \sim \mathcal{U}(\mathcal{A}_k), a_{-k} \sim x_{-k})e_l$ |
| $\hat{\nabla}_{x_k}^{k(p)}$ Bounds | $[0, 1]$ | $[0, 1]$ | $[0, m_k]$ |
| $\hat{\nabla}_{x_k}^{k(p)}$ Query Cost | $\prod_{i=1}^n m_i$ | $m_k$ | $1$ |
| $\mathcal{L}$ Bounds | $\pm\frac{1}{4}\sum_k \eta_k m_k$ | $\pm\frac{1}{4}\sum_k \eta_k m_k$ | $\pm\frac{1}{4}\sum_k \eta_k m_k^3$ |
| $\mathcal{L}$ Query Cost | $n\prod_{i=1}^n m_i$ | $2n\bar{m}$ | $2n$ |

Table 2: Examples and Properties of Unbiased Estimators of Loss and Player Gradients ($\hat{\nabla}_{x_k}^{k(p)}$).

$\mathbb{E}[Y^{(1)}Y^{(2)}]$ by properties of expected value over products of independent random variables. This is a common technique to construct unbiased estimates of expectations over polynomial functions of random variables. Proceeding in this way, define $\nabla_{x_k}^{k(1)}$ as a random variable distributed according to the distribution induced by all other players' mixed strategies ($j \neq k$). Let $\nabla_{x_k}^{k(2)}$ be independent and distributed identically to $\nabla_{x_k}^{k(1)}$. Then

$$\mathcal{L}(\boldsymbol{x}) = \mathbb{E}[\sum_k \eta_k (\underbrace{\hat{\nabla}_{x_k}^{k(1)} - \frac{1}{m_k}(\mathbf{1}^\top \hat{\nabla}_{x_k}^{k(1)})\mathbf{1}}_{\text{projected-gradient 1}})^\top (\underbrace{\hat{\nabla}_{x_k}^{k(2)} - \frac{1}{m_k}(\mathbf{1}^\top \hat{\nabla}_{x_k}^{k(2)})\mathbf{1}}_{\text{projected-gradient 2}})] \qquad (6)$$

where $\hat{\nabla}_{x_k}^{k(p)}$ is an unbiased estimator of player $k$'s gradient. This unbiased estimator can be constructed in several ways. The most expensive, an exact estimator, is constructed by marginalizing player $k$'s payoff tensor over all other players' strategies. However, a cheaper estimate can be obtained at the expense of higher variance by approximating this marginalization with a Monte Carlo estimate of the expectation. Specifically, if we sample a single action for each of the remaining players, we can construct an unbiased estimate of player $k$'s gradient by considering the payoff of each of its actions against the sampled background strategy. Lastly, we can consider constructing a Monte Carlo estimate of player $k$'s gradient by sampling only a single action from player $k$ to represent their entire gradient. Each of these approaches is outlined in Table 2 along with the query complexity [3] of computing the estimator and bounds on the values it can take (derived via Lemma 19).

We can extend Lemma 3 to one that holds under $T$ samples with probability $1 - \delta$ by applying, for example, a Hoeffding bound: $\epsilon \leq f\big(\hat{\mathcal{L}}(\boldsymbol{x}) + \mathcal{O}(\sqrt{\frac{1}{T}\ln(1/\delta)})\big)$.

## 4.5 Interior Equilibria

We discussed earlier that $\mathcal{L}(\boldsymbol{x})$ captures interior equilibria. But some games may only have *pure* equilibria. We show how to circumvent this shortcoming by considering quantal response equilibria (QREs), specifically, logit equilibria. By adding an entropy bonus to each player's utility, we can

- guarantee **all** equilibria are interior,
- still obtain unbiased estimates of our loss,
- maintain an upper bound on the exploitability $\epsilon$ of any approximate equilibrium in the original game (i.e., the game without an entropy bonus).

Define $u_k^\tau(\boldsymbol{x}) = u_k(\boldsymbol{x}) + \tau S(x_k)$ where the Shannon entropy $S(x_k) = -\sum_l x_{kl} \ln(x_{kl})$ is a 1-strongly concave function with respect to the 1-norm [6]. Also define $\mathcal{L}^\tau(\boldsymbol{x})$ as before except where $\nabla_{x_k}^k$ is replaced with $\nabla_{x_k}^{k\tau} = \nabla_{x_k} u_k^\tau(\boldsymbol{x})$, i.e., the gradient of player $k$'s utility *with* the entropy bonus.

It is well known that Nash equilibria of entropy-regularized games satisfy the conditions for logit equilibria [23], which are solutions to the fixed point equation $x_k = \texttt{softmax}(\frac{\nabla_{x_k}^k}{\tau})$. The appearance of the $\texttt{softmax}$ makes clear that all probabilities have positive mass at positive temperature.

Recall that in order to construct an unbiased estimate of our loss, we simply needed to construct unbiased estimates of player gradients. The introduction of the entropy term to player $k$'s utility is special in that it depends entirely on known quantities, i.e., the player's own mixed strategy. We can directly and deterministically compute $\tau \frac{dS}{dx_k} = -\tau(\ln(x_k) + \mathbf{1})$ and add this to our estimator of $\nabla_{x_k}^{k(p)}$: $\hat{\nabla}_{x_k}^{k\tau(p)} = \hat{\nabla}_{x_k}^{k(p)} + \tau \frac{dS}{dx_k}$. Consider our refined loss function with changes in blue:

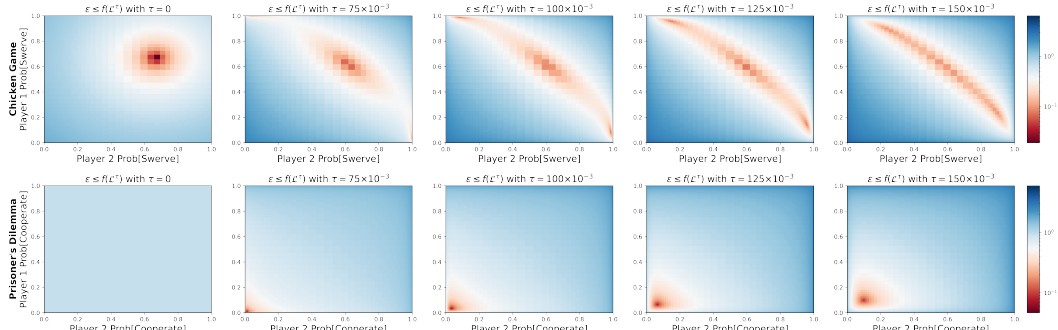

Figure 1: Upper Bound ($\epsilon \leq f(\mathcal{L}^\tau)$) Heatmap Visualization. The first row examines the loss landscape for the classic anti-coordination game of Chicken (Nash equilibria: $(0,1), (1,0), (2/3, 1/3)$) while the second row examines the Prisoner's dilemma (Unique Nash equilibrium: $(0,0)$). Temperature increases for each plot moving to the right. For high temperatures, interior (fully-mixed) strategies are incentivized while for lower temperatures, nearly pure strategies can achieve minimum exploitability. For zero temperature, pure strategy equilibria (e.g., defect-defect) are not captured by the loss as illustrated by the bottom-left Prisoner's Dilemma plot with a constant loss surface.

$$\mathcal{L}^\tau(\boldsymbol{x}) = \sum_k \eta_k ||\Pi_{T\Delta}(\nabla_{x_k}^{k\tau})||^2. \tag{7}$$

As mentioned above, the utilities with entropy bonuses are still concave, therefore, a similar bound to Lemma 2 applies. We use this to prove the QRE counterpart to Lemma 3 where $\epsilon_{QRE}$ is the exploitability of an approximate equilibrium in a game with entropy bonuses.

**Lemma 4.** *The entropy regularized exploitability, $\epsilon_{QRE}$, of a joint strategy $\boldsymbol{x}$, is upper bounded as:*

$$\epsilon_{QRE} \leq \sqrt{\frac{2n}{\min_k \eta_k}} \sqrt{\mathcal{L}^\tau(\boldsymbol{x})} \stackrel{\mathsf{def}}{=} f(\mathcal{L}^\tau). \tag{8}$$

Lastly, we establish a connection between quantal response equilibria and Nash equilibria that allows us to approximate Nash equilibria in the original game via minimizing our modified loss $\mathcal{L}^\tau(\boldsymbol{x})$.

**Lemma 14** ($\mathcal{L}^\tau$ Scores Nash Equilibria). *Let $\mathcal{L}^\tau(\boldsymbol{x})$ be our proposed entropy regularized loss function with payoffs bounded in $[0,1]$ and $\boldsymbol{x}$ be an approximate QRE. Then it holds that*

$$\epsilon \leq n\tau\left(W(1/e) + \frac{\bar{m} - 2}{e}\right) + 2\sqrt{\frac{n \max_k m_k}{\min_k \eta_k}} \sqrt{\mathcal{L}^\tau(\boldsymbol{x})} \tag{9}$$

*where $W$ is the Lambert function: $W(1/e) = W(\exp(-1)) \approx 0.278$.*

This upper bound is plotted as a heatmap for familiar games in Figure 1. Notice how pure equilibria are not visible as minima for zero temperature, but appear for slightly warmer temperatures.

## 5 Analysis

In the preceding section we established a loss function that upper bounds the exploitability of an approximate equilibrium. In addition, the zeros of this loss function have a one-to-one correspondence with quantal response equilibria (which approximate Nash equilibria at low temperature).

Here, we derive properties that suggest it is "easy" to optimize. While this function is generally non-convex and may suffer from a proliferation of saddle points and local maxima (Figure 2) , it is Lipschitz continuous (over a subset of the interior) and bounded. These are two commonly made assumptions in the literature on non-convex optimization, which we leverage in Section 6. In addition, we can derive its gradient, its Hessian, and characterize its behavior around global minima.

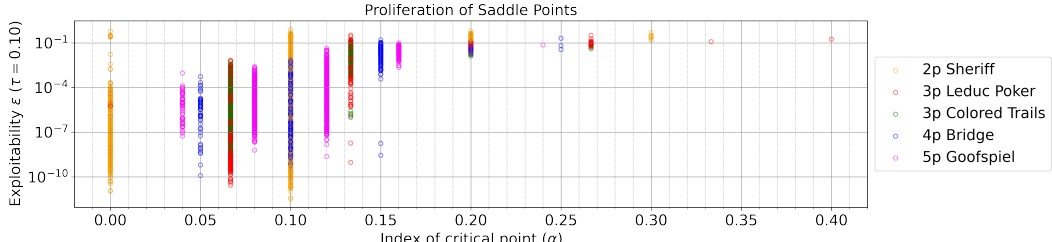

Figure 2: We reapply the analysis of [12], originally designed to understand the success of SGD in deep learning, to "slices" of several popular extensive form games. To construct a slice (or *meta-game*), we randomly sample 6 deterministic policies and then consider the corresponding $n$-player, 6-action normal-form game at $\tau = 0.1$ (with payoffs normalized to $[0, 1]$). The index of a critical point $\boldsymbol{x}_c$ ($\nabla_{\boldsymbol{x}} \mathcal{L}^\tau(\boldsymbol{x}_c) = \boldsymbol{0}$) indicates the fraction of negative eigenvalues in the Hessian of $\mathcal{L}^\tau$ at $\boldsymbol{x}_c$; $\alpha = 0$ indicates a local minimum, 1 a maximum, else a saddle point. We see a positive correlation between exploitability and $\alpha$ indicating a lower prevalence of local minima at high exploitability.

**Lemma 15.** *The gradient of $\mathcal{L}^\tau(\boldsymbol{x})$ with respect to player $l$'s strategy $x_l$ is*

$$\nabla_{x_l} \mathcal{L}^\tau(\boldsymbol{x}) = 2 \sum_k \eta_k B_{kl}^\top \Pi_{T\Delta}(\nabla_{x_k}^{k\tau}) \tag{10}$$

*where $B_{ll} = -\tau[I - \frac{1}{m_l}\boldsymbol{1}\boldsymbol{1}^\top]\boldsymbol{diag}(\frac{1}{x_l})$ and $B_{kl} = [I - \frac{1}{m_k}\boldsymbol{1}\boldsymbol{1}^\top]H_{kl}^k$ for $k \neq l$.*

**Lemma 17.** *The Hessian of $\mathcal{L}^\tau(\boldsymbol{x})$ can be written*

$$\boldsymbol{Hess}(\mathcal{L}^\tau) = 2\big[\tilde{B}^\top \tilde{B} + T\Pi_{T\Delta}(\tilde{\nabla}^\tau)\big] \tag{11}$$

*where $\tilde{B}_{kl} = \sqrt{\eta_k}B_{kl}$, $\Pi_{T\Delta}(\tilde{\nabla}^\tau) = [\eta_1\Pi_{T\Delta}(\nabla_{x_1}^{1\tau}), \dots, \eta_n\Pi_{T\Delta}(\nabla_{x_n}^{n\tau})]$, and we augment $T$ (the 3-player approximation to the game, $T_{lqk}^k$) so that $T_{lll}^l = \tau \boldsymbol{diag3}(\frac{1}{x_l^2})$.*

At an equilibrium, the latter term disappears because $\Pi_{T\Delta}(\nabla_{x_k}^{k\tau}) = \boldsymbol{0}$ for all $k$ (Lemma 1). If $\mathcal{X}$ was $\mathbb{R}^{n\bar{m}}$, then we could simply check if $\tilde{B}$ is full-rank to determine if $Hess \succ 0$. However, $\mathcal{X}$ is a simplex product, and we only care about curvature in directions toward which we can update our equilibrium. Toward that end, define $M$ to be the $n(\bar{m}+1) \times n\bar{m}$ matrix that stacks $\tilde{B}$ on top of a repeated identity matrix that encodes orthogonality to the simplex:

$$M(\boldsymbol{x}) = \begin{bmatrix} -\tau\sqrt{\eta_1}\Pi_{T\Delta}(\frac{1}{x_1}) & \sqrt{\eta_1}\Pi_{T\Delta}(H_{12}^1) & \cdots & \sqrt{\eta_1}\Pi_{T\Delta}(H_{1n}^1) \\ \vdots & \vdots & \vdots & \vdots \\ \sqrt{\eta_n}\Pi_{T\Delta}(H_{n1}^n) & \cdots & \sqrt{\eta_n}\Pi_{T\Delta}(H_{n,n-1}^n) & -\tau\sqrt{\eta_n}\Pi_{T\Delta}(\frac{1}{x_n}) \\ \boldsymbol{1}_1^\top & 0 & \cdots & 0 \\ \vdots & \vdots & \vdots & \vdots \\ 0 & \cdots & 0 & \boldsymbol{1}_n^\top \end{bmatrix} \tag{12}$$

where $\Pi_{T\Delta}(z \in \mathbb{R}^{a\times b}) = [I_a - \frac{1}{a}\boldsymbol{1}_a\boldsymbol{1}_a^\top]z$ subtracts the mean from each column of $z$ and $\frac{1}{x_i}$ is shorthand for $\boldsymbol{diag}(\frac{1}{x_i})$. If $M(x)z = \boldsymbol{0}$ for a nonzero vector $z \in \mathbb{R}^{n\bar{m}}$, this implies there exists a $z$ that 1) is orthogonal to the ones vectors of each simplex (i.e., is a valid equilibrium update direction) and 2) achieves zero curvature in the direction $z$, i.e., $z^\top(\tilde{B}^\top \tilde{B})z = z^\top(Hess)z = 0$, and so $Hess$ is not positive definite. Conversely, if $M(\boldsymbol{x})$ is of rank $n\bar{m}$ for a quantal response equilibrium $\boldsymbol{x}$, then the Hessian of $\mathcal{L}^\tau$ at $\boldsymbol{x}$ in the tangent space of the simplex product ($\mathcal{X} = \prod_i \mathcal{X}_i$) is positive definite. In this case, we call $\boldsymbol{x}$ *well*-isolated because it implies it is not connected to any other equilibria.

By analyzing the rank of $M$, we can confirm that many classical matrix games including Rock-Paper-Scissors, Chicken, Matching Pennies, and Shapley's game all induce strongly convex $\mathcal{L}^\tau$'s at zero temperature (i.e., they have unique mixed Nash equilibria). In contrast, a game like Prisoner's Dilemma has a unique pure strategy that will not be captured by our loss at zero temperature.

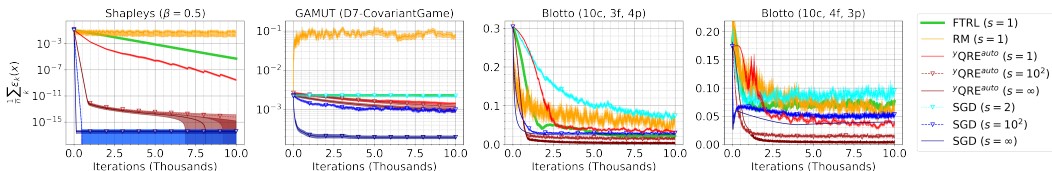

Figure 3: Comparison of SGD on $\mathcal{L}^{\tau=0}$ against baselines on four games evaluated in [15]. From left to right: 2-player, 3-action, nonsymmetric; 6-player, 5-action, nonsymmetric; 4-player, 66-action, symmetric; 3-player, 286-action, symmetric. SGD struggles at saddle points in Blotto.

## 6 Algorithms

We have formally transformed the approximation of Nash equilibria in NFGs into a **stochastic optimization** problem. To our knowledge, this is the first such formulation that allows one-shot unbiased Monte-Carlo estimation which is critical to introduce the use of powerful algorithms capable of solving high dimensional optimization problems. We explore two off-the-shelf approaches.

Stochastic gradient descent is the workhorse of high-dimensional stochastic optimization. It comes with guaranteed convergence to stationary points [10], however, it may converge to local, rather than global minima. It also enjoys implicit gradient regularization [4], seeking "flat" minima and performs approximate Bayesian inference [26]. Despite the lack of global convergence guarantee, in the next section, we find it performs well empirically in games previously examined by the literature.

We explore one other algorithmic approach to non-convex optimization based on minimizing regret, which enjoys finite time convergence rates. $\mathcal{X}$-armed bandits [8] systematically explore the space of solutions by refining a mesh over the joint strategy space, trading off exploration versus exploitation of promising regions.[2] Several approaches exist [5, 37] with open source implementations (e.g., [24]).

### 6.1 High Probability, Polynomial Convergence Rates

We use a recent $\mathcal{X}$-armed bandit approach called BLiN [14] to establish a high probability $\tilde{\mathcal{O}}(T^{-1/4})$ convergence rate to Nash equilibria in $n$-player, general-sum games under mild assumptions. The quality of this approximation improves as $\tau \to 0$, at the same time increasing the constant on the convergence rate via the Lipschitz constant $\sqrt{\hat{L}}$ defined below. For clarity, we assume users provide a temperature in the form $\tau = \frac{1}{\ln(1/p)}$ with $p \in (0, 1)$ which ensures all equilibria have probability mass greater than $\frac{p}{m^*}$ for all actions (Lemma 9). Lower $p$ corresponds with lower temperature.

The following convergence rate depends on bounds on the exploitability in terms of the loss (Lemma 14), bounds on the magnitude of estimates of the loss (Lemma 8), Lipschitz bounds on the infinity norm of the gradient (Corollary 2), and the number of distinct strategies ($n\bar{m} = \sum_k m_k$).

**Theorem 1** (BLiN PAC Rate). *Assume $\eta_k = \eta = 2/\hat{L}$, $\tau = \frac{1}{\ln(1/p)}$, and a previously pulled arm is returned uniformly at random (i.e., $t \sim U([T])$). Then for any $w > 0$*

$$\epsilon_t \leq w\Big[\frac{n}{\ln(1/p)}\big(W(1/e) + \frac{\bar{m} - 2}{e}\big) + 4(1 + (4c^2)^{1/3})\sqrt{nm^*\hat{L}}\Big(\frac{\ln T}{T}\Big)^{\frac{1}{2(d_z+2)}}\Big] \qquad (13)$$

*with probability $(1 - w^{-1})(1 - 2T^{-2})$ where $W$ is the Lambert function ($W(1/e) \approx 0.278$), $m^* = \max_k m_k$, $c \leq \frac{1}{4}\frac{n\bar{m}}{\hat{L}}\Big(\frac{\ln(m^*)}{\ln(1/p)} + 2\Big)^2 \leq \frac{1}{4}\Big(\frac{\ln(m^*)}{\ln(1/p)} + 2\Big)$ upper bounds the range of stochastic estimates of $\mathcal{L}^\tau$ (see Lemma 8), and $\hat{L} = \Big(\frac{\ln(m^*)}{\ln(1/p)} + 2\Big)\Big(\frac{m^{*2}}{p\ln(1/p)} + n\bar{m}\Big)$ (see Corollary 2).*

This result depends on the *near-optimality* [37] or *zooming*-dimension $d_z = n\bar{m}\big(\frac{\alpha_{hi} - \alpha_{lo}}{\alpha_{lo}\alpha_{hi}}\big) \in [0, \infty)$ (Theorem 2) where $\alpha_{lo}$ and $\alpha_{hi}$ denote the degree of the polynomials that lower and upper bound the function $\mathcal{L}^\tau \circ s$ locally around an equilibrium. For example, in the case where the Hessian is positive definite, $\alpha_{lo} = \alpha_{hi} = 2$ and $d_z = 0$. Here, $s : [0, 1]^{n(\bar{m}-1)} \to \prod_i \Delta^{m_i-1}$ is any function that maps from the unit hypercube to a product of simplices; we analyze two such maps in the appendix.

---

[2]Zhou et al. [39] developed a similar approach but only for pure Nash equilibria.

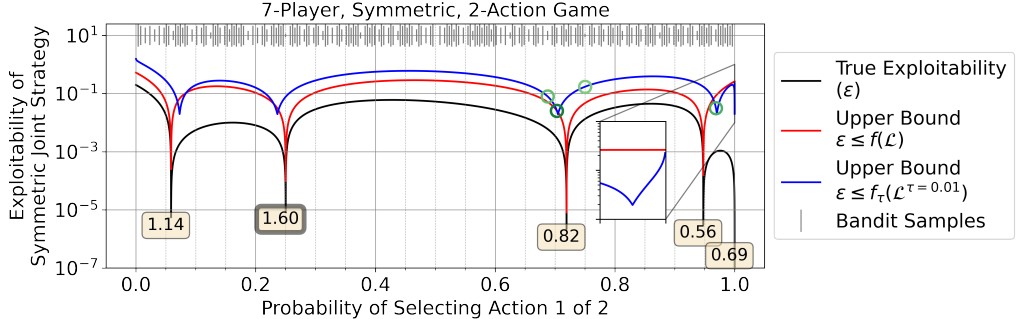

Figure 4: Bandit-based (BLiN) Nash solver applied to an artificial 7-player, symmetric, 2-action game. We search for a symmetric equilibrium, which is represented succinctly as the probability of selecting action 1. The plot shows the true exploitability $\epsilon$ of all symmetric strategies in black and indicates there exist potentially 5 NEs (the dips in the curve). Upper bounds on our unregularized loss $\mathcal{L}$ capture 4 of these equilibria, missing only the pure NE on the right. By considering our regularized loss, $\mathcal{L}^\tau$, we are able to capture this pure NE (see zoomed inset). The bandit algorithm selects strategies to evaluate, using 10 Monte-Carlo samples for each evaluation (arm pull) of $\mathcal{L}^\tau$. These samples are displayed as vertical bars above with the height of the vertical bar representing additional arm pulls. The best arms throughout search are denoted by green circles (darker indicates later in the search). The boxed numbers near equilibria display the welfare of the strategy.

Note that Theorem 1 implies that for games whose corresponding $\mathcal{L}^\tau$ has zooming dimension $d_z = 0$, NEs can be approximated with high probability in polynomial time. This general property is difficult to translate concisely into game theory parlance. For this reason, we present the following more interpretable corollary which applies to a more restricted class of games.

**Corollary 1.** *Consider the class of NFGs with at least one QRE($\tau$) whose local polymatrix approximation indicates it is isolated (i.e., $M$ from equation (12) is rank-$n\bar{m}$ implies Hess $\succ 0$ implies $d_z = n\bar{m}(\frac{2-2}{4}) = 0$). Then by Theorem 1, BLiN is a fully polynomial-time randomized approximation scheme (FPRAS) for QREs and is a PRAS for NEs of games in this class.*

To convey the impact of stochastic optimization guarantees more concretely, assume we are given that an interior well-isolated NE exists. Then for a 20-player, 50-action game, it is $1000\times$ cheaper to compute a $1/100$-NE with probability 95% than it is to just list the $nm^n$ payoffs that define the game.

### 6.2 Empirical Evaluation

Figure 3 shows SGD is competitive with scalable techniques to approximating NEs. Shapley's game induces a strongly convex $\mathcal{L}$ (see Section 5) leading to SGD's strong performance. Blotto shows signs of convergence to low, but nonzero $\epsilon$, demonstrating the challenges of local minima.

We demonstrate BLiN (applied to $\mathcal{L}^\tau$) on a 7-player, symmetric, 2-action game. Figure 4 shows the bandit algorithm discovers two equilibria, settling on one near $\boldsymbol{x} = [0.7, 0.3] \times 7$ with a wider basin of attraction (and higher welfare). In theory, BLiN can enumerate all NEs as $T \to \infty$.

## 7 Conclusion

In this work, we proposed a stochastic loss for approximate Nash equilibria in normal-form games. An unbiased loss estimator of Nash equilibria is the "key" to the stochastic optimization "door" which holds a wealth of research innovations uncovered over several decades. Thus, it allows the development of new algorithmic techniques for computing equilibria. We consider bandit and vanilla SGD methods in this work, but theses are only two of the many options now at our disposal (e.g, adaptive methods [1], Gaussian processes [9], evolutionary algorithms [17], etc.). Such approaches as well as generalizations of these techniques to imperfect-information games are promising directions for future work. Similarly to how deep learning research first balked at and then marched on to train neural networks via NP-hard non-convex optimization, we hope computational game theory can march ahead to make useful equilibrium predictions of large multiplayer systems.

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
