## A Loss: Connection to Exploitability, Unbiased Estimation, and Upper Bounds

### A.1 KKT Conditions Imply Fixed Point Sufficiency

Consider the following constrained optimization problem:

$$\max_{\boldsymbol{x} \in \mathbb{R}^d} f(\boldsymbol{x}) \tag{14}$$

$$s.t. g_i(\boldsymbol{x}) \leq 0 \ \forall i \tag{15}$$

$$h_j(\boldsymbol{x}) = 0 \ \forall j \tag{16}$$

where $f$ is concave and $g_i$ and $h_j$ represent inequality and equality constraints respectively. If $g_i$ and $h_i$ are affine functions, then any maximizer $\boldsymbol{x}^*$ of $f$ must satisfy the following KKT conditions (necessary and sufficient):

- Stationarity: $0 \in \partial f(\boldsymbol{x}^*) - \sum_j \lambda_j \partial h_j(\boldsymbol{x}^*) - \sum_i \mu_i \partial g_i(\boldsymbol{x}^*)$

- Primal feasibility: $h_j(\boldsymbol{x}^*) = 0$ for all $j$ and $g_i(\boldsymbol{x}^*) \leq 0$ for all $i$

- Dual feasibility: $\mu_i \geq 0$ for all $i$

- Complementary slackness: $\mu_i g_i(\boldsymbol{x}^*) = 0$ for all $i$.

**Lemma 1.** *Assuming player $i$'s utility, $u_i(x_i, x_{-i})$, is concave in its own strategy $x_i$, any best response in the interior of the simplex has zero projected-gradient norm:*

$$z^* \in \left( int\Delta \cup \arg\max_z u_i(z, x_{-i}) - u_i(x_i, x_{-i}) \right) \iff (z^* \in int\Delta) \wedge (||\Pi_\Delta[\nabla_{z^*}^i]|| = 0). \tag{17}$$

*Proof.* Consider the problem of formally computing $exp_i(\boldsymbol{x}) = \max_{z \in int\Delta} u_i(z, x_{-i}) - u_i(x_i, x_{-i})$:

$$\max_{z \in \mathbb{R}^d} u_i(z, x_{-i}) - u_i(x_i, x_{-i}) \tag{18}$$

$$s.t. -z_i + x_{\min} \leq 0 \ \forall i \tag{19}$$

$$1 - \sum_i z_i = 0. \tag{20}$$

where $x_{\min} > 0$ is some constant that captures our given assumption that the solution $z^*$ lies in the interior of the simplex. Note that the objective is linear (concave) in $z$ and the constraints are affine, therefore the KKT conditions are necessary and sufficient for optimality. Mapping the KKT conditions onto this problem yields the following:

- Stationarity: $0 \in \partial u_i(z^*, x_{-i}) + \lambda \mathbf{1} + \sum_i \mu_i e_i$

- Primal feasibility: $\sum_i z_i^* = 1$ and $z_i^* \geq x_{\min}$ for all $i$

- Dual feasibility: $\mu_i \geq 0$ for all $i$

- Complementary slackness: $\mu_i z_i^* = 0$ for all $i$.

For any point $z \in int\Delta$, primal feasibility will be satisfied for some $x_{\min} > 0$. This implies each $z_j$ is strictly positive. By complementary slackness and dual feasibility, each $\mu_i$ must be identically zero. This implies the stationarity condition can be simplified to $0 \in \partial u_i(z^*, x_{-i}) + \lambda \mathbf{1}$. Rearranging terms we find that for any $z^*$, there exists a $\lambda$ such that

$$\partial u_i(z^*, x_{-i}) \in \lambda \mathbf{1}. \tag{21}$$

Equivalently, $\partial u_i(z^*, x_{-i}) \propto \mathbf{1}$ at $z^* \in int\Delta$. Any vector proportional to the ones vector has zero projected-gradient norm, completing the claim. $\square$

## A.2 Norm of Projected-Gradient and Equivalence to NFG Exploitability with Approximate Best Responses

**Proposition 1.** *The loss $\mathcal{L}$ is equivalent to* `NashConv`*, but where player $k$'s best response is approximated by a single step of projected-gradient ascent with step size $\eta_k$:* $\mathtt{aBR}_k = x_k + \eta_k \Pi_\Delta[\nabla_{x_k}^k]$.

*Proof.* Define an approximate best response as the result of a player adjusting their strategy via a projected-gradient ascent step, i.e., $\mathtt{aBR}_k = x_k + \eta_k \Pi_\Delta[\nabla_{x_k}^k]$ for player $k$.

In a normal form game, player $k$'s utility at this new strategy is $u_k(\mathtt{aBR}_k, x_{-k}) = (\nabla_{x_k}^k)^\top (x_k + \eta_k \Pi_\Delta[\nabla_{x_k}^k]) = u_k(\boldsymbol{x}) + \eta_k (\nabla_{x_k}^k)^\top \Pi_\Delta[\nabla_{x_k}^k]$.

Therefore, the amount player $k$ gains by playing $\mathtt{aBR}$ is

$$\hat{\epsilon}_k(\boldsymbol{x}) = u_k(\mathtt{aBR}_k, x_{-k}) - u_k(\boldsymbol{x}) \tag{22}$$

$$= \eta_k (\nabla_{x_k}^k)^\top \Pi_\Delta[\nabla_{x_k}^k] \tag{23}$$

$$= \eta_k (\nabla_{x_k}^k - \frac{1}{m_k}(\mathbf{1}^\top \nabla_{x_k}^k)\mathbf{1})^\top \Pi_\Delta[\nabla_{x_k}^k] \tag{24}$$

$$= \eta_k ||\Pi_\Delta[\nabla_{x_k}^k]||^2 \tag{25}$$

where the third equality follows from the fact that the projected-gradient, $\Pi_\Delta[\nabla_{x_k}^k]$, is orthogonal to the ones vector. $\square$

## A.3 Connection to True Exploitability

**Lemma 2.** *The amount a player can gain by deviating is upper bounded by a quantity proportional to the norm of the projected-gradient:*

$$\epsilon_k(\boldsymbol{x}) \leq \sqrt{2}||\Pi_\Delta(\nabla_{x_k}^k)||. \tag{26}$$

*Proof.* Let $z$ be any point on the simplex. Then

$$u_k(z, x_{-k}) - u_k(\boldsymbol{x}) \leq (\nabla_{x_k}^k)^\top (z - x_k) \tag{27}$$

$$= (\nabla_{x_k}^k)^\top (z - x_k) - \frac{1}{m_k}(\mathbf{1}^\top \nabla_{x_k}^k) \overbrace{\mathbf{1}^\top (z - x_k)}^{1-1=0} \tag{28}$$

$$= (\Pi_\Delta[\nabla_{x_k}^k])^\top (z - x_k) \tag{29}$$

$$\leq \sqrt{2}||\Pi_\Delta(\nabla_{x_k}^k)||. \tag{30}$$

$\square$

Continuing, we can prove a bound on NashConv in terms of projected-gradient loss:

**Lemma 3.** *The exploitability, $\epsilon$, of a joint strategy $\boldsymbol{x}$, is upper bounded as a function of our proposed loss:*

$$\epsilon \leq \sqrt{\frac{2n}{\min_k \eta_k}} \sqrt{\mathcal{L}(\boldsymbol{x})}. \tag{31}$$

*Proof.*

$$\epsilon = \max_k \max_z u_k(z, x_{-k}) - u_k(\boldsymbol{x}) \tag{32}$$

$$\leq \sum_k \max_z u_k(z, x_{-k}) - u_k(\boldsymbol{x}) \tag{33}$$

$$\leq \sum_k \sqrt{2} ||\Pi_\Delta(\nabla_{x_k}^k)||_2 \tag{34}$$

$$= \sqrt{2} \left|\left| ||\Pi_\Delta(\nabla_{x_1}^1)||_2, \ldots \sqrt{2} ||\Pi_\Delta(\nabla_{x_n}^n)||_2 \right|\right|_1 \tag{35}$$

$$\leq \sqrt{2n} \left|\left| ||\Pi_\Delta(\nabla_{x_1}^1)||_2, \ldots ||\Pi_\Delta(\nabla_{x_n}^n)||_2 \right|\right|_2 \tag{36}$$

$$= \sqrt{2n} \sqrt{\sum_k ||\Pi_\Delta(\nabla_{x_k}^k)||_2^2} \tag{37}$$

$$\leq \sqrt{2n} \sqrt{\sum_k \left(\frac{1}{\eta_k}\right) \eta_k ||\Pi_\Delta(\nabla_{x_k}^k)||_2^2} \tag{38}$$

$$\leq \sqrt{\frac{2n}{\min_k \eta_k}} \sqrt{\sum_k \eta_k ||\Pi_\Delta(\nabla_{x_k}^k)||_2^2} \tag{39}$$

$$= \sqrt{\frac{2n}{\min_k \eta_k}} \sqrt{\mathcal{L}(\boldsymbol{x})} \tag{40}$$

$\square$

**Lemma 4.** *The entropy regularized exploitability, $\epsilon_{QRE}$, of a joint strategy $\boldsymbol{x}$, is upper bounded as a function of our proposed loss:*

$$\epsilon_{QRE} \leq \sqrt{\frac{2n}{\min_k \eta_k}} \sqrt{\mathcal{L}^\tau(\boldsymbol{x})}. \tag{41}$$

*Proof.* Recall that $u_k^\tau(x_k, x_{-k})$ is also concave with respect to $x_k$. Then

$$\epsilon_{QRE} = \max_k \max_z u_k^\tau(z, x_{-k}) - u_k^\tau(\boldsymbol{x}) \tag{42}$$

$$\leq \sum_k \max_z u_k^\tau(z, x_{-k}) - u_k^\tau(\boldsymbol{x}) \tag{43}$$

$$\leq \sum_k \sqrt{2} ||\Pi_\Delta(\nabla_{x_k}^{k\tau})||_2 \tag{44}$$

$$= \sqrt{2} \left|\left| ||\Pi_\Delta(\nabla_{x_1}^{1\tau})||_2, \ldots \sqrt{2} ||\Pi_\Delta(\nabla_{x_n}^{n\tau})||_2 \right|\right|_1 \tag{45}$$

$$\leq \sqrt{2n} \left|\left| ||\Pi_\Delta(\nabla_{x_1}^{1\tau})||_2, \ldots ||\Pi_\Delta(\nabla_{x_n}^{n\tau})||_2 \right|\right|_2 \tag{46}$$

$$= \sqrt{2n} \sqrt{\sum_k ||\Pi_\Delta(\nabla_{x_k}^{k\tau})||_2^2} \tag{47}$$

$$\leq \sqrt{2n} \sqrt{\sum_k \left(\frac{1}{\eta_k}\right) \eta_k ||\Pi_\Delta(\nabla_{x_k}^{k\tau})||_2^2} \tag{48}$$

$$\leq \sqrt{\frac{2n}{\min_k \eta_k}} \sqrt{\sum_k \eta_k ||\Pi_\Delta(\nabla_{x_k}^{k\tau})||_2^2} \tag{49}$$

$$= \sqrt{\frac{2n}{\min_k \eta_k}} \sqrt{\mathcal{L}^\tau(\boldsymbol{x})} \tag{50}$$

$\square$

 ## A.4 Unbiased Estimation

481 **Lemma 5.** *An unbiased estimate of $\mathcal{L}(\boldsymbol{x})$ can be obtained by drawing two samples (pure strategies)*
482 *from each players' mixed strategy and observing payoffs.*

483 *Proof.* Define $\nabla_{x_k}^k$ as the random variable distributed according to the distribution induced by all
484 players' mixed strategies. Let $\nabla_{x_k}^{k(1)}$ and $\nabla_{x_k}^{k(2)}$ represent two other independent random variables,
485 distributed identically to $\nabla_{x_k}^k$. Then

$$\mathbb{E}_{a_k \sim x_k \forall k}[\mathcal{L}(\boldsymbol{x})] = \mathbb{E}_{a_k \sim x_k \forall k}[\sum_k \eta_k(||\nabla_{x_k}^k||^2 - \frac{1}{m_k}(\mathbf{1}^\top \nabla_{x_k}^k)^2)] \tag{51}$$

$$= \sum_k \eta_k(\mathbb{E}_{a_k \sim x_k \forall k}[||\nabla_{x_k}^k||^2] - \frac{1}{m_k}\mathbb{E}_{a_k \sim x_k \forall k}[(\mathbf{1}^\top \nabla_{x_k}^k)^2]) \tag{52}$$

$$= \sum_k \eta_k(\mathbb{E}_{a_k \sim x_k \forall k}[\sum_l (\nabla_{x_{kl}}^k)^2] - \frac{1}{m_k}\mathbb{E}_{a_k \sim x_k \forall k}[(\sum_l \nabla_{x_{kl}}^k)^2]) \tag{53}$$

$$= \sum_k \eta_k(\sum_l \mathbb{E}_{a_k \sim x_k \forall k}[(\nabla_{x_{kl}}^k)^2] - \frac{1}{m_k}\mathbb{E}_{a_k \sim x_k \forall k}[(\sum_l \nabla_{x_{kl}}^k)^2]) \tag{54}$$

$$= \sum_k \eta_k\Big( \sum_l \mathbb{E}_{a_k \sim x_k \forall k}[\nabla_{x_{kl}}^{k(1)}]\mathbb{E}_{a_k \sim x_k \forall k}[\nabla_{x_{kl}}^{k(2)}] \tag{55}$$

$$- \frac{1}{m_k}\mathbb{E}_{a_k \sim x_k \forall k}[\sum_l \nabla_{x_{kl}}^{k(1)}]\mathbb{E}_{a_k \sim x_k \forall k}[\sum_l \nabla_{x_{kl}}^{k(2)}]\Big) \tag{56}$$

$$= \sum_k \eta_k\Big( \sum_l \mathbb{E}_{a_j \sim x_j \forall j \neq k}[\nabla_{x_{kl}}^{k(1)}]\mathbb{E}_{a_j \sim x_j \forall j \neq k}[\nabla_{x_{kl}}^{k(2)}] \tag{57}$$

$$- \frac{1}{m_k}\mathbb{E}_{a_j \sim x_j \forall j \neq k}[\sum_l \nabla_{x_{kl}}^{k(1)}]\mathbb{E}_{a_j \sim x_j \forall j \neq k}[\sum_l \nabla_{x_{kl}}^{k(2)}]\Big) \tag{58}$$

$$= \sum_k \eta_k\Big( \big[\hat{\nabla}_{x_k}^{k(1)}\big]^\top \hat{\nabla}_{x_k}^{k(2)} - \frac{1}{m_k}(\mathbf{1}^\top \hat{\nabla}_{x_k}^{k(1)})(\mathbf{1}^\top \hat{\nabla}_{x_k}^{k(2)})\Big) \tag{59}$$

$$= \sum_k \eta_k\underbrace{(\hat{\nabla}_{x_k}^{k(1)} - \frac{1}{m_k}(\mathbf{1}^\top \hat{\nabla}_{x_k}^{k(1)})\mathbf{1})^\top}_{\text{appx. br gradient}} \underbrace{\hat{\nabla}_{x_k}^{k(2)}}_{\text{exp. payoffs}} \tag{60}$$

486 where $\hat{\nabla}_{x_k}^{k(p)}$ is an unbiased estimator of player $k$'s gradient.

487 □

488 **Lemma 6.** *The loss formed as the sum of the squared norms of the projected-gradients, $\mathcal{L}^\tau$, can be*
489 *decomposed into three terms as follows:*

$$\mathcal{L}^\tau(\boldsymbol{x}) = \underbrace{\sum_k \eta_k x_q^\top B_{kq}^\top B_{kq} x_q}_{(A)} + 2\underbrace{\sum_k \eta_k E_k^\top B_{kq} x_q}_{(B)} + \underbrace{\sum_k \eta_k E_k^\top E_k}_{(C)} \tag{61}$$

490 *where q is any player other than k.*

*Proof.* Let $S^\tau = -\tau \sum_l x_{kl} \log(x_{kl})$ so that $\frac{\partial S^\tau}{\partial x_k} = -\tau(\ln(x_k) + \mathbf{1})$. Note that $\Pi_{T\Delta}[\frac{\partial S^\tau}{\partial x_k}] = -\tau \ln(x_k)$.

$$\mathcal{L}^\tau(\boldsymbol{x}) = \sum_k \eta_k (\Pi_{T\Delta}[\nabla_{x_k}^k])^\top \Pi_{T\Delta}[\nabla_{x_k}^k] \tag{62}$$

$$= \sum_k \eta_k [H_{kq}^k x_q + \frac{\partial S^\tau}{\partial x_k}]^\top [I - \frac{1}{m_k}\mathbf{1}\mathbf{1}^\top][I - \frac{1}{m_k}\mathbf{1}\mathbf{1}^\top][H_{kq}^k x_q + \frac{\partial S^\tau}{\partial x_k}] \tag{63}$$

$$= \sum_k \eta_k \Big( x_q^\top [H_{kq}^k]^\top [I - \frac{1}{m_k}\mathbf{1}\mathbf{1}^\top]^2 [H_{kq}^k] x_q + 2[\frac{\partial S^\tau}{\partial x_k}]^\top [I - \frac{1}{m_k}\mathbf{1}\mathbf{1}^\top]^2 [H_{kq}^k x_q] \tag{64}$$

$$+ [\frac{\partial S^\tau}{\partial x_k}]^\top [I - \frac{1}{m_k}\mathbf{1}\mathbf{1}^\top]^2 [\frac{\partial S^\tau}{\partial x_k}] \Big) \tag{65}$$

$$= \underbrace{\sum_k \eta_k x_q^\top B_{kq}^\top B_{kq} x_q}_{(A)} + \underbrace{2\sum_k \eta_k E_k^\top B_{kq} x_q}_{(B)} + \underbrace{\sum_k \eta_k E_k^\top E_k}_{(C)} \tag{66}$$

where $B_{kq} = [I - \frac{1}{m_k}\mathbf{1}\mathbf{1}^\top]H_{kq}^k$ and $E_k = [I - \frac{1}{m_k}\mathbf{1}\mathbf{1}^\top][\frac{\partial S^\tau}{\partial x_k}] = -\tau \ln(x_k)$.

$\square$

## A.5 Bound on Loss

By equation (51), we can also rewrite this loss as a weighted sum of 2-norms, $\mathcal{L}(\boldsymbol{x}) = \sum_k \eta_k ||\nabla_{x_k}^k - \mu_k||_2^2$ where $\mu_k = \frac{1}{m_k}(\mathbf{1}^\top \nabla_{x_k}^k) \in [0, 1]$ for brevity. This will allow us to more easily analyze our loss.

**Lemma 7.** *Assume payoffs are bounded by 1, then setting $\eta_k \leq \frac{4}{nm_k}$ or $\eta_k \leq \frac{4}{n\bar{m}}$ or $\sum_k \eta_k \leq \frac{4}{\bar{m}}$ ensures $0 \leq \mathcal{L}(x) \leq 1$ for all $x \in \mathcal{X}$.*

*Proof.*

$$0 \leq \mathcal{L}(\boldsymbol{x}) = \sum_k \eta_k ||\nabla_{x_k}^k - \mu_k||_2^2 \tag{67}$$

$$= \sum_k \eta_k m_k \Big[ \frac{1}{m_k} \sum_l (\nabla_{x_{kl}}^k - \mu_k)^2 \Big] \tag{68}$$

$$= \sum_k \eta_k m_k Var[\nabla_{x_k}^k] \tag{69}$$

$$\leq \frac{1}{4} \sum_k \eta_k m_k \tag{70}$$

$$\leq \frac{1}{4} (\max_k \eta_k)(\sum_k m_k) \tag{71}$$

$$= \frac{1}{4} (\max_k \eta_k) n\bar{m} \leq 1 \tag{72}$$

$$\implies (\max_k \eta_k) \leq \frac{4}{n\bar{m}}. \tag{73}$$

$\square$

The $k$th element of the sum in the loss does not depend on agent $k$'s strategy. We will rewrite the loss to make its dependence on all other players' strategies more obvious ($l, q \neq k$ below).

$$\mathcal{L}(\boldsymbol{x}) = \sum_k \eta_k \big( [H_{kl}^k x_l]^\top [H_{kq}^k x_q] - \frac{1}{m_k} (\mathbf{1}^\top [H_{kl}^k x_l])(\mathbf{1}^\top [H_{kq}^k x_q]) \big) \tag{74}$$

$$= \sum_k \eta_k \big( [H_{kl}^k x_l]^\top [H_{kq}^k x_q] - \frac{1}{m_k} [H_{kl}^k x_l]^\top \mathbf{1}\mathbf{1}^\top [H_{kq}^k x_q] \big) \tag{75}$$

$$= \sum_k \eta_k [H_{kl}^k x_l]^\top [I - \frac{1}{m_k} \mathbf{1}\mathbf{1}^\top][H_{kq}^k x_q] \tag{76}$$

$$= \sum_k \eta_k x_l^\top [H_{kl}^k]^\top [I - \frac{1}{m_k} \mathbf{1}\mathbf{1}^\top][H_{kq}^k] x_q \tag{77}$$

$$= \sum_k \eta_k x_q^\top [H_{kq}^k]^\top [I - \frac{1}{m_k} \mathbf{1}\mathbf{1}^\top][H_{kq}^k] x_q \quad \text{isolate dep. on } q \tag{78}$$

$$= \sum_k \eta_k x_q^\top [H_{qk}^k][I - \frac{1}{m_k} \mathbf{1}\mathbf{1}^\top][H_{kq}^k] x_q \tag{79}$$

$$= \sum_k \eta_k x_q^\top A_{qkq} x_q. \tag{80}$$

where $A_{qkq} = [H_{qk}^k][I - \frac{1}{m_k}\mathbf{1}\mathbf{1}^\top][H_{kq}^k]$ does not depend on $x_k$.

Note this means we can also write $\mathcal{L}(\boldsymbol{x}) = \sum_k \eta_k x_l^\top A_{lkq} x_q$ for any $l, q \neq k$.

**Lemma 8.** *Assume payoffs are bounded in* $[0, 1]$*, then*

$$|\mathcal{L}^\tau(\boldsymbol{x})| \leq \frac{1}{4}(\max_k \eta_k) n \bar{m} \Big( \frac{\ln(m^*)}{\ln(1/p)} + 2 \Big)^2 \tag{81}$$

*for any* $\boldsymbol{x}$ *such that* $x_{kl} \geq \frac{p}{m^*} \ \forall k, l$.

*Proof.* Starting from the definition of $\mathcal{L}^\tau$ and applying Lemma 19 along with intermediate results from Lemma 16, we find

$$|\mathcal{L}^\tau(\boldsymbol{x})| = |\sum_k \eta_k ||\Pi_{T\Delta}(\nabla_{x_k}^k)||^2| \tag{82}$$

$$\leq \frac{1}{4} \sum_k \eta_k m_k (\tau \ln(\frac{1}{x_{\min}}) + 1)^2 \tag{83}$$

$$= \frac{1}{4} \sum_k \eta_k m_k \Big( \frac{1}{\ln(1/p)} \ln\big(\frac{m^*}{p}\big) + 1 \Big)^2 \tag{84}$$

$$= \frac{1}{4} \sum_k \eta_k m_k \Big( \frac{\ln(m^*)}{\ln(1/p)} + 2 \Big)^2 \tag{85}$$

$$= \frac{1}{4}(\max_k \eta_k) n \bar{m} \Big( \frac{\ln(m^*)}{\ln(1/p)} + 2 \Big)^2. \tag{86}$$

$\square$

# B QREs Approximate NEs at Low Temperature

**Lemma 9.** *Setting* $\tau = \ln(1/p)^{-1}$ *with* $p \in [0, 1)$ *ensures that all QREs contain probabilities greater than* $\frac{p}{\max_k m_k}$.

*Proof.* What must $\tau$ be to ensure $x_{lp} \geq x_{\min}$ for any $l, p$? We can check the case where $\nabla = e_i$. Let $m^* = \max_k m_k$. Then

$$x_{\min} = \min_k \min_{\nabla_{x_k}^k} \min_l \left[ \texttt{softmax}\Big(\frac{\nabla_{x_k}^k}{\tau}\Big)\right]_l \tag{87}$$

$$= \frac{e^0}{(m^* - 1)e^{\frac{1}{\tau}} + e^0} \tag{88}$$

$$= \frac{1}{(m^* - 1)e^{\frac{1}{\tau}} + 1} \tag{89}$$

$$\implies e^{\frac{1}{\tau}} = \frac{1}{m^* - 1}\Big(\frac{1}{x_{\min}} - 1\Big) \tag{90}$$

$$\implies \tau = \frac{1}{\ln\Big(\frac{1}{m^* - 1}\big(\frac{1}{x_{\min}} - 1\big)\Big)}. \tag{91}$$

If $x_{\min} = \frac{p}{m^*}$ with $p \in [0, 1]$, then

$$\tau^* = \frac{1}{\ln\Big(\frac{1}{m^* - 1}\big(\frac{1}{x_{\min}} - 1\big)\Big)} \tag{92}$$

$$= \frac{1}{\ln\Big(\frac{1}{m^* - 1}\big(\frac{m^*}{p} - 1\big)\Big)} \tag{93}$$

$$= \frac{1}{\ln\Big(\frac{m^* - p}{m^* - 1}\frac{1}{p}\Big)} \tag{94}$$

$$\leq \frac{1}{\ln(\frac{1}{p})}. \tag{95}$$

This implies if we set $\tau = \ln(1/p)^{-1}$, then we are guaranteed that all QREs contain probabilities greater than $x_{\min} = \frac{p}{\max_k m_k}$. $\qquad\square$

**Lemma 10** (Repeated from Lemma 1 of [29]). *Let $\nabla_{x_k}^k$ be player $k$'s gradient ($m_k \geq 2$) with payoffs bounded in $[0, 1]$ and $\boldsymbol{x}$ be a QRE at temperature $\tau$. Then it holds that*

$$u_k(BR_k, x_{-k}) - u_k(\boldsymbol{x}) = \max(\nabla_{x_k}^k) - (\nabla_{x_k}^k)^\top \texttt{softmax}\Big(\frac{\nabla_{x_k}^k}{\tau}\Big) \leq \tau\Big(W(1/e) + \frac{m_k - 2}{e}\Big) \tag{96}$$

*where $W$ is the Lambert function ($W(1/e) \approx 0.278$).*

**Lemma 11** (Slightly modified from Proposition 5.1a of [6]). *Let $\psi_e(x_k) = \sum_l x_{kl} \ln(x_{kl})$ if $x_k \in \Delta^{m_k - 1}$ else $+\infty$. Then $\psi_e(x_k)$ is 1-strongly convex over $int\Delta^{m_k - 1}$ w.r.t. the $||\cdot||_1$ and $||\cdot||_2$ norms, i.e.,*

$$\langle \nabla\psi_e(x) - \nabla\psi_e(y), x - y \rangle \geq ||x - y||_1^2 \geq ||x - y||_2^2 \tag{97}$$

$$\implies \psi_e(y) \geq \psi_e(x) + \nabla\psi_e(x)^\top(y - x) + \frac{1}{2}||y - x||_2^2. \tag{98}$$

*for all $x, y \in int\Delta^{m_k - 1}$.*

**Lemma 12.** *Let $l(x|x_k) = \langle \nabla f_k(x_k), x \rangle + \frac{1}{t_k} B_{\psi_e}(x, x_k)$ where $t_k > 0$, $f_k(z) = -\epsilon_k(z) = -[u_k(z, x_{-i}) + S^\tau(z) - u_k(\boldsymbol{x}) - S^\tau(x_k)]$, and $B_{\psi_e}(x, x_k) = \psi_e(x) - \psi_e(y) - \langle x - y, \nabla\psi_e(y) \rangle$ with $\psi_e$ defined in Lemma 11. Finally, let $x_{k+1} = \arg\min_{x \in int\Delta} l(x|x_k)$. Then*

$$||x_k - x_{k+1}|| \leq 2||\Pi_\Delta(\nabla_{x_k}^{k\tau})||. \tag{99}$$

*Proof.* Plugging $\psi_e(x_k) = \sum_l x_{kl} \ln(x_{kl}) = -S(x_k)$ on $int\Delta$ into $B_{\psi_e}(x, x_k)$, we find

$$B_{\psi_e}(x, x_k) = S(x_k) - S(x) - \langle \ln(x_k) + 1, x - x_k \rangle \tag{100}$$

$$= S(x_k) - S(x) - \langle \ln(x_k), x - x_k \rangle \tag{101}$$

for all $x, x_k \in int\Delta$. Note that $-S(x)$ is 1-strongly convex on $int\Delta$, therefore, $B_{\psi_e}(x, x_k)$ is also 1-strongly convex in $x$. Continuing, this also implies $l(x|x_k)$ is 1-strongly convex.

Let $x_{k+1} = \arg\min_{x \in int\Delta} l(x|x_k)$ and note that $\nabla f_k(x_k) = -\nabla \epsilon_k = -\nabla_{x_k}^{k\tau}$. Strong convexity of $l$ implies

$$l(x_{k+1}) \geq l(x_k) + \nabla_x l(x_k)^\top (x_{k+1} - x_k) + \frac{1}{2}||x_{k+1} - x_k||_2^2 \tag{102}$$

$$\implies ||x_k - x_{k+1}||_2^2 \leq 2\Big[\underbrace{l(x_{k+1}) - l(x_k)}_{\leq 0} + \nabla_x l(x_k)^\top (x_k - x_{k+1})\Big] \tag{103}$$

$$\leq 2\nabla_x l(x_k)^\top (x_k - x_{k+1}) = 2(\nabla f_k(x_k) + \frac{1}{t_k}[\ln(x_k) + \mathbf{1} - \ln(x_k)])^\top (x_k - x_{k+1}) \tag{104}$$

$$= 2\nabla f_k(x_k)^\top (x_k - x_{k+1}) = 2(\nabla_{x_k}^{k\tau})^\top (x_{k+1} - x_k) = 2(\Pi_\Delta(\nabla_{x_k}^{k\tau}))^\top (x_{k+1} - x_k) \tag{105}$$

$$\leq 2||\Pi_\Delta(\nabla_{x_k}^{k\tau})|| \, ||x_k - x_{k+1}||. \tag{106}$$

Rearranging the inequality achieves the desired result. $\qquad\square$

**Lemma 13.** *[Low Temperature Approximate QREs are Approximate Nash Equilibria] Let $\nabla_{x_k}^{k\tau}$ be player $k$'s entropy regularized gradient with payoffs bounded in $[0, 1]$ and $\boldsymbol{x}$ be an approximate QRE. Then it holds that*

$$u_k(BR_k, x_{-k}) - u_k(\boldsymbol{x}) \leq \tau(W(1/e) + \frac{m_k - 2}{e}) + 2\sqrt{m_k}||\Pi_\Delta(\nabla_{x_k}^{k\tau})|| \tag{107}$$

*where $W$ is the Lambert function ($W(1/e) \approx 0.278$).*

*Proof.* First note that $x_k = \mathtt{softmax}(\ln(x_k))$ for $x_k \in int\Delta$. Recall that the $\mathtt{softmax}$ is invariant to constant offsets to its argument, i.e., $\mathtt{softmax}(z + c\mathbf{1}) = \mathtt{softmax}(z)$ for any $c \in \mathbb{R}$. Then

$$\mathtt{softmax}(\frac{\nabla_{x_k}^k}{\tau}) = \mathtt{softmax}(\ln(x_k) - \frac{1}{\tau}[-\nabla_{x_k}^k + \tau\ln(x_k)]) \tag{108}$$

$$= \mathtt{softmax}(\ln(x_k) - \frac{1}{\tau}[-\nabla_{x_k}^k + \tau\ln(x_k) + \tau\mathbf{1}]) \tag{109}$$

$$= \mathtt{softmax}(\ln(x_k) - \frac{1}{\tau}\nabla f_k(x_k)) \tag{110}$$

$$= \arg\min_{x \in int\Delta} l(x|x_k) \text{ with } t_k = \frac{1}{\tau} \tag{111}$$

$$= x_k^* \tag{112}$$

where the closed-form solution to the minimization problem as a $\mathtt{softmax}$ formula comes from inspecting the Entropic Descent Algorithm (EDA) of [6].

Then, beginning with the definition of exploitability, we find

$$u_k(BR_k, x_{-k}) - u_k(\boldsymbol{x}) = u_k(BR_k, x_{-k}) - (\nabla_{x_k}^k)^\top x_k \tag{113}$$

$$= u_k(BR_k, x_{-k}) - (\nabla_{x_k}^k)^\top \mathtt{softmax}(\frac{\nabla_{x_k}^k}{\tau}) - (\nabla_{x_k}^k)^\top (x_k - \mathtt{softmax}(\frac{\nabla_{x_k}^k}{\tau})) \tag{114}$$

$$\leq \tau(W(1/e) + \frac{m_k - 2}{e}) + ||\nabla_{x_k}^k|| \cdot ||x_k - \mathtt{softmax}(\frac{\nabla_{x_k}^k}{\tau})|| \tag{115}$$

$$= \tau(W(1/e) + \frac{m_k - 2}{e}) + ||\nabla_{x_k}^k|| \cdot ||x_k - x_k^*|| \tag{116}$$

$$\leq \tau(W(1/e) + \frac{m_k - 2}{e}) + 2\sqrt{m_k}||\Pi_\Delta(\nabla_{x_k}^{k\tau})||. \tag{117}$$

$\qquad\square$

**Lemma 14.** *[$\mathcal{L}^\tau$ Scores Nash Equilibria] Let $\mathcal{L}^\tau(\boldsymbol{x})$ be our proposed entropy regularized loss function with payoffs bounded in $[0,1]$ and $\boldsymbol{x}$ be an approximate QRE. Then it holds that*

$$\epsilon \leq n\tau(W(1/e) + \frac{\bar{m}-2}{e}) + 2\sqrt{\frac{n\max_k m_k}{\min_k \eta_k}}\sqrt{\mathcal{L}^\tau(\boldsymbol{x})} \tag{118}$$

*where $W$ is the Lambert function ($W(1/e) \approx 0.278$).*

*Proof.* Beginning with the definition of exploitability and applying Lemma 13, we find

$$\epsilon = \max_k u_k(\mathrm{BR}_k, x_{-k}) - u_k(\boldsymbol{x}) \tag{119}$$

$$\leq \sum_k u_k(\mathrm{BR}_k, x_{-k}) - u_k(\boldsymbol{x}) \tag{120}$$

$$\leq \sum_k \left[\tau(W(1/e) + \frac{m_k-2}{e}) + 2\sqrt{m_k}||\Pi_\Delta(\nabla_{x_k}^{k\tau})||\right] \tag{121}$$

$$= n\tau(W(1/e) + \frac{\bar{m}-2}{e}) + 2\sum_k \sqrt{m_k}||\Pi_\Delta(\nabla_{x_k}^{k\tau})|| \tag{122}$$

$$\leq n\tau(W(1/e) + \frac{\bar{m}-2}{e}) + 2\sqrt{\max_k m_k}\sum_k ||\Pi_\Delta(\nabla_{x_k}^{k\tau})|| \tag{123}$$

$$\leq n\tau(W(1/e) + \frac{\bar{m}-2}{e}) + 2\sqrt{\frac{n\max_k m_k}{\min_k \eta_k}}\sqrt{\mathcal{L}^\tau(\boldsymbol{x})}. \tag{124}$$

where the last inequality follows from the same steps outlined in Lemma 3, which established the relationship between $\mathcal{L}(\boldsymbol{x})$ and $\epsilon$.

$\square$

## C  Gradient of Loss

**Lemma 15.** *The gradient of $\mathcal{L}^\tau(\boldsymbol{x})$ with respect to player $l$'s strategy $x_l$ is*

$$\nabla_{x_l}\mathcal{L}(\boldsymbol{x}) = 2\sum_k \eta_k B_{kl}^\top \Pi_{T\Delta}(\nabla_{x_k}^{k\tau}) \tag{125}$$

*where $B_{ll} = -\tau[I - \frac{1}{m_l}\mathbf{1}\mathbf{1}^\top]\boldsymbol{diag}(\frac{1}{x_l})$ and $B_{kl} = [I - \frac{1}{m_k}\mathbf{1}\mathbf{1}^\top]H_{kl}^k$ for $k \neq l$.*

*Proof.* Recall from Lemma 6 that the loss can be decomposed as $\mathcal{L}^\tau(\boldsymbol{x}) = (A) + (B) + (C)$.

Then

$$D_{x_l}[(A)] = D_{x_l}[\sum_k \eta_k x_q^\top B_{kq}^\top B_{kq} x_q] = 2\sum_{k\neq l}\eta_k B_{kl}^\top B_{kl}x_l \tag{126}$$

where $q \neq k$ and $B_{kq} = [I - \frac{1}{m_k}\mathbf{1}\mathbf{1}^\top][H_{kq}^k]$ does not depend on $x_k$.

Also, letting $B_{ll} = -\tau[I - \frac{1}{m_l}\mathbf{1}\mathbf{1}^\top]\mathtt{diag}(\frac{1}{x_l})$,

$$D_{x_l}[(B)] = D_{x_l}[-2\tau\sum_k \eta_k \ln(x_k)^\top B_{kq}x_q] \tag{127}$$

$$= -2\tau\left[\eta_l D_{x_l}[\ln(x_l)^\top B_{lq}x_q] + \sum_{k\neq l}\eta_k D_{x_l}[\ln(x_k)^\top B_{kl}x_l]\right] \tag{128}$$

$$= -2\tau\left[\eta_l\mathtt{diag}(\frac{1}{x_l})B_{lq}x_q + \sum_{k\neq l}\eta_k B_{kl}^\top \ln(x_k)\right] \tag{129}$$

$$= -2\tau\left[\eta_l([I - \frac{1}{m_l}\mathbf{1}\mathbf{1}^\top]\mathtt{diag}(\frac{1}{x_l}))^\top \Pi_{T\Delta}(\nabla^l) + \sum_{k\neq l}\eta_k B_{kl}^\top \ln(x_k)\right] \tag{130}$$

$$= 2\left[\eta_l B_{ll}^\top \Pi_{T\Delta}(\nabla^l) - \tau\sum_{k\neq l}\eta_k B_{kl}^\top \ln(x_k)\right]. \tag{131}$$

And

$$D_{x_l}[(C)] = D_{x_l}[\sum_k \eta_k \tau^2 \ln(x_k)^\top [I - \frac{1}{m_k} \mathbf{1}\mathbf{1}^\top] \ln(x_k)] \tag{132}$$

$$= 2\tau^2 \Big[ \eta_l \mathtt{diag}(\frac{1}{x_l})[I - \frac{1}{m_l} \mathbf{1}\mathbf{1}^\top] \ln(x_l) \Big] \tag{133}$$

$$= -2\tau \eta_l ([I - \frac{1}{m_l} \mathbf{1}\mathbf{1}^\top] \mathtt{diag}(\frac{1}{x_l}))^\top \Pi_{T\Delta}(-\tau \ln(x_l)) \tag{134}$$

$$= 2\eta_l B_{ll}^\top \Pi_{T\Delta}(-\tau \ln(x_l)). \tag{135}$$

Putting these together, we find

$$\nabla_{x_l} \mathcal{L}(\boldsymbol{x}) = 2 \sum_{k \neq l} \eta_k B_{kl}^\top (B_{kl} x_l - \tau \ln(x_k)) + 2\eta_l B_{ll}^\top \big[ \Pi_{T\Delta}(\nabla^l) + \Pi_{T\Delta}(-\tau \ln(x_l)) \big] \tag{136}$$

$$= 2\eta_l B_{ll}^\top \Pi_{T\Delta}(\nabla_{x_k}^{k\tau}) + 2 \sum_{k \neq l} \eta_k B_{kl}^\top \Pi_{T\Delta}(\nabla_{x_k}^{k\tau}) \tag{137}$$

$$= 2 \sum_k \eta_k B_{kl}^\top \Pi_{T\Delta}(\nabla_{x_k}^{k\tau}). \tag{138}$$

$\square$

## C.1 Unbiased Estimation

In order to construct an unbiased estimate of $A_{lkl}$, we will need to form two independent unbiased estimates of $H_{kl}^k$. Recall that $H_{kl}^k$ is simply the expected bimatrix game between players $k$ and $l$ when all other players sample their actions according to their current strategies.

## C.2 Bound on Gradient / Lipschitz Property

**Lemma 16.** *Assume payoffs are upper bounded by 1, then the infinity norm of the gradient is bounded as*

$$||\nabla_{\boldsymbol{x}} \mathcal{L}^\tau(\boldsymbol{x})||_\infty \leq \frac{1}{2} (\max_k \eta_k)(\tau \ln\big(\frac{1}{x_{\min}}\big) + 1) \Big[ \tau m^* \big(\frac{1}{x_{\min}} - 1\big) + n\bar{m} \Big]. \tag{139}$$

*Proof.* Recall from Lemma 15 that the gradient of $\mathcal{L}(\boldsymbol{x})$ with respect to player $l$'s strategy $x_l$ is

$$\nabla_{x_l} \mathcal{L}(\boldsymbol{x}) = 2 \sum_k \eta_k B_{kl}^\top \Pi_{T\Delta}(\nabla_{x_k}^{k\tau}) \tag{140}$$

where $B_{ll} = -\tau[I - \frac{1}{m_l} \mathbf{1}\mathbf{1}^\top]\mathtt{diag}(\frac{1}{x_l})$ and $B_{kl} = [I - \frac{1}{m_k} \mathbf{1}\mathbf{1}^\top]H_{kl}^k$ for $k \neq l$.

For payoffs in $[0, 1]$, the entries in $\nabla_{x_k}^{k\tau} = \nabla_{x_k}^k - \tau \ln(x_k)$ are bounded within $[0, \tau \ln(\frac{1}{x_{\min}}) + 1]$ with a range $\tau \ln(\frac{1}{x_{\min}}) + 1$. Similarly, the entries in $-\tau \mathtt{diag}(\frac{1}{x_l})$ are bounded within $[-\tau \frac{1}{x_{\min}}, -\tau]$ with a range of $\tau(\frac{1}{x_{\min}} - 1)$.

The infinity norm of the gradient can then be bounded as

$$||\nabla_{\boldsymbol{x}}\mathcal{L}^\tau(\boldsymbol{x})||_\infty = \max_l ||\nabla_{x_l}\mathcal{L}(\boldsymbol{x})||_\infty \tag{141}$$

$$= \max_l ||2\sum_k \eta_k B_{kl}^\top \Pi_{T\Delta}(\nabla_{x_k}^{k\tau})||_\infty \tag{142}$$

$$\leq 2\sum_k \eta_k \max_l ||B_{kl}^\top \Pi_{T\Delta}(\nabla_{x_k}^{k\tau})||_\infty \tag{143}$$

$$\leq \frac{1}{2}\sum_{k \neq l^*} \eta_k m_k \big(\tau\ln\big(\frac{1}{x_{\min}}\big)+1\big) + \frac{1}{2}\eta_{l^*} m_{l^*}\tau\big(\frac{1}{x_{\min}}-1\big)\big(\tau\ln\big(\frac{1}{x_{\min}}\big)+1\big) \tag{144}$$

$$= \frac{1}{2}\big(\tau\ln\big(\frac{1}{x_{\min}}\big)+1\big)\Big[\eta_{l^*} m_{l^*}\tau\big(\frac{1}{x_{\min}}-1\big) + \sum_{k\neq l^*}\eta_k m_k\Big] \tag{145}$$

$$\leq \frac{1}{2}(\max_k \eta_k)\big(\tau\ln\big(\frac{1}{x_{\min}}\big)+1\big)\Big[\tau m_{l^*}\big(\frac{1}{x_{\min}}-1\big) + \sum_{k\neq l^*} m_k\Big] \tag{146}$$

$$\leq \frac{1}{2}(\max_k \eta_k)\big(\tau\ln\big(\frac{1}{x_{\min}}\big)+1\big)\Big[\tau m^*\big(\frac{1}{x_{\min}}-1\big) + n\bar{m}\Big] \tag{147}$$

where the second inequality follows from Lemma 19.

$$\square$$

**Corollary 2.** *If $\tau$ is set according to Lemma 9, then the infinity norm of the gradient is bounded as*

$$||\nabla_{\boldsymbol{x}}\mathcal{L}^\tau(\boldsymbol{x})||_\infty \leq \frac{1}{2}(\max_k \eta_k)\Big[\frac{\ln(m^*)}{\ln(1/p)}+2\Big]\Big[\frac{m^{*2}}{p\ln(1/p)}+n\bar{m}\Big] = \frac{1}{2}(\max_k \eta_k)\hat{L} \tag{148}$$

*where $m^* = \max_k m_k$ and $\hat{L}$ is defined implicitly for convenience in other derivations.*

*Proof.* Starting with Lemma 16 and applying Lemma 9 (i.e., $\tau = \ln(1/p)^{-1}$ and $x_{\min} = \frac{p}{m^*}$ where $m^* = \max_k m_k$), we find

$$||\nabla_{\boldsymbol{x}}\mathcal{L}^\tau(\boldsymbol{x})||_\infty \leq \frac{1}{2}(\max_k \eta_k)\big(\tau\ln\big(\frac{1}{x_{\min}}\big)+1\big)\Big[\tau m^*\big(\frac{1}{x_{\min}}-1\big)+n\bar{m}\Big] \tag{149}$$

$$= \frac{1}{2}(\max_k \eta_k)\Big[\frac{\ln(m^*/p)}{\ln(1/p)}+1\Big]\Big[\frac{m^*}{\ln(1/p)}\big(\frac{m^*}{p}-1\big)+n\bar{m}\Big] \tag{150}$$

$$\leq \frac{1}{2}(\max_k \eta_k)\Big[\frac{\ln(m^*)}{\ln(1/p)}+2\Big]\Big[\frac{m^{*2}}{p\ln(1/p)}+n\bar{m}\Big]. \tag{151}$$

As $p \to 0^+$, the norm of the gradient blows up because the gradient of Shannon entropy blows up for small probabilities. As $p \to 1$, the norm of the gradient blows up because we require infinite temperature $\tau$ to guarantee all QREs are nearly uniform; recall $\tau$ is the regularization coefficient on the entropy bonus terms which means our modified utilities blow up for large $\tau$. In practice, setting $p$ to $\mathcal{O}(1)$, e.g., $p = \frac{1}{10}$ is sufficient. $\square$

# D  Hessian of Loss

We will now derive the Hessian of our loss. This will be useful in establishing properties about global minima that enable the application of tailored minimization algorithms. Let $D_z[f(z)]$ denote the differential operator applied to (possibly multivalued) function $f$ with respect to $z$. For example, $D_{x_q}[H_{lk}^k] = D_{x_q}[x_q T_{qlk}^k] = T_{qlk}^k$ where $T_{qlk}^k$ is player $k$'s payoff tensor according to the three-way approximation between players $k$, $l$, and $q$ to the game at $\boldsymbol{x}$.

**Lemma 17.** *The Hessian of $\mathcal{L}^\tau(\boldsymbol{x})$ can be written*

$$\texttt{Hess}(\mathcal{L}^\tau) = 2\tilde{B}^\top \tilde{B} + T\Pi_{T\Delta}(\tilde{\nabla}^\tau) \tag{152}$$

*where $\tilde{B}_{kl} = \sqrt{\eta_k}B_{kl}$, $\Pi_{T\Delta}(\tilde{\nabla}^\tau) = [\eta_1\Pi_{T\Delta}(\nabla_{x_1}^{1\tau}),\ldots,\eta_n\Pi_{T\Delta}(\nabla_{x_n}^{n\tau})]$, and we augment $T$ (the 3-player tensor approximation to the game, $T_{lqk}^k$) so that $T_{lll}^l = \tau\texttt{diag3}(\frac{1}{x_l^2})$ and otherwise 0.*

595    *Proof.* Recall the gradient of our proposed loss:

$$\nabla_{x_l}\mathcal{L}(\boldsymbol{x}) = 2\sum_k \eta_k B_{kl}^\top \Pi_{T\Delta}(\nabla_{x_k}^{k\tau}) \tag{153}$$

596    where $B_{ll} = -\tau[I - \frac{1}{m_l}\mathbf{1}\mathbf{1}^\top]\texttt{diag}(\frac{1}{x_l})$ and $B_{kl} = [I - \frac{1}{m_k}\mathbf{1}\mathbf{1}^\top]H_{kl}^k$ for $k \neq l$.

597    Consider the following Jacobians, which will play an auxiliary role in our derivation of the Hessian:

$$D_l[B_{ll}] = \tau[I - \frac{1}{m_l}\mathbf{1}\mathbf{1}^\top]\texttt{diag3}(\frac{1}{x_l^2}) \tag{154}$$

$$D_q[B_{ll}] = \mathbf{0} \tag{155}$$

$$D_l[B_{kl}] = \mathbf{0} \tag{156}$$

$$D_q[B_{kl}] = [I - \frac{1}{m_k}\mathbf{1}\mathbf{1}^\top]T_{klq}^k \tag{157}$$

$$D_k[\Pi_{T\Delta}(\nabla_{x_k}^{k\tau})] = [I - \frac{1}{m_k}\mathbf{1}\mathbf{1}^\top]D_k[\nabla_{x_k}^{k\tau}] \tag{158}$$

$$= [I - \frac{1}{m_k}\mathbf{1}\mathbf{1}^\top]D_k[\nabla_{x_k}^k - \tau\ln(x_k)] \tag{159}$$

$$= [I - \frac{1}{m_k}\mathbf{1}\mathbf{1}^\top][-\tau\texttt{diag}(\frac{1}{x_k})] \tag{160}$$

$$= B_{kk} \tag{161}$$

$$D_l[\Pi_{T\Delta}(\nabla_{x_k}^{k\tau})] = [I - \frac{1}{m_k}\mathbf{1}\mathbf{1}^\top]D_l[\nabla_{x_k}^{k\tau}] \tag{162}$$

$$= [I - \frac{1}{m_k}\mathbf{1}\mathbf{1}^\top]D_l[\nabla_{x_k}^k - \tau\ln(x_k)] \tag{163}$$

$$= [I - \frac{1}{m_k}\mathbf{1}\mathbf{1}^\top][H_{kl}^k] \tag{164}$$

$$= B_{kl}. \tag{165}$$

598    We can derive the diagonal blocks of the Hessian as

$$D_{ll}[\mathcal{L}(\boldsymbol{x})] = D_l[\nabla_{x_l}\mathcal{L}(\boldsymbol{x})] \tag{166}$$

$$= 2D_l[\sum_k \eta_k B_{kl}^\top \Pi_{T\Delta}(\nabla_{x_k}^{k\tau})] \tag{167}$$

$$= 2\Big[\eta_l D_l\big[B_{ll}^\top \Pi_{T\Delta}(\nabla_{x_l}^{l\tau})\big] + \sum_{k\neq l} \eta_k D_l\big[B_{kl}^\top \Pi_{T\Delta}(\nabla_{x_k}^{k\tau})\big]\Big] \tag{168}$$

$$= 2\Big[\eta_l\big[D_l[B_{ll}]^\top \Pi_{T\Delta}(\nabla_{x_l}^{l\tau}) + B_{ll}^\top D_l[\Pi_{T\Delta}(\nabla_{x_l}^{l\tau})]\big] \tag{169}$$

$$+ \sum_{k\neq l} \eta_k\big[D_l[B_{kl}]^\top \Pi_{T\Delta}(\nabla_{x_k}^{k\tau}) + B_{kl}^\top D_l[\Pi_{T\Delta}(\nabla_{x_k}^{k\tau})]\big]\Big] \tag{170}$$

$$= 2\Big[\eta_l\big[\tau\texttt{diag3}(\frac{1}{x_l^2})[I - \frac{1}{m_l}\mathbf{1}\mathbf{1}^\top]\Pi_{T\Delta}(\nabla_{x_l}^{l\tau}) + B_{ll}^\top B_{ll}\big] + \sum_{k\neq l}\eta_k B_{kl}^\top B_{kl}\Big] \tag{171}$$

$$= 2\Big[\tau\eta_l\texttt{diag}([\frac{1}{x_l^2}]\odot\Pi_{T\Delta}(\nabla_{x_l}^{l\tau})) + \sum_k \eta_k B_{kl}^\top B_{kl}\Big] \tag{172}$$

and the off-diagonal blocks as

$$D_{lq}[\mathcal{L}(\boldsymbol{x})] = D_q[\nabla_{x_l}\mathcal{L}(\boldsymbol{x})] \tag{173}$$

$$= 2D_q[\sum_k \eta_k B_{kl}^\top \Pi_{T\Delta}(\nabla_{x_k}^{k\tau})] \tag{174}$$

$$= 2\Big[\eta_l D_q\big[B_{ll}^\top \Pi_{T\Delta}(\nabla_{x_l}^{l\tau})\big] + \sum_{k\neq l} \eta_k D_q\big[B_{kl}^\top \Pi_{T\Delta}(\nabla_{x_k}^{k\tau})\big]\Big] \tag{175}$$

$$= 2\Big[\eta_l \big[D_q[B_{ll}]^\top \Pi_{T\Delta}(\nabla_{x_l}^{l\tau}) + B_{ll}^\top D_q[\Pi_{T\Delta}(\nabla_{x_l}^{l\tau})]\big] \tag{176}$$

$$+ \sum_{k\neq l} \eta_k \big[D_q[B_{kl}]^\top \Pi_{T\Delta}(\nabla_{x_k}^{k\tau}) + B_{kl}^\top D_q[\Pi_{T\Delta}(\nabla_{x_k}^{k\tau})]\big]\Big] \tag{177}$$

$$= 2\Big[\eta_l B_{ll}^\top B_{lq} + \sum_{k\neq l} \eta_k \big[T_{lqk}^k[I - \frac{1}{m_k}\mathbf{1}\mathbf{1}^\top]\Pi_{T\Delta}(\nabla_{x_k}^{k\tau}) + B_{kl}^\top B_{kq}\big]\Big] \tag{178}$$

$$= 2\Big[\sum_k \eta_k B_{kl}^\top B_{kq} + \sum_{k\neq l} \eta_k T_{lqk}^k \Pi_{T\Delta}(\nabla_{x_k}^{k\tau})\Big]. \tag{179}$$

Therefore, the Hessian can be written concisely as

$$2\big[\tilde{B}^\top \tilde{B} + T\Pi_{T\Delta}(\tilde{\nabla}^\tau)\big] \tag{180}$$

where $\tilde{B}_{kl} = \sqrt{\eta_k} B_{kl}$, $\Pi_{T\Delta}(\tilde{\nabla}^\tau) = [\eta_1 \Pi_{T\Delta}(\nabla_{x_1}^{1\tau}), \ldots, \eta_n \Pi_{T\Delta}(\nabla_{x_n}^{n\tau})]$, and we augment $T$ (the 3-player tensor approximation to the game, $T_{lqk}^k$) so that $T_{lll}^l = \tau \texttt{diag3}(\frac{1}{x_l^2})$ and otherwise 0.

$\square$

# E   Regret Bounds

**Lemma 18.** *[Loss Regret to Exploitability Regret] Assume exploitability of a joint strategy $\boldsymbol{x}$ is upper bounded by $f(\mathcal{L}^\tau(\boldsymbol{x}))$ where $f$ is a concave function and $\mathcal{L}^\tau$ is a loss function. Let $\boldsymbol{x}_t$ be a joint strategy randomly drawn from the set of predictions made by an online learning algorithm $\mathcal{A}$ over $T$ steps. Then the expected exploitability of $\boldsymbol{x}_t$ is bounded by the average regret of $\mathcal{A}$:*

$$\mathbb{E}[\epsilon_t] \leq f(\frac{1}{T}\sum_t \mathcal{L}_t). \tag{181}$$

*Proof.*

$$\mathbb{E}[\epsilon_t] = \mathbb{E}[f(\mathcal{L}(\boldsymbol{x}_t))] \tag{182}$$

$$\leq f(\mathbb{E}[\mathcal{L}(\boldsymbol{x}_t)]) \tag{183}$$

$$= f(\frac{1}{T}\sum_t \mathcal{L}(\boldsymbol{x}_t)) \tag{184}$$

where the second inequality follows from Jensen's inequality. $\square$

**Theorem 1.** *[BLiN PAC Rate] Assume $\eta_k = \eta = 2/\hat{L}$ as defined in Lemma 2, $\tau = \frac{1}{\ln(1/p)}$ so that all equilibria place at least $\frac{p}{m^*}$ mass on each strategy, and a previously pulled arm is returned uniformly at random (i.e., $t \sim U(T)$). Then for any $w > 0$,*

$$\epsilon_t \leq w\Big[n\tau(W(1/e) + \frac{\bar{m}-2}{e}) + 4(1 + (4c^2)^{1/3})\sqrt{nm^*\hat{L}}\Big(\frac{\ln T}{T}\Big)^{\frac{1}{2(d_z+2)}}\Big] \tag{185}$$

*with probability $(1 - w^{-1})(1 - 2T^{-2})$ where $W$ is the Lambert function ($W(1/e) \approx 0.278$), $m^* = \max_k m_k$, and $c \leq \frac{1}{4}\frac{n\bar{m}}{\hat{L}}\Big(\frac{\ln(m^*)}{\ln(1/p)} + 2\Big)^2$ is an upper bound on the maximum sampled value from $\mathcal{L}^\tau$ (see Lemma 8).*

*Proof.* Assume $\eta_k = \eta = \frac{2}{\hat{L}}$ as defined in Lemma 2 so that $\mathcal{L}^\tau$ is 1-Lipschitz with respect to $|| \cdot ||_\infty$. Also assume a previously pulled arm is returned uniformly at random. Starting with Lemma 14 and applying Corollary 9, we find

$$\mathbb{E}[\epsilon_t] \leq n\tau(W(1/e) + \frac{\bar{m} - 2}{e}) + 2\sqrt{\frac{n \max_k m_k}{\min_k \eta_k}}\sqrt{\mathcal{L}^\tau(\boldsymbol{x})} \tag{186}$$

$$= \frac{n}{\ln(1/p)}(W(1/e) + \frac{\bar{m} - 2}{e}) + \sqrt{2nm^*\hat{L}}\sqrt{8(1 + (4c^2)^{1/3})^2 T^{\frac{-1}{(d_z+2)}} \ln T^{\frac{1}{(d_z+2)}}} \tag{187}$$

$$= \frac{n}{\ln(1/p)}(W(1/e) + \frac{\bar{m} - 2}{e}) + 4(1 + (4c^2)^{1/3})\sqrt{nm^*\hat{L}}\Big(\frac{\ln T}{T}\Big)^{\frac{1}{2(d_z+2)}} \tag{188}$$

with probability $1 - 2T^{-2}$ where $W$ is the Lambert function ($W(1/e) \approx 0.278$), $m^* = \max_k m_k$, and $c \leq \frac{1}{4}\frac{n\bar{m}}{\hat{L}}\Big(\frac{\ln(m^*)}{\ln(1/p)} + 2\Big)^2$ is an upper bound on the range of sampled values from $\mathcal{L}^\tau$ (see Lemma 8).

Recall $\hat{L} = \Big[\frac{\ln(m^*)}{\ln(1/p)} + 2\Big]\Big[\frac{m^{*2}}{p\ln(1/p)} + n\bar{m}\Big]$. Therefore,

$$c \leq \frac{1}{4}\frac{n\bar{m}}{\hat{L}}\Big(\frac{\ln(m^*)}{\ln(1/p)} + 2\Big)^2 \tag{189}$$

$$= \frac{1}{4}n\bar{m}\Big(\frac{\frac{\ln(m^*)}{\ln(1/p)} + 2}{\frac{m^{*2}}{p\ln(1/p)} + n\bar{m}}\Big). \tag{190}$$

Markov's inequality then allows us to bound the pointwise exploitability of any arm returned by the algorithm as

$$\epsilon_t \leq w\Big[\frac{n}{\ln(1/p)}(W(1/e) + \frac{\bar{m} - 2}{e}) + 4(1 + (4c^2)^{1/3})\sqrt{nm^*\hat{L}}\Big(\frac{\ln T}{T}\Big)^{\frac{1}{2(d_z+2)}}\Big] \tag{191}$$

with probability $(1 - w^{-1})(1 - 2T^{-2})$ for any $w > 0$. $\qquad\square$

# F  Complexity

## F.1  Polymatrix Games

Interestingly, at zero temperature (where QRE = Nash), $M$ is constant for a polymatrix game, so the rank of this matrix can be computed just once to extract information about all possible interior equilibria in the game. Furthermore, the Hessian is positive semi-definite over the entire joint strategy space, implying the loss function is convex (see Figure 1 (left) for empirical support). This indicates, by convex optimization theory, 1) all mixed Nash equilibria in polymatrix games form a convex set (i.e., they are connected) and 2) assuming mixed equilibria exist, they can be computed simply by stochastic gradient descent on $\mathcal{L}$. If $M$ is rank-$n\bar{m}$, then this interior equilibrium is unique.

**Complexity**  Approximation of Nash equilibria in polymatrix games is known to be PPAD-hard [13]. In contrast, if we restrict our class of polymatrix games to those with at least one interior Nash equilibrium, our analysis proves we can find an approximate Nash equilibrium in deterministic, polynomial time (Corollary 3). This follows directly from the fact that $\mathcal{L}$ is convex, our decision set $\mathcal{X} = \prod_i \mathcal{X}_i$ is convex, and convex optimization theory admits polynomial time approximation algorithms (e.g., gradient descent). We consider the assumption of the existence of an interior Nash equilibrium to be relatively mild[3], so this positive complexity result is surprising.

Also, note that the Hessian of the loss at Nash equilibria is encoded entirely by the polymatrix approximation at the equilibrium. Therefore, approximating the Hessian of $\mathcal{L}$ about the equilibrium (which amounts to observing near-equilibrium behavior [25]) allows one to recover this polymatrix approximation (up to constant offsets of the columns which equilibria are invariant to [27]).

---

[3]Marris et al. [27] shows 2-player, 2-action polymatrix games with interior Nash equilibria make up a non-trivial $1/4$ of the space of possible $2 \times 2$ games.

**Corollary 3** (Approximating Nash Equilibria of Polymatrix Games with Interior Equilibria). *Consider the class of polymatrix games with interior Nash equilibria. This class of games admits a fully polynomial time deterministic approximation scheme (FPTAS).*

*Proof.* Lemma 3 relates the approximation of Nash equilibria to the minimization of the loss function $\mathcal{L}(\boldsymbol{x})$. By Lemma 1, this loss function attains its minimum value of zero if and only if $\boldsymbol{x}$ is a Nash equilibrium. For polymatrix games, Hessian of this loss function is everywhere finite and positive definite (Lemma 17), therefore, this loss function is convex. The decision set for this minimization problem is the product space of simplices, therefore it is also convex. Given that we only consider polymatrix games with interior Nash equilibria, we know that our loss function attains a global minimum within this set. By convex optimization theory, this function can be approximately minimized in a polynomial number of steps by, for example, (projected) gradient descent. Gradient descent requires computing the gradient of the loss function at each step. From Lemma 15, we see that computing the gradient (at zero temperature) simply requires reading the polymatrix description of the game (i.e., each bi-matrix game $H_{kl}^{k}$ between players), which is clearly polynomial in the size of the input (the polymatrix description). The remaining computational steps of gradient descent (e.g., projection onto simplices) are polynomial as well. In conclusion, gradient descent approximates a Nash equilibrium in polynomial number of steps (logarithmic if strongly-convex), each of which costs polynomial time, therefore the entire scheme is polynomial. □

### F.2 Normal-Form Games

**Corollary 1.** *Consider the class of NFGs with at least one QRE($\tau$) whose local polymatrix approximation indicates it is isolated (i.e., $M$ from equation (12) is rank-$n\bar{m}$ implies Hess $\succ 0$ implies $d_z = n\bar{m}(\frac{2-2}{4}) = 0$). Then by Theorem 1, BLiN is a fully polynomial-time randomized approximation scheme (FPRAS) for QREs and is a PRAS for NEs of games in this class.*

*Proof.* If $\alpha = 0$, an $\epsilon$-QRE can be obtained with BLiN in a number of iterations that is polynomial in the game description length ($nm^n$). The same holds for an $\epsilon$-NE, however, the temperature must be exponentially small to achieve a given $\epsilon$; hence, we lose the *fully* qualifier. Specifically,

$$p \leq e^{-\frac{8n}{\epsilon}\left(W(1/e) + \frac{\bar{m}-2}{e}\right)}. \tag{192}$$

This, in turn, causes the Lipschitz constant $\hat{L}$ to grow exponentially large, leading to an exponential blow up in the number of iterations required for convergence. □

#### F.2.1 Concrete Example

The end of Section 6 stated a concrete result for a 20-player, 50-action game *assuming* we are given that the game as an interior Nash equilibrium. This result requires re-deriving a rate similar to Theorem 1, but for the unregularized game.

For example, revisiting Corollary 2 but for zero temperature, we find $\hat{L} = n\bar{m}$. Let $\eta = \frac{2}{\hat{L}}$ as before. Now, consulting Table 2, we find that samples from $\mathcal{L}$ are constrained to a range of size $c = \frac{1}{2}n\bar{m}\eta = 1$. Applying Corollary 9 to Lemma 3, we find:

$$\epsilon_t \leq w\left[2\sqrt{2}(1 + 4^{1/3})n\sqrt{\bar{m}}\left(\frac{\ln T}{T}\right)^{\frac{1}{4}}\right] \tag{193}$$

with probability $(1 - w^{-1})(1 - 2T^{-2}) = 0.95(1 - 2T^{-2})$. Plugging in $w = 20$, $n = 20$, and $m = 50$ and solving for $T$ numerically, we find that $T \leq 10^{28.7}$. For such large $T$, $0.95(1 - 2T^{-2}) \approx 0.95$. Again consulting Table 2, each call (arm pull) of BLiN costs $2nm$, implying a total query cost of $10^{32.0}$. In contrast, there exist $10^{35.2}$ scalar entries in the $nm^n$ payoff tensor, which is a factor larger by 1000.

## G  Helpful Lemmas and Propositions

**Proposition 2.** *The matrix $I - \frac{1}{m_k}\mathbf{1}\mathbf{1}^{\top}$ is a projection matrix and therefore idempotent. It is also symmetric, which implies it is its own square root.*

*Proof.*

$$[I - \frac{1}{m_k}\mathbf{1}\mathbf{1}^\top]^\top[I - \frac{1}{m_k}\mathbf{1}\mathbf{1}^\top] = I - \frac{2}{m_k}\mathbf{1}\mathbf{1}^\top + \frac{1}{m_k^2}\mathbf{1}(\mathbf{1}^\top\mathbf{1})\mathbf{1}^\top \tag{194}$$

$$= I - \frac{2}{m_k}\mathbf{1}\mathbf{1}^\top + \frac{1}{m_k}\mathbf{1}\mathbf{1}^\top \tag{195}$$

$$= [I - \frac{1}{m_k}\mathbf{1}\mathbf{1}^\top]. \tag{196}$$

$\square$

**Proposition 3.** *The matrix $I - \frac{1}{m_k}\mathbf{1}\mathbf{1}^\top$ is positive semi-definite.*

*Proof.* Let $z \in \mathbb{R}^{m_k}$. Then

$$z^\top[I - \frac{1}{m_k}\mathbf{1}\mathbf{1}^\top]z = ||z||_2^2 - \frac{1}{m_k}\langle z, \mathbf{1}\rangle^2 \tag{197}$$

$$\geq ||z||_2^2 - \frac{1}{m_k}\langle |z|, \mathbf{1}\rangle^2 \tag{198}$$

$$= ||z||_2^2 - \frac{1}{m_k}||z||_1^2 \tag{199}$$

$$\geq ||z||_2^2 - ||z||_2^2 = 0 \ \forall z \tag{200}$$

$$\implies [I - \frac{1}{m_k}\mathbf{1}\mathbf{1}^\top] \succeq 0. \tag{201}$$

$\square$

**Proposition 4.** *The matrix $I - \frac{1}{m_k}\mathbf{1}\mathbf{1}^\top$ has rank $m_k - 1$ and its 1-d nullspace lies along $\mathbf{1}_k$.*

*Proof.* Note that $rank(A+B) \leq rank(A) + rank(B)$ for matrices $A$ and $B$ of the same dimension. Let $A = I - \frac{1}{m_k}\mathbf{1}\mathbf{1}^\top$ and $B = \frac{1}{m_k}\mathbf{1}\mathbf{1}^\top$ and apply $rank(A) \geq rank(A+B) - rank(B)$:

$$rank(I - \frac{1}{m_k}\mathbf{1}\mathbf{1}^\top) \geq rank(I) - rank(\frac{1}{m_k}\mathbf{1}\mathbf{1}^\top) = m_k - 1. \tag{202}$$

We can confirm the nullspace by inspection:

$$[I - \frac{1}{m_k}\mathbf{1}\mathbf{1}^\top]\mathbf{1} = \mathbf{1} - \frac{m_k}{m_k}\mathbf{1} = 0. \tag{203}$$

$\square$

**Lemma 19.** *The product $A[I_m - \frac{1}{m}\mathbf{1}_m\mathbf{1}_m^\top]^p B$ for any $p > 0$ has entries whose absolute value is bounded by $\frac{m}{4}(A_{\max} - A_{\min})(B_{\max} - B_{\min})$ where $A_{\min}, A_{\max}, B_{\min}, B_{\max}$ represent the minima and maxima of the matrices respectively.*

*Proof.* The matrix $[I - \frac{1}{m}\mathbf{1}\mathbf{1}^\top]$ is idempotent so we can rewrite the product for any $p$ as

$$A[I - \frac{1}{m}\mathbf{1}\mathbf{1}^\top][I - \frac{1}{m}\mathbf{1}\mathbf{1}^\top]B. \tag{204}$$

The matrix $[I - \frac{1}{m}\mathbf{1}\mathbf{1}^\top]$ has the property that it removes the mean from every row of a matrix when right multiplied against it, i.e., $A[I - \frac{1}{m}\mathbf{1}\mathbf{1}^\top]$ removes the means from the rows of $A$. Similarly, left multiplying it removes the means from the column. Let $\tilde{A}$ and $\tilde{B}$ represent these mean-centered results respectively. The absolute value of the $ij$th entry in the resulting product can then be recognized as

$$|\sum_k \tilde{A}_{ik}\tilde{B}_{kj}| = |\sum_k (A_{ik} - \frac{1}{m}\sum_{k'} A_{ik'})(B_{kj} - \frac{1}{m}\sum_{k'} B_{k'j})| \tag{205}$$

$$= |m \cdot Corr(A_{i,\cdot}, B_{\cdot,j}) \cdot \sigma_{A_{i,\cdot}}\sigma_{B_{\cdot,j}}| \tag{206}$$

$$\leq m\sigma_{A_{i,\cdot}}\sigma_{B_{\cdot,j}}. \tag{207}$$

The variance of a bounded random variable $X$ is upper bounded by $Var[X] \leq \frac{1}{4}(\max_X - \min_X)^2$. Hence its standard deviation is bounded by $Std[X] \leq \frac{1}{2}(\max_X - \min_X)$. Plugging these bounds for $A$ and $B$ into equation (207) completes the claim. $\square$

 # H Maps from Hypercube to Simplex Product

In this section, we derive properties of a map $s$ from the unit-hypercube to the simplex product. This map is necessary to to adapt our proposed loss $\mathcal{L}^\tau$ to the commonly assumed setting in the $\mathcal{X}$-armed bandit literature [8]. We derive relevant properties of two such maps: the `softmax` and a mapping that interprets dimensions of the hypercube as angles on a unit-sphere that is then $\ell_1$-normalized.

**Lemma 20.** *Let* $f(x) = -\mathcal{L}(s(x))$. *Then* $||\nabla f(x)||_\infty \leq ||J(s(x))^\top||_\infty ||\nabla \mathcal{L}(s(x))||_\infty$.

*Proof.*

$$||\nabla f(x)||_\infty = ||J(s(x))^\top \nabla \mathcal{L}(s(x))||_\infty \leq ||J(s(x))^\top||_\infty ||\nabla \mathcal{L}(s(x))||_\infty. \tag{208}$$

$\square$

**Lemma 21.** *The $\infty$-norm of the Jacobian-transpose of a transformation $s(x)$ applied elementwise to a product space is bounded by the $\infty$-norm of the Jacobian-transpose of a single transformation from that product space, i.e.,* $||J(s(\boldsymbol{x}))^\top||_\infty \leq \max_{x_i \in \mathcal{X}_i} ||J(s(x_i))^\top||_\infty$ *for any $i$.*

*Proof.* Let $\boldsymbol{x} \in \mathcal{X} = \prod_{i=1}^n \mathcal{X}_i$, $\mathcal{Z} = \prod_{i=1}^n \mathcal{Z}_i$ and $S : \mathcal{X} \to \mathcal{Z} = [s(x_1); \cdots ; s(x_n)]^\top$ where ; denotes column-wise stacking, $x_i \in \mathcal{X}_i$. Also, $\mathcal{X}_i = \mathcal{X}_j$ and $\mathcal{Z}_i = \mathcal{Z}_j$ for all $i$ and $j$. Then the Jacobian of $S(\boldsymbol{x})$ is

$$J(S(\boldsymbol{x}))^\top = \begin{bmatrix} J(s(x_1))^\top & 0\ldots & & 0 \\ 0 & J(s(x_2))^\top \ldots & & 0 \\ 0 & 0 \ddots & & 0 \\ 0 & 0 \ldots & & J(s(x_n))^\top \end{bmatrix}. \tag{209}$$

The $\infty$-norm of this matrix is the max 1-norm of any row. This matrix is diagonal, therefore, the $\infty$-norm of each elementwise Jacobian-transpose represents the max 1-norm of the rows spanned by its block. Given that the domains, ranges, and transformations $s$ for all blocks are the same, their $\infty$-norms are also the same. The max $\infty$ over the blocks is then equal to the $\infty$-norm of any individual $J(s(x_i))^\top$. $\square$

## H.1 Hessian of Bandit Reward Function

**Lemma 22.** *Let $s(x)$ be a function that maps the unit hypercube to the simplex product (mixed strategy space). Then the objective function $f(x) = -\mathcal{L}(s(x))$. The Hessian of $-f(x)$ at an optimum $x^*$ in direction $\Delta$ is $\Delta x^\top [Ds(x)^\top H_\mathcal{L}(x) Ds(x)]\big|_{x^*} \Delta x$ where $H_\mathcal{L}$ is the Hessian of $\mathcal{L}$ and $Ds(x)$ is the Jacobian of $s(x)$.*

*Proof.*

$$(D^2(\mathcal{L} \circ s)(x^*))(\Delta x, \Delta x) = \Delta x^\top \Big[ \sum_i \overbrace{\partial_i \mathcal{L}(s(x))}^{=0 \text{ at } x=x^*} D^2 h_i(x) \Big]\Big|_{x^*} \Delta x + \Delta x^\top [Ds(x)^\top H_\mathcal{L}(x) Ds(x)]\big|_{x^*} \Delta x \tag{210}$$

$$= \Delta x^\top [Ds(x)^\top H_\mathcal{L}(x) Ds(x)]\big|_{x^*} \Delta x. \tag{211}$$

$\square$

**Lemma 23.** *Let $s(x) : \mathcal{X} \to \prod_k \Delta^{m_k-1}$ be an injective function, i.e., $x \neq y \implies s(x) \neq s(y)$. Also let $J = J(s(x))$ be the Jacobian of $s$ with respect to $x$ and $\Delta x$ be a nonzero vector in the tangent space of $\mathcal{X}$. Then*

$$J\Delta x \neq \mathbf{0}. \tag{212}$$

*Proof.* Recall that the $ij$th entry of the Jacobian represents $\frac{\partial s_i}{\partial x_j}$ so that the $i$th entry of $J\Delta x$ is

$$[J\Delta x]_i = \sum_j \frac{\partial s_i}{\partial x_j} \Delta x_j = ds_i. \tag{213}$$

Assume $J\Delta x = \mathbf{0}$. This would imply a change in $x \in \mathcal{X}$ results in no change in $s$ ($ds = \mathbf{0}$), contradicting the fact that $s$ is injective. Therefore, we must conclude the claim that $J\Delta x \neq \mathbf{0}$. $\square$

**Lemma 24.** *Let $J$ be the Jacobian of the softmax operator. Then $||J||_\infty \leq 2$ and $||J^\top||_\infty \leq 2$.*

*Proof.* Let $S_i$ represent the $i$th entry of $S = \texttt{softmax}(z)$ for any $z \in \mathbb{R}^m$. Then the 1-norm of row $i$ is upper bounded as

$$D_j S_i = S_i(\delta_{ij} - S_j) \tag{214}$$

$$\implies \sum_j |D_j S_i| = \sum_j |S_i(\delta_{ij} - S_j)| \tag{215}$$

$$\leq \sum_j |\delta_{ij} S_i| + |S_i S_j| \tag{216}$$

$$= S_i + \sum_j S_i S_j \tag{217}$$

$$= S_i + S_i \sum_j S_j \tag{218}$$

$$= 2S_i \tag{219}$$

$$\leq 2 \; \forall i. \tag{220}$$

Also, the 1-norm of row $j$ is upper bounded similarly as

$$\tag{221}$$

$$\sum_i |D_j S_i| = \sum_i |S_i(\delta_{ij} - S_j)| \tag{222}$$

$$\leq \sum_i |\delta_{ij} S_i| + |S_i S_j| \tag{223}$$

$$= S_j + \sum_i S_i S_j \tag{224}$$

$$= S_i + S_j \sum_i S_i \tag{225}$$

$$= 2S_j \tag{226}$$

$$\leq 2 \; \forall j. \tag{227}$$

The $\infty$-norm of a matrix is the maximum 1-norm of any row. Therefore, $||J||_\infty$ and $||J^\top||_\infty$ are both upper bounded by 2. $\square$

**Lemma 25.** *Let $J = J(s(x))$ be the Jacobian of any composition of transformations $s = s_t \circ \ldots s_1$ where $s_t(z) = [z_i / \sum_j z_j]_i$. Then $J\Delta x$ lies in the tangent space of the simplex.*

*Proof.* We aim to show $\mathbf{1}^\top J\Delta x = \mathbf{0}$ for any $\Delta x$ and $x$. By chain rule, the Jacobian of $s$ is $J = J(s) = \prod_{t'=t}^{t'=1} J(s_t')$. Therefore, $\mathbf{1}^\top J\Delta x = \mathbf{1}^\top (\prod_{t'=t}^{t'=1} J(s_t'))\Delta x$. Consider the first product:

$$\mathbf{1}^\top J(s_t) = \mathbf{0} \tag{228}$$

by Lemma 27. Therefore $\mathbf{1}^\top J\Delta x = \mathbf{1}^\top J(s_t)(\prod_{t'=t-1}^{t'=1} J(s_t'))\Delta x = \mathbf{0}^\top (\prod_{t'=t-1}^{t'=1} J(s_t'))\Delta x = 0$. This implies $J\Delta x$ is orthogonal to $\mathbf{1}$ for any $x \in \mathcal{X}$ and $\Delta x$, therefore $J\Delta x$ lies in the tangent space of the simplex for any $x \in \mathcal{X}$ and $\Delta x$. $\square$

For spherical coordinates, $s(x) = n(l(c(x)))$ where $c(x) = \pi/2x$, $l(\psi)$ maps angles to the unit sphere, and $n(z) = [z_i / \sum_j z_j]_i$.

**Definition 1.** *Define $l(\psi)$ as the transformation to the unit-sphere using spherical coordinates:*

$$l_1(\psi) = \cos(\psi_1) \tag{229}$$

$$l_2(\psi) = \sin(\psi_1)\cos(\psi_2) \tag{230}$$

$$l_3(\psi) = \sin(\psi_1)\sin(\psi_2)\cos(\psi_3) \tag{231}$$

$$\vdots = \vdots \tag{232}$$

$$l_{m-1}(\psi) = \sin(\psi_1)\sin(\psi_2)\ldots\cos(\psi_{m-1}) \tag{233}$$

$$l_m(\psi) = \sin(\psi_1)\sin(\psi_2)\ldots\sin(\psi_{m-1}). \tag{234}$$

**Lemma 26.** *Let $J$ be the Jacobian of the transformation to the unit-sphere using spherical coordinates, i.e. $z = l(\psi)$ where $||l||^2 = 1$ and $\psi_i \in [0, \frac{\pi}{2}]$ represents an angle for each $i$. Then $||J||_F \leq \sqrt{m}$.*

*Proof.* The Jacobian of the transformation is

$$J(l) = \begin{bmatrix} -\sin(\psi_1) & 0 & \cdots & 0 \\ \cos(\psi_1)\cos(\psi_2) & -\sin(\psi_1)\sin(\psi_2) & \cdots & 0 \\ \vdots & \vdots & \ddots & \vdots \\ \cos(\psi_1)\sin(\psi_2)\ldots\cos(\psi_{m-1}) & \cdots & \cdots & -\sin(\psi_1)\ldots\sin(\psi_{m-2})\sin(\psi_{m-1}) \\ \cos(\psi_1)\sin(\psi_2)\ldots\sin(\psi_{m-1}) & \cdots & \cdots & \sin(\psi_1)\ldots\sin(\psi_{m-2})\cos(\psi_{m-1}) \end{bmatrix} \tag{235}$$

and it square is

$$J(l) = \begin{bmatrix} t_1 & 0 & \cdots & 0 \\ \cos(\psi_1)^2\cos(\psi_2)^2 & \sin(\psi_1)^2 t_2 & \cdots & 0 \\ \vdots & \vdots & \ddots & \vdots \\ \cos(\psi_1)^2\sin(\psi_2)^2\ldots\cos(\psi_{m-1})^2 & \cdots & \cdots & \sin(\psi_1)^2\ldots\sin(\psi_{m-2})^2 t_{m-1} \\ \cos(\psi_1)^2\sin(\psi_2)^2\ldots\sin(\psi_{m-1})^2 & \cdots & \cdots & \sin(\psi_1)^2\ldots\sin(\psi_{m-2})^2 t_m \end{bmatrix} \tag{236}$$

where

$$\delta_{im} = 1 \text{ if } i = m, 0 \text{ else} \tag{237}$$

$$t_i = \delta_{im}\cos^2(\psi_{i-1}) + (1 - \delta_{im})\sin^2(\psi_i) \leq 1. \tag{238}$$

To compute the Frobenius norm, we will need the sum of the squares of all entries. We will consider the sum of each row individually using the following auxiliary variable $R_{i,k\leq i}$ where $\sum_j J_{ij}^2 = R_{i,1}$ and apply a recursive inequality.

$$\tag{239}$$

$$R_{i,k\leq i} = \sum_{k'=k}^{i-1}\cos^2(\psi_{k'})\Big[\prod_{l=k,l\neq k'}^{i-1}\sin^2(\psi_l)\Big]\cos^2(\psi_i) + t_i\prod_{l=k}^{i-1}\sin^2(\psi_l) \tag{240}$$

$$= \cos^2(\psi_k)\underbrace{\Big[\prod_{l=k+1}^{i-1}\sin^2(\psi_l)\Big]\cos^2(\psi_i)}_{\leq 1} \tag{241}$$

$$+ \sin^2(\psi_k)\sum_{k'=k+1}^{i-1}\cos^2(\psi_{k'})\Big[\prod_{l=k+1,l\neq k'}^{i-1}\sin^2(\psi_l)\Big]\cos^2(\psi_i) \tag{242}$$

$$+ \sin^2(\psi_k)t_i\prod_{l=k+1}^{i-1}\sin^2(\psi_l) \tag{243}$$

$$\leq \cos^2(\psi_k) \tag{244}$$

$$+ \sin^2(\psi_k)\Big(\sum_{k'=k+1}^{i-1}\cos^2(\psi_{k'})\Big[\prod_{l=k+1,l\neq k'}^{i-1}\sin^2(\psi_l)\Big]\cos^2(\psi_i) + t_i\prod_{l=k+1}^{i-1}\sin^2(\psi_l)\Big) \tag{245}$$

$$= \cos^2(\psi_k) + \sin^2(\psi_k)R_{i,k+1}. \tag{246}$$

763  Note then that $R_{i,k+1} \le 1 \implies R_{i,k} \le 1$. We know $R_{i,i} = t_i \le 1$, therefore, $R_{i,1} \le 1$ by applying
764  the inequality recursively. Finally, $\sum_j J_{ij}^2 = R_{i,1} \le 1$ implies the claim $||J||_F^2 = \sum_i R_{i,1} \le m$.   $\square$

765  **Lemma 27.** *Let $J$ be the Jacobian of $n(z) = z/Z$ where $Z = \sum_k z_k$. Then $\mathbf{1}^\top J = \mathbf{0}^\top$.*

766  *Proof.* The $ij$th entry of the Jacobian of $n(z)$ is

$$J(n)_{ij} = \frac{1}{Z^2}(-z_i + \delta_{ij}Z). \tag{247}$$

767  Therefore $[\mathbf{1}^\top J]_j = \sum_i J(n)_{ij} = \frac{1}{Z^2}(-Z + Z) = 0$ where $z$ is a point on the unit-sphere in the
768  positive orthant.   $\square$

# I    A2: Bounded Diameters and Well-shaped Cells

770  We assume the feasible set is a unit-hypercube of dimensionality $d$ where cells are evenly split along
771  the longest edge to give $b$ new partitions and $x_{h,i}$ represents the center of each cell.

772  There exists a decreasing sequence $w(h) > 0$, such that for any depth $h \ge 0$ and for any cell $\mathcal{X}_{h,i}$ of
773  depth $h$, we have $\sup_{x \in \mathcal{X}_{h,i}} \ell(x_{h,i}, x) \le w(h)$. Moreover, there exists $\nu > 0$ such that for any depth
774  $h \ge 0$, any cell $\mathcal{X}_{h,i}$ contains an $\ell$-ball of radius $\nu w(h)$ centered at $x_{h,i}$.

| $\ell(x,y)$ | $c$ | $\gamma$ | $\nu$ |
|---|---|---|---|
| $\ell(x,y) = \|x-y\|_2^\alpha$ | $d^{\alpha/2}\left(\frac{b}{2}\right)^\alpha$ | $b^{-\alpha/d}$ | $d^{-\alpha/2}b^{-2\alpha}$ |
| $\ell(x,y) = \|x-y\|_\infty^\alpha$ | $\left(\frac{b}{2}\right)^\alpha$ | $b^{-\alpha/d}$ | $b^{-2\alpha}$ |

Table 3: Bounding Constants: $\sup_{x \in \mathcal{X}_{h,i}} \ell(x_{h,i}, x) \le w(h) = c\gamma^h$.

## I.1    $L_2$-Norm

776  **Lemma 28** ($L_2$-Norm Bounding Ball). *Let $\ell(x,y) = \|x-y\|_2^\alpha$. Then $\sup_{x \in \mathcal{X}_{h,i}} \ell(x_{h,i}, x) \le$*
777  *$w_2(h) = c\gamma^h$ where $c = \left(\frac{db^2}{4}\right)^{\alpha/2}$ and $\gamma = b^{-\alpha/d}$.*

*Proof.*

$$w(0) = \Big[ \sum_{i=1}^{d} (1/2)^2 \Big]^{\alpha/2} = \Big( \frac{d}{4} \Big)^{\alpha/2} \tag{248}$$

$$w(1) = \Big[ (1/b \cdot 1/2)^2 + \sum_{i=2}^{d} (1/2)^2 \Big]^{\alpha/2} = [(1/b^2)(1/4) + (d-1)(1/4)]^{\alpha/2} \tag{249}$$

$$= \Big( \frac{d-1+1/b^2}{4} \Big)^{\alpha/2} \tag{250}$$

$$w(d) = \Big[ \sum_{i=1}^{d} (1/b \cdot 1/2)^2 \Big]^{\alpha/2} = \Big( \frac{d}{4 \cdot b^2} \Big)^{\alpha/2} \tag{251}$$

$$w(h) = \Big[ r(1/b)^{2(q+1)}(1/2)^2 + \sum_{i=r}^{d} (1/b)^{2q}(1/2)^2 \Big]^{\alpha/2} \tag{252}$$

$$= \Big[ (1/b)^{2q}(1/2)^2 \big( r(1/b)^2 + (d-r) \big) \Big]^{\alpha/2} \tag{253}$$

$$= \Big[ (1/b^2)^q (1/4) \big( d - r(1 - \frac{1}{b^2}) \big) \Big]^{\alpha/2} \tag{254}$$

$$\leq \Big[ (1/b^2)^q (1/4) d \Big]^{\alpha/2} \tag{255}$$

$$\leq \Big[ (1/b^2)^{h/d-1} (1/4) d \Big]^{\alpha/2} \tag{256}$$

$$= \Big[ (1/b^2)^{h/d} (b^2/4) d \Big]^{\alpha/2} \tag{257}$$

$$= \Big( \frac{db^2}{4} \Big)^{\alpha/2} (1/b)^{\frac{\alpha}{d} h} \tag{258}$$

$$= c\gamma^h \tag{259}$$

where $q, r = divmod(h, d) \implies q \geq h/d - 1$, $c = \big( \frac{db^2}{4} \big)^{\alpha/2}$, and $\gamma = (1/b)^{\alpha/d} = b^{-\alpha/d}$. $\qquad\square$

**Lemma 29** ($L_2$-Norm Inner Ball). *Let $\ell(x,y) = ||x - y||_2^{\alpha}$. Any cell $\mathcal{X}_{h,i}$ contains an $\ell$-ball of radius $\nu w_2(h)$ where $\nu = (db^4)^{-\alpha/2}$.*

*Proof.* Any cell $\mathcal{X}_{h,i}$ contains an $\ell$-ball of radius equal to its shortest axis:

$$r_{\min} = \Big[ (1/4)(1/b^2)^{\lceil h/d \rceil} \Big]^{\alpha/2} \tag{260}$$

$$\geq \Big[ (1/4)(1/b^2)^{h/d+1} \Big]^{\alpha/2} \tag{261}$$

$$= \Big( \frac{1}{b^2 \cdot 4} \Big)^{\alpha/2} (1/b)^{\frac{\alpha}{d} h} \tag{262}$$

$$= w(h) \cdot \Big( \frac{1}{db^4} \Big)^{\alpha/2}. \tag{263}$$

$\qquad\square$

## I.2 $L_\infty$-Norm

**Lemma 30** ($L_\infty$-Norm Bounding Ball). *Let $\ell(x,y) = ||x - y||_\infty^{\alpha}$. Then $\sup_{x \in \mathcal{X}_{h,i}} \ell(x_{h,i}, x) \leq w_\infty(h) = c\gamma^h$ where $c = \big( \frac{b}{2} \big)^{\alpha}$ and $\gamma = b^{-\alpha/d}$.*

*Proof.* Any cell $\mathcal{X}_{h,i}$ is contained by an $\ell$-ball of radius equal to its longest axis:

$$r_{\max} = \Big[ (1/4)(1/b^2)^{\lfloor h/d \rfloor} \Big]^{\alpha/2} \tag{264}$$

$$\leq \Big[ (1/4)(1/b^2)^{h/d-1} \Big]^{\alpha/2} \tag{265}$$

$$= \Big( \frac{b^2}{4} \Big)^{\alpha/2} (1/b)^{\frac{\alpha}{d} h} \tag{266}$$

$$= c\gamma^h \tag{267}$$

787    where $c = \left(\frac{b^2}{4}\right)^{\alpha/2}$, and $\gamma = (1/b)^{\alpha/d} = b^{-\alpha/d}$.      $\square$

788 **Lemma 31** ($L_\infty$-Norm Inner Ball). *Let $\ell(x, y) = ||x - y||_\infty^\alpha$. Any cell $\mathcal{X}_{h,i}$ contains an $\ell$-ball of*
789 *radius $\nu w_\infty(h)$ where $\nu = b^{-2\alpha}$.*

790 *Proof.* Any cell $\mathcal{X}_{h,i}$ contains an $\ell$-ball of radius equal to its shortest axis:

$$r_{\min} = \left[ (1/4)(1/b^2)^{\lceil h/d \rceil} \right]^{\alpha/2} \tag{268}$$

$$\geq \left[ (1/4)(1/b^2)^{h/d+1} \right]^{\alpha/2} \tag{269}$$

$$= \left( \frac{1}{b^2 \cdot 4} \right)^{\alpha/2} (1/b)^{\frac{\alpha}{d}h} \tag{270}$$

$$= w(h) \cdot \left( \frac{1}{b^4} \right)^{\alpha/2}. \tag{271}$$

791      $\square$

## I.3   Near Optimality Dimension

793 This is written in terms of a maximizing $f$.

794 **Assumption 1.** *Locally around each interior $x^*$, $-f(x)$ is lower bounded by $-f(x^*) + \sigma_-||x -$*
795 *$x^*||^{\alpha_{hi}}$ and upper bounded by $-f(x^*) + \ell(x, x^*)$ where $\ell(x, x^*) = \sigma_+||x - x^*||^{\alpha_{lo}}$ with $\alpha_{lo} \leq \alpha_{hi}$*
796 *and $\sigma_- \leq \sigma_+$ if $\alpha_{lo} = \alpha_{hi}$. In other words, for all $f(x) \geq f(x^*) - \eta$:*

$$f(x^*) - f(x) \leq \sigma_+||x - x^*||^{\alpha_{lo}} \tag{272}$$
$$f(x^*) - f(x) \geq \sigma_-||x - x^*||^{\alpha_{hi}} \tag{273}$$

797 *where we have left the precise norm unspecified for generality.*

798 **Definition 2.** $\mathcal{X}_\epsilon \overset{\text{def}}{=} \{x \in \mathcal{X} | f(x) \geq f(x^*) - \epsilon\}$

799 **Definition 3.** $\mathcal{X}_\epsilon^{lower} \overset{\text{def}}{=} \{x \in \mathcal{X} | f(x^*) - \sigma_-||x - x^*||^{\alpha_{hi}} \geq f(x^*) - \epsilon\}$

800 **Corollary 4.** $\mathcal{X}_\epsilon \subseteq \mathcal{X}_\epsilon^{lower}$.

801 *Proof.* By Assumption 1, $f(x^*) - \sigma_-||x - x^*||^{\alpha_{hi}} \geq f(x)$. Therefore, any $x \in \mathcal{X}$ that satisfies the
802 requirement for an element of $\mathcal{X}_\epsilon$, $f(x) \geq f(x^*) - \epsilon$, will also satisfy the requirement for an element
803 of $\mathcal{X}_\epsilon^{lower}$.      $\square$

804 **Definition 4.** *The $\psi$-near optimality dimension is the smallest $d' > 0$ such that there exists $C > 0$*
805 *such that for any $\epsilon > 0$, the maximum number of disjoint $\ell$-balls of radius $\psi\epsilon$ and center in $\mathcal{X}_\epsilon$ is less*
806 *than $C\epsilon^{-d'}$.*

807 **Theorem 2.** *The $\psi$-near optimality dimension of $f : x \in [0,1]^d \to [-1, 1]$ under $\ell$ is $d' =$*
808 *$d(\frac{\alpha_{hi} - \alpha_{lo}}{\alpha_{lo}\alpha_{hi}})$ with constant*

$$C = \max \left\{ 1, S_d^{-1} \left( r_\eta^{\frac{\alpha_{hi}}{\alpha_{lo}}} \sigma_-^{\left( \frac{\alpha_{hi} - \alpha_{lo}}{\alpha_{lo}\alpha_{hi}} \right)} \right)^{-d} \right\} \left( \frac{\sigma_+}{\psi \sigma_-^{\alpha_{lo}/\alpha_{hi}}} \right)^{d/\alpha_{lo}}. \tag{274}$$

809 *Proof.* First, let us define $r_\eta = \left( \frac{\eta}{\sigma_-} \right)^{1/\alpha_{hi}}$ as in equation (285) which implies $\eta = \sigma_- r_\eta^{\alpha_{hi}}$. Then
810 apply Lemmas 32 ($N_{\epsilon \leq \eta} \leq C_{\epsilon \leq \eta} \epsilon^{-d'}$) and 34 ($N_{\epsilon \geq \eta} \leq C_{\epsilon \geq \eta}$) which bound the number of $\ell$-balls
811 required to pack $\mathcal{X}_\epsilon$ when $\epsilon$ is less than and greater than $\eta$ respectively:

$$C_{\epsilon \leq \eta} = \left( \frac{\sigma_+}{\psi \sigma_-^{\alpha_{lo}/\alpha_{hi}}} \right)^{d/\alpha_{lo}} \tag{275}$$

$$d' = d\left( \frac{\alpha_{hi} - \alpha_{lo}}{\alpha_{lo}\alpha_{hi}} \right) \tag{276}$$

and

$$C_{\epsilon \geq \eta} = S_d^{-1} \left( \frac{\sigma_+}{\psi \eta} \right)^{d/\alpha_{lo}} \tag{277}$$

$$= S_d^{-1} \eta^{-d/\alpha_{lo}} \sigma_-^{d/\alpha_{hi}} \left( \frac{\sigma_+}{\psi \sigma_-^{\alpha_{lo}/\alpha_{hi}}} \right)^{d/\alpha_{lo}} \tag{278}$$

$$= S_d^{-1} \eta^{-d/\alpha_{lo}} \sigma_-^{d/\alpha_{hi}} C_{\epsilon \leq \eta} \tag{279}$$

$$= S_d^{-1} r_\eta^{-d\alpha_{hi}/\alpha_{lo}} \sigma_-^{-d/\alpha_{lo}} \sigma_-^{d/\alpha_{hi}} C_{\epsilon \leq \eta} \tag{280}$$

$$= S_d^{-1} r_\eta^{-d\frac{\alpha_{hi}}{\alpha_{lo}}} \sigma_-^{-d\left( \frac{\alpha_{hi}-\alpha_{lo}}{\alpha_{lo}\alpha_{hi}} \right)} C_{\epsilon \leq \eta} \tag{281}$$

$$= S_d^{-1} \left( r_\eta^{\frac{\alpha_{hi}}{\alpha_{lo}}} \sigma_-^{\left( \frac{\alpha_{hi}-\alpha_{lo}}{\alpha_{lo}\alpha_{hi}} \right)} \right)^{-d} C_{\epsilon \leq \eta} \tag{282}$$

where $S_d$ is the volume constant for a $d$-sphere under the given norm. $S_d^{-1}$ has been upper bounded for the 2-norm in Lemma 33. For the $\infty$-norm, $S_d^{-1} = 2^{-d}$. We have written $C_{\epsilon \geq \eta}$ in terms of $C_{\epsilon \leq \eta}$ to clarify which is larger.

Therefore,

$$C = \max \left\{ 1, S_d^{-1} \left( r_\eta^{\frac{\alpha_{hi}}{\alpha_{lo}}} \sigma_-^{\left( \frac{\alpha_{hi}-\alpha_{lo}}{\alpha_{lo}\alpha_{hi}} \right)} \right)^{-d} \right\} C_{\epsilon \leq \eta} \tag{283}$$

$$= \max \left\{ 1, S_d^{-1} \left( r_\eta^{\frac{\alpha_{hi}}{\alpha_{lo}}} \sigma_-^{\left( \frac{\alpha_{hi}-\alpha_{lo}}{\alpha_{lo}\alpha_{hi}} \right)} \right)^{-d} \right\} \left( \frac{\sigma_+}{\psi \sigma_-^{\alpha_{lo}/\alpha_{hi}}} \right)^{d/\alpha_{lo}}. \tag{284}$$

Intuitively, if the radius for which the polynomial bounds hold ($r_\eta$) is large and the minimum curvature constant $\sigma_-$ is also large, then the bound $C_{\epsilon \leq \eta}$ holds for large deviations from optimality $\eta$. The number of $\eta$-radius $\ell$-balls required to cover the remaining space, $C_{\epsilon \geq \eta}$, will be comparatively small. $\qquad \square$

**Corollary 5** (Zooming Dimension). *The zooming dimension of $f : x \in [0,1]^d \to [-1,1]$ under $\ell(x,y) = ||x-y||_\infty$ is $d_z = d(\frac{\alpha_{hi}-\alpha_{lo}}{\alpha_{lo}\alpha_{hi}})$.*

*Proof.* Mapping the definition of zooming dimension onto $\psi$-near optimality, we find $\psi \epsilon = r/2$ and $\epsilon = 16r$. Then we can infer $\psi = 1/32$. This result only effects the constant $C_z$, not the zooming dimension.

If $\epsilon = 8(1 + \sqrt{c_1/c_2}) r_m$, then $\psi = \frac{1}{16(1+\sqrt{c_1/c_2})}$. $\qquad \square$

**Lemma 32** ($N_{\epsilon \leq \eta} \leq C_{\epsilon \leq \eta} \epsilon^{-d'}$). *The number of disjoint $\ell$-balls that can pack into a set $\mathcal{X}_{\epsilon \leq \eta}$, $N_{\epsilon \leq \eta}$, is upper bounded by $C_{\epsilon \leq \eta} \epsilon^{-d'}$ where $C_{\epsilon \leq \eta} = \left( \frac{\sigma_+}{\psi \sigma_-^{\alpha_{lo}/\alpha_{hi}}} \right)^{d/\alpha_{lo}}$ and $d' = d(\frac{\alpha_{hi}-\alpha_{lo}}{\alpha_{lo}\alpha_{hi}})$ and $S_d$ is the volume constant for a $d$-sphere under the given norm $|| \cdot ||$.*

*Proof.* The number of disjoint $\ell$-balls of radius $\psi \epsilon$ and center in $\mathcal{X}_{\epsilon \leq \eta}$ can be upper bounded as follows.

Rewrite $\mathcal{X}_\epsilon^{lower}$ by rearranging terms as

$$\mathcal{X}_\epsilon^{lower} = \left\{ x \in \mathcal{X} \mid ||x - x^*|| \leq \left( \frac{\epsilon}{\sigma_-} \right)^{1/\alpha_{hi}} \overset{\text{def}}{=} r_\epsilon \right\} \tag{285}$$

and recall that from Corollary 4 that $\mathcal{X}_\epsilon \subseteq \mathcal{X}_\epsilon^{lower}$. Furthermore, an $\ell$-ball of radius $\psi \epsilon$ implies

$$\sigma_+ ||x-y||^{\alpha_{lo}} \leq \psi \epsilon \implies ||x-y|| \leq \left( \frac{\psi \epsilon}{\sigma_+} \right)^{1/\alpha_{lo}} \overset{\text{def}}{=} r_\ell. \tag{286}$$

The number of disjoint $\ell$-balls that can pack into a set $\mathcal{X}_\epsilon$, $N_{\epsilon \leq \eta}$, is upper bounded by the ratio of the volumes of the two sets:

$$N_{\epsilon \leq \eta} \leq \frac{Vol(\mathcal{X}_\epsilon)}{Vol(\mathcal{B}_\ell)} \tag{287}$$

$$\leq \frac{Vol(\mathcal{X}_\epsilon^{lower})}{Vol(\mathcal{B}_\ell)} \tag{288}$$

$$= \frac{S_d r_\epsilon^d}{S_d r_\ell^d} \tag{289}$$

$$\leq \frac{\left(\frac{\epsilon}{\sigma_-}\right)^{d/\alpha_{hi}}}{\left(\frac{\psi\epsilon}{\sigma_+}\right)^{d/\alpha_{lo}}} \tag{290}$$

$$= \left(\frac{\sigma_+^{1/\alpha_{lo}} \psi^{-1/\alpha_{lo}}}{\sigma_-^{1/\alpha_{hi}}}\right)^d \epsilon^{d(1/\alpha_{hi} - 1/\alpha_{lo})} \tag{291}$$

$$= \left(\frac{\sigma_+}{\psi\sigma_-^{\alpha_{lo}/\alpha_{hi}}}\right)^{d/\alpha_{lo}} \epsilon^{-d\left(\frac{\alpha_{hi}-\alpha_{lo}}{\alpha_{lo}\alpha_{hi}}\right)} \tag{292}$$

$$= C_{\epsilon \leq \eta} \epsilon^{-d'} \tag{293}$$

where $C_{\epsilon \leq \eta} = \left(\frac{\sigma_+}{\psi\sigma_-^{\alpha_{lo}/\alpha_{hi}}}\right)^{d/\alpha_{lo}}$ and $d' = d\left(\frac{\alpha_{hi}-\alpha_{lo}}{\alpha_{lo}\alpha_{hi}}\right)$ and $S_d$ is the volume constant for a $d$-sphere

under the given norm $||\cdot||$, e.g., $S_d = 2^d$ for $||\cdot||_\infty$. $\qquad\square$

**Corollary 6.** *If $\alpha_{lo} = \alpha_{hi} = \alpha$,*

$$N_{\epsilon \leq \eta} \leq \left(\frac{\kappa}{\psi}\right)^{d/\alpha}. \tag{294}$$

*In other words, $N_{\epsilon \leq \eta} \leq C_{\epsilon \leq \eta} \epsilon^{-d'}$ where $C_{\epsilon \leq \eta} = \left(\frac{\sigma_+}{\psi\sigma_-}\right)^{d/\alpha}$ and $d' = 0$.*

**Corollary 7.** *If Assumption 1 is given in terms of the 2-norm, these can be translated to bounds in terms of the $\infty$-norm resulting in the same $\psi$-near optimality dimension but incurring an additional exponential factor in the constant $C_{\epsilon \leq \eta}^{(\infty)} \leftarrow C_{\epsilon \leq \eta}^{(2)} d^{d/2}$.*

*Proof.* Recall that $||\cdot||_\infty \leq ||\cdot||_2 \leq \sqrt{d}||\cdot||_\infty$, therefore

$$f(x^*) - f(x) \leq \sigma_{+2}||x-x^*||_2^{\alpha_{lo}} \leq \sigma_{+\infty}||x-x^*||_\infty^{\alpha_{lo}} \tag{295}$$

$$f(x^*) - f(x) \geq \sigma_{-2}||x-x^*||_2^{\alpha_{hi}} \geq \sigma_{-\infty}||x-x^*||_\infty^{\alpha_{hi}} \tag{296}$$

where $\sigma_{+\infty} = \sigma_{+2}d^{\alpha_{lo}/2}$ and $\sigma_{-\infty} = \sigma_{-2}$. Then

$$C_{\epsilon \leq \eta}^{(\infty)} = \left(\frac{\sigma_{+2}d^{\alpha_{lo}/2}}{\psi\sigma_{-2}^{\alpha_{lo}/\alpha_{hi}}}\right)^{d/\alpha_{lo}} = \left(\frac{\sigma_{+2}}{\psi\sigma_{-2}^{\alpha_{lo}/\alpha_{hi}}}\right)^{d/\alpha_{lo}} d^{d/2} = C_{\epsilon \leq \eta}^{(2)} d^{d/2}. \tag{297}$$

$\qquad\square$

Recall, these results apply when $f(x) \geq f(x^*) - \eta$, i.e., when $\epsilon \leq \eta$. Otherwise, we can upper bound the number of $\ell$-balls by considering the entire set $\mathcal{X}$ which has volume 1. First, we will bound the constant associated with the volume of a $d$-sphere.

**Lemma 33.** *The volume of a $d$-sphere with radius $r$ and $d$ even is given by $S_d r^d$ where $S_d^{-1} \leq \sqrt{2\pi d}\left(\frac{d}{2\pi e}\right)^{d/2}$.*

*Proof.* First, we recall Stirling's bounds on the factorial: $\sqrt{2\pi n}(\frac{n}{e})^n e^{\frac{1}{12n+1}} < n! < \sqrt{2\pi n}(\frac{n}{e})^n e^{\frac{1}{12n}}$. This will be useful for bounding the Gamma function: $\Gamma(d) = (d-1)!$ for even $d$.

Given $d$ is even, we start with the exact formula for $S_d$:

$$S_d^{-1} = \frac{\Gamma(d/2+1)}{\pi^{d/2}} \tag{298}$$

$$= \frac{(d/2)!}{\pi^{d/2}} \tag{299}$$

$$< \frac{\sqrt{2\pi(d/2)}(\frac{d/2}{e})^{d/2}e^{\frac{1}{12(d/2)}}}{\pi^{d/2}} \tag{300}$$

$$= \frac{\pi^{1/2}d^{1/2}(\frac{d}{2e})^{d/2}e^{\frac{1}{6d}}}{\pi^{d/2}} \tag{301}$$

$$= \frac{\pi^{1/2}d^{(d+1)/2}e^{\frac{1}{6d}}}{(2\pi e)^{d/2}} \tag{302}$$

$$\leq \sqrt{2\pi d}\left(\frac{d}{2\pi e}\right)^{d/2}. \tag{303}$$

$\square$

**Lemma 34** ($N_{\epsilon\geq\eta} \leq C_{\epsilon\geq\eta}$). *The number of disjoint $\ell$-balls that can pack into a set $\mathcal{X}_{\epsilon\geq\eta}$, $N_{\epsilon\geq\eta}$, is upper bounded by $C_{\epsilon\geq\eta}$ where $C_{\epsilon\geq\eta} = S_d^{-1}\left(\frac{\sigma_+}{\psi\eta}\right)^{d/\alpha_{lo}}$ and $S_d$ is the volume constant for a $d$-sphere under a given norm.*

*Proof.* We can upper bound the number of $\ell$-balls needed to pack the entire space as follows:

$$N_{\epsilon\geq\eta} \leq \frac{Vol(\mathcal{X})}{Vol(\mathcal{B}_\ell)} \tag{304}$$

$$= \frac{1}{S_d r_\ell^d} \tag{305}$$

$$\leq \frac{1}{S_d\left(\frac{\psi\eta}{\sigma_+}\right)^{d/\alpha_{lo}}} \tag{306}$$

$$= S_d^{-1}\left(\frac{\sigma_+}{\psi\eta}\right)^{d/\alpha_{lo}} \tag{307}$$

$$= C_{\epsilon\geq\eta} \tag{308}$$

where $r_l$ was defined in equation (286). $S_d^{-1}$ has been upper bounded for the 2-norm in Lemma 33. For the $\infty$-norm, $S_d^{-1} = 2^{-d}$. $\square$

**Corollary 8.** *If Assumption 1 is given in terms of the 2-norm, these can be translated to bounds in terms of the $\infty$-norm resulting in the same $\psi$-near optimality dimension but incurring an additional exponential factor in the constant $C_{\epsilon\geq\eta}^{(\infty)} = \left(\frac{\sigma_{+2}^{1/\alpha_{hi}}}{2\eta^{1/\alpha_{lo}}}\right)^d C_{\epsilon\leq\eta}^{(\infty)} = \left(\frac{\sigma_{+2}^{1/\alpha_{hi}}}{2\eta^{1/\alpha_{lo}}}\right)^d C_{\epsilon\leq\eta}^{(2)} d^{d/2}.$*

*Proof.*

$$C_{\epsilon\geq\eta}^{(\infty)} = 2^{-d}\left(\frac{\sigma_{+2}}{\psi\eta}\right)^{d/\alpha_{lo}} d^{d/2} \tag{309}$$

$$= 2^{-d}\eta^{-d/\alpha_{lo}}\sigma_{-2}^{d/\alpha_{hi}}\left(\frac{\sigma_{+2}}{\psi\sigma_{-2}^{\alpha_{lo}/\alpha_{hi}}}\right)^{d/\alpha_{lo}} d^{d/2} \tag{310}$$

$$= 2^{-d}\eta^{-d/\alpha_{lo}}\sigma_{-2}^{d/\alpha_{hi}}C_{\epsilon\leq\eta}^{(\infty)} \tag{311}$$

$$= \left(\frac{\sigma_{+2}^{1/\alpha_{hi}}}{2\eta^{1/\alpha_{lo}}}\right)^d C_{\epsilon\leq\eta}^{(\infty)}. \tag{312}$$

$\square$

If we further assume $\alpha = \alpha_{lo} = \alpha_{hi} = 2$, then we can bound the number of $\ell$-balls required with a constant, independent of $\epsilon$, as

$$C = \max\{N_{\epsilon \leq \eta}, N_{\epsilon \geq \eta}\} \tag{313}$$

$$= \max\left\{ \left(\frac{\kappa}{\psi}\right)^{d/2}, \sqrt{2\pi d}\left(\frac{d\sigma_{max}}{2\pi e \psi \eta}\right)^{d/2} \right\} \tag{314}$$

$$= \beta^{d/2} \psi^{-d/2} d^{\xi/2(d+1)} \tag{315}$$

where $\beta = \kappa$, $\xi = 0$ for $N_{\epsilon \leq \eta}$ and $\beta = \frac{\sigma_{max}(2\pi)^{1/d}}{2\pi e \eta} < \frac{2\sigma_{max}}{\pi e \eta} = \frac{2\kappa}{\pi e r_\eta^2} < \frac{\kappa}{(2r_\eta)^2}$ for $d \geq 2$, $\xi = 1$ for $N_{\epsilon \geq \eta}$. $N_{\epsilon \geq \eta}$ dominates for large $d$. The cross over occurs at

$$\left(\frac{\kappa}{\psi}\right)^{d/2} = \sqrt{2\pi d}\left(\frac{d\sigma_{max}}{2\pi e \psi \eta}\right)^{d/2} \tag{316}$$

$$\implies \frac{\kappa}{\psi} = (2\pi d)^{1/d}\left(\frac{d\sigma_{max}}{2\pi e \psi \eta}\right) \tag{317}$$

$$\implies r_\eta^2 = \frac{\eta}{\sigma_-} = (2\pi d)^{1/d}\left(\frac{d}{2\pi e}\right) = z(d). \tag{318}$$

where $r_\eta$ was defined in equation (285). As $d$ grows and $z(d)$ exceeds $r_\eta^2$, $N_{\epsilon \geq \eta}$ begins to dominate, therefore we will upper bound $C$ as

$$C \leq \left(\frac{\kappa}{\psi(2r_\eta)^2}\right)^{d/2} d^{\frac{1}{2}(d+1)}. \tag{319}$$

| Locality | $C$ |
|---|---|
| $(*)$ $r_\eta^2 \leq z(d)$ | $N_{\epsilon \geq \eta} \leq \left(\frac{\kappa}{\psi(2r_\eta)^2}\right)^{d/2} d^{\frac{1}{2}(d+1)} = \left(\frac{3\kappa b^2}{(2r_\eta)^2}\right)^{d/2} d^{d+\frac{1}{2}}$ |
| $r_\eta^2 > z(d)$ | $N_{\epsilon \leq \eta} \leq \left(\frac{\kappa}{\psi}\right)^{d/2} = \left(3\kappa b^2\right)^{d/2} d^{d/2}$ |

Table 4: Bounding Constants for $\ell(x,y) = \|x - y\|_2^2$, $\psi = \nu/3 = (3b^2 d)^{-1}$ and $z(d) = (2\pi d)^{1/d}\left(\frac{d}{2\pi e}\right)$ with smoothness radius $r_\eta$ and $\psi = \nu/3$. $(*)$ indicates the case that is more likely for difficult problems.

For convenience, we repeat the other relevant constants in Table 5.

| $\ell(x,y)$ | $c$ | $\gamma$ | $\nu$ |
|---|---|---|---|
| $\ell(x,y) = \|x - y\|_2^2$ | $d\left(\frac{b}{2}\right)^2$ | $b^{-2/d}$ | $d^{-1}b^{-2}$ |

Table 5: Bounding Constants

# J  D-BLiN

The regret bound for Doubling BLiN [14] was originally proved assuming a standard normal distribution, however, the authors state their proof can be easily adapted to any sub-Gaussian distribution, which includes bounded random variables. This matches our setting with bounded payoffs, so we repeat their analysis here for that setting.

**Definition 5** (Global Arm Accuracy). $\mathcal{E} \stackrel{\text{def}}{=} \left\{ |\mu(x) - \hat{\mu}_m(C)| \leq r_m + \sqrt{c_1 \frac{\ln T}{n_m}}, \ \forall 1 \leq m \leq B_{stop} - 1, \ \forall C \in \mathcal{A}_m, \ \forall x \in C \right\}$.

Define: $n_m = c_2 \frac{\ln T}{r_m^2} \implies r_m = \sqrt{c_2 \frac{\ln T}{n_m}}$.

**Definition 6** (Elimination Rule). *Eliminate $C \in \mathcal{A}_m$ if $\hat{\mu}_m^{\max} - \hat{\mu}_m(C) \geq 2(1 + \sqrt{c_1/c_2})r_m = 2(\sqrt{c_2} + \sqrt{c_1})\sqrt{\frac{\ln T}{n_m}}$ where $\hat{\mu}_m^{\max} \overset{\text{def}}{=} \max_{C \in \mathcal{A}_m} \hat{\mu}_m(C)$.*

**Lemma 35.** $Pr[\mathcal{E}] \geq 1 - 2T^{-2(c_1/c^2 - 1)}$.

*Proof.* Assume $y_{C,i} \in [a, b]$ with $c = b - a$ and $\hat{\mu}(C) = \frac{1}{n_m}\sum_{i=1}^{n_m} y_{C,i}$. Applying a Hoeffding inequality gives

$$Pr\left[|\hat{\mu}(C) - \mathbb{E}[\hat{\mu}(C)]| \geq \sqrt{c_1 \frac{\ln T}{n_m}}\right] \leq 2e^{-2c_1 \ln T/c^2} \tag{320}$$

$$= 2(e^{\ln T})^{-2c_1/c^2} \tag{321}$$

$$= 2T^{-2c_1/c^2} \; \forall C. \tag{322}$$

By Lipschitzness of $\mu$ we also have

$$|\mathbb{E}[\hat{\mu}(C)] - \mu(x)| \leq r_m, \; \forall x \in C. \tag{323}$$

Then consider

$$\sup_{x \in C} |\mu(x) - \hat{\mu}(C)| = \sup_{x \in C} |\mu(x) - \mathbb{E}[\hat{\mu}(C)] + \mathbb{E}[\hat{\mu}(C)] - \hat{\mu}(C)| \tag{324}$$

$$\leq \sup_{x \in C} \left(|\mu(x) - \mathbb{E}[\hat{\mu}(C)]| + |\mathbb{E}[\hat{\mu}(C)] - \hat{\mu}(C)|\right) \tag{325}$$

$$= \sup_{x \in C} |\mu(x) - \mathbb{E}[\hat{\mu}(C)]| + |\mathbb{E}[\hat{\mu}(C)] - \hat{\mu}(C)| \tag{326}$$

$$\leq \sqrt{c_1 \frac{\ln T}{n_m}} + r_m \tag{327}$$

with probability $1 - 2T^{-2c_1/c^2}$. The first inequality follows by triangle inequality and the second follows from equation (323) and considering the complement of equation (322).

The complement of this result occurs with probability

$$Pr\left[\sup_{x \in C} |\mu(x) - \hat{\mu}(C)| \geq r_m + \sqrt{c_1 \frac{\ln T}{n_m}}\right] \leq 2T^{-2c_1/c^2}. \tag{328}$$

At least 1 arm is played in each cube $C \in \mathcal{A}_m$ for $1 \leq m \leq B_{stop} - 1$, therefore, $|\mathcal{A}_m| \leq T$ must be true given the exit condition of the algorithm. In addition, assume $B_{stop} \leq T$ ($B_{stop}$ will be defined such that this is true). Then a union bound over all $T^2$ events gives

$$Pr\left[\exists m \in [1, B_{stop} - 1], C \in \mathcal{A}_m \; s.t. \; \sup_{x \in C} |\mu(x) - \hat{\mu}(C)| \geq r_m + \sqrt{c_1 \frac{\ln T}{n_m}}\right] \tag{329}$$

$$\leq \sum_{m=1}^{B_{stop}-1} \sum_{C \in \mathcal{A}_m} Pr\left[\sup_{x \in C} |\mu(x) - \hat{\mu}(C)| \geq r_m + \sqrt{c_1 \frac{\ln T}{n_m}}\right] \tag{330}$$

$$\leq \sum_{m=1}^{B_{stop}-1} \sum_{C \in \mathcal{A}_m} 2T^{-2c_1/c^2} \tag{331}$$

$$\leq 2T^{-2c_1/c^2}T^2. \tag{332}$$

Taking the complement of this event and noting that $\sup_{x \in C} |\mu(x) - \hat{\mu}(C)| \leq r_m + \sqrt{c_1 \frac{\ln T}{n_m}} \implies |\mu(x) - \hat{\mu}(C)| \leq r_m + \sqrt{c_1 \frac{\ln T}{n_m}} \; \forall x \in C$ gives the desired result. $\square$

**Lemma 36** (Optimal Arm Survives). *Under event $\mathcal{E}$, the optimal arm $x^* = \arg\max \mu(x)$ is not eliminated after the first $B_{stop} - 1$ batches.*

*Proof.* Let $C_m^*$ denote the cube containing $x^*$ in $\mathcal{A}_m$. Under event $\mathcal{E}$, for any cube $C \in \mathcal{A}_m$ and $x \in C$, the following relation shows that $C_m^*$ avoids the elimination rule in round $m$:

$$\hat{\mu}(C) - \hat{\mu}(C_m^*) \leq \left(\mu(x) + r_m + \sqrt{c_1 \frac{\ln T}{n_m}}\right) + \left(-\mu(x^*) + r_m + \sqrt{c_1 \frac{\ln T}{n_m}}\right) \tag{333}$$

$$= \underbrace{(\mu(x) - \mu(x^*))}_{\leq 0} + 2r_m + 2\sqrt{c_1 \frac{\ln T}{n_m}} \tag{334}$$

$$\leq 2\sqrt{c_2 \frac{\ln T}{n_m}} + 2\sqrt{c_1 \frac{\ln T}{n_m}} \tag{335}$$

$$= 2(\sqrt{c_1} + \sqrt{c_2})\sqrt{\frac{\ln T}{n_m}} \tag{336}$$

where the first inequality follows from applying Lemma 35 to upper bound $\hat{\mu}(C)$ and $\hat{\mu}(C_m^*)$ individually. The remaining steps use the optimality of $x^*$, the definition of $r_m$, and the elimination rule. $\qquad\square$

**Lemma 37.** *Under event $\mathcal{E}$, for any $1 \leq m \leq B_{stop}$, any $C \in A_m$ and any $x \in C$, $\Delta_x$ satisfies*

$$\Delta_x \leq 4(1 + \sqrt{c_1/c_2})r_{m-1} \tag{337}$$

*Proof.* For $m = 1$, recall that $r_m$ is the side length of a cube $C \in \mathcal{A}_m$, therefore, $\Delta_x \leq r_{m-1} \leq 4(1 + \sqrt{c_1/c_2})r_{m-1}$ holds directly from the Lipschitzness of $\mu$.

For $m > 1$, let $C_{m-1}^* \in \mathcal{A}_{m-1}$ be the cube containing $x^*$. From Lemma 36, this cube has not been eliminated under event $\mathcal{E}$. For any cube $C \in \mathcal{A}_m$ and $x \in C$, it is clear that $x$ is also in the parent of $C$, denoted $C_{par}$ ($x \in C \subset C_{par}$). Then for any $x \in C$, it holds that

$$\Delta_x = \mu(x^*) - \mu(x) \leq \left(\hat{\mu}_{m-1}(C_{m-1}^*) + r_{m-1} + \sqrt{c_1 \frac{\ln T}{n_{m-1}}}\right) + \left(-\hat{\mu}_{m-1}(C_{par}) + r_{m-1} + \sqrt{c_1 \frac{\ln T}{n_{m-1}}}\right) \tag{338}$$

$$= (\hat{\mu}_{m-1}(C_{m-1}^*) - \hat{\mu}_{m-1}(C_{par})) + 2(\sqrt{c_1} + \sqrt{c_2})\sqrt{\frac{\ln T}{n_{m-1}}} \tag{339}$$

$$\leq (\hat{\mu}_{m-1}^{\max} - \hat{\mu}_{m-1}(C_{par})) + 2(\sqrt{c_1} + \sqrt{c_2})\sqrt{\frac{\ln T}{n_{m-1}}} \tag{340}$$

$$\leq 2(\sqrt{c_1} + \sqrt{c_2})\sqrt{\frac{\ln T}{n_{m-1}}} + 2(\sqrt{c_1} + \sqrt{c_2})\sqrt{\frac{\ln T}{n_{m-1}}} \tag{341}$$

$$= 4(\sqrt{c_1} + \sqrt{c_2})\sqrt{\frac{\ln T}{n_{m-1}}} \tag{342}$$

$$= 4(1 + \sqrt{c_1/c_2})r_{m-1} \tag{343}$$

where we have applied Lemma 35 similarly as in Lemma 36 and also used the definition of $r_{m-1}$. The last two inequalities use the fact that $\hat{\mu}_{m-1}(C_{m-1}^*) \leq \hat{\mu}_{m-1}^{\max}$ and $C_{par}$ was not eliminated. $\quad\square$

**Theorem 3.** *With probability exceeding $1 - 2T^{-2(c_1/c^2 - 1)}$, the $T$-step total regret $R(T)$ of BLiN with Doubling Edge-length Sequence (D-BLiN) [14] satisfies*

$$R(T) \leq 8(1 + \sqrt{c_1/c_2})(2c_2 + 1)\ln(T)^{\frac{1}{d_z + 2}} T^{\frac{d_z + 1}{d_z + 2}} \tag{344}$$

*where $d_z$ is the zooming dimension of the problem instance. In addition, D-BLiN only needs no more than $B^* = \frac{\log 2(T) - \log 2(\ln(T))}{d_z + 2} + 2$ rounds of communications to achieve this regret rate.*

*Proof.* Since $r_m = \frac{r_{m-1}}{2} \implies r_{m-1} = 2r_m$ for the Doubling Edge-length Sequence, Lemma 37 implies that every cube $C \in A_m$ is a subset of $S(8(1 + \sqrt{c_1/c_2})r_m)$. Thus from the definition of zooming number (Corollary 5 with appropriate condition), we have

$$|\mathcal{A}_m| \leq N_{r_m} \leq C_z r_m^{-d_z}. \tag{345}$$

Fix any positive number $B$. Also by Lemma 37, we know that any arm played after batch $B$ incurs a regret bounded by $8(1 + \sqrt{c_1/c_2})r_B$, since the cubes played after batch $B$ have edge length no larger than $r_B$. Then the total regret that occurs after batch $B$ is bounded by $8(1 + \sqrt{c_1/c_2})r_B T$ (where $T$ is an upper bound on the number of arms).

Thus the regret can be bounded as

$$R(T) \leq \sum_{m=1}^{B} \sum_{C \in \mathcal{A}_m} \sum_{i=1}^{n_m} \Delta_{x_{C,i}} + 8(1 + \sqrt{c_1/c_2})r_B T \tag{346}$$

where the first term bounds the regret in the first $B$ batches of D-BLiN, and the second term bounds the regret after the first $B$ batches. If the algorithm stops at batch $\tilde{B} < B$, we define $\mathcal{A}_m = $ for any $\tilde{B} < m \leq B$ and inequality equation (346) still holds.

By Lemma 37, we have $\Delta_{x_{C,i}} \leq 8(1 + \sqrt{c_1/c_2})r_m$ for all $C \in \mathcal{A}_m$. We can thus bound equation (346) by

$$R(T) \leq \sum_{m=1}^{B} |\mathcal{A}_m| \cdot n_m \cdot 8(1 + \sqrt{c_1/c_2})r_m + 8(1 + \sqrt{c_1/c_2})r_B T \tag{347}$$

$$\leq \sum_{m=1}^{B} N_{r_m} \cdot n_m \cdot 8(1 + \sqrt{c_1/c_2})r_m + 8(1 + \sqrt{c_1/c_2})r_B T \tag{348}$$

$$= \sum_{m=1}^{B} N_{r_m} \cdot c_2 \frac{\ln T}{r_m^2} \cdot 8(1 + \sqrt{c_1/c_2})r_m + 8(1 + \sqrt{c_1/c_2})r_B T \tag{349}$$

$$= \sum_{m=1}^{B} N_{r_m} \cdot \frac{\ln T}{r_m} \cdot 8c_2(1 + \sqrt{c_1/c_2}) + 8(1 + \sqrt{c_1/c_2})r_B T \tag{350}$$

where equation (348) uses equation (345), and equation (349) uses equality $n_m = c_2 \frac{\ln T}{r_m^2}$. Since $r_m = 2^{-m+1}$ and $N_{r_m} \leq r_m^{-d_z} \leq 2^{(m-1)d_z}$, we have

$$R(T) \leq \sum_{m=1}^{B} 2^{(m-1)d_z} \cdot \frac{\ln T}{2^{-m+1}} \cdot 8c_2(1 + \sqrt{c_1/c_2}) + 8(1 + \sqrt{c_1/c_2})2^{-B+1}T \tag{351}$$

$$= 8(1 + \sqrt{c_1/c_2})\left[c_2 \ln T \sum_{m=1}^{B} 2^{(m-1)(d_z+1)} + 2^{-B+1}T\right]. \tag{352}$$

Continuing we find

$$R(T) \leq 8(1 + \sqrt{c_1/c_2})\Big[c_2 \ln T \sum_{m=1}^{B} 2^{(m-1)(d_z+1)} + 2^{-B+1}T\Big] \tag{353}$$

$$= 8(1 + \sqrt{c_1/c_2})\Big[c_2 \ln T \sum_{m=1}^{B} \big(2^{d_z+1}\big)^{m-1} + 2^{-B+1}T\Big] \tag{354}$$

$$= 8(1 + \sqrt{c_1/c_2})\Big[c_2 \ln T \sum_{m=0}^{B-1} \big(2^{d_z+1}\big)^{m} + 2^{-B+1}T\Big] \tag{355}$$

$$= 8(1 + \sqrt{c_1/c_2})\Big[c_2 \ln T \Big(\frac{2^{B(d_z+1)} - 1}{2^{d_z+1} - 1}\Big) + 2^{-B+1}T\Big] \quad \text{via geometric series} \tag{356}$$

$$\leq 8(1 + \sqrt{c_1/c_2})\Big[c_2 \ln T \Big(\frac{2^{B(d_z+1)}}{2^{d_z+1} - 1}\Big) + 2^{-B+1}T\Big] \tag{357}$$

$$\leq 8(1 + \sqrt{c_1/c_2})\Big[c_2 \ln T \Big(2 \cdot \frac{2^{B(d_z+1)}}{2^{d_z+1}}\Big) + 2^{-B+1}T\Big] \tag{358}$$

$$= 8(1 + \sqrt{c_1/c_2})\Big[2c_2 2^{(B-1)(d_z+1)} \ln T + 2^{-(B-1)}T\Big]. \tag{359}$$

This inequality holds for any positive $B$. By choosing $B^* = 1 + \frac{\log_2(\frac{T}{\ln T})}{d_z+2}$, we have

$$R(T) \leq 8(1 + \sqrt{c_1/c_2})\Big[2c_2\Big(\frac{T}{\ln T}\Big)^{\frac{(d_z+1)}{(d_z+2)}} \ln T + \Big(\frac{\ln T}{T}\Big)^{\frac{1}{(d_z+2)}} T\Big] \tag{360}$$

$$= 8(1 + \sqrt{c_1/c_2})\Big[2c_2 T^{\frac{(d_z+1)}{(d_z+2)}} \ln T^{1 - \frac{(d_z+1)}{(d_z+2)}} + T^{1 - \frac{1}{(d_z+2)}} \ln T^{\frac{1}{(d_z+2)}}\Big] \tag{361}$$

$$= 8(1 + \sqrt{c_1/c_2})\Big[2c_2 T^{\frac{(d_z+1)}{(d_z+2)}} \ln T^{\frac{1}{(d_z+2)}} + T^{\frac{(d_z+1)}{(d_z+2)}} \ln T^{\frac{1}{(d_z+2)}}\Big] \tag{362}$$

$$= 8(1 + \sqrt{c_1/c_2})(2c_2 + 1)T^{\frac{(d_z+1)}{(d_z+2)}} \ln T^{\frac{1}{(d_z+2)}}. \tag{363}$$

$\square$

**Corollary 9.** *Setting $c_1 = 2c^2$ and $c_2 = \big(\frac{c^2}{2}\big)^{1/3}$ simplifies Theorem 3 such that*

$$R(T) \leq 8(1 + (4c^2)^{1/3})^2 T^{\frac{(d_z+1)}{(d_z+2)}} \ln T^{\frac{1}{(d_z+2)}}. \tag{364}$$

*with probability $1 - 2T^{-2}$.*

*Proof.* $\square$

# K  Experimental Setup and Details

Here we provide further details on the experiments.

## K.1  Loss Visualization and Rank Test

Figure 1 and claims made in Section 5 analyze several classical matrix games. We report the payoff matrices in standard row-player / column-player payoff form below. All games are then shifted and scaled so payoffs lie in $[0, 1]$ (i.e., first by subtracting the minimum and then scaling by the max).

RPS:

$$\begin{bmatrix} 0/0 & -1/1 & 1/-1 \\ 1/-1 & 0/0 & -1/1 \\ -1/1 & 1/-1 & 0/0 \end{bmatrix}. \tag{365}$$

Chicken:

$$\begin{bmatrix} 0/0 & -1/1 \\ 1/-1 & -3/-3 \end{bmatrix}. \tag{366}$$

Matching Pennies:

$$\begin{bmatrix} 1/-1 & -1/1 \\ -1/1 & 1/-1 \end{bmatrix}. \tag{367}$$

Modified-Shapleys:

$$\begin{bmatrix} 1/-0.5 & 0/1 & 0.5/0 \\ 0.5/0 & 1/-0.5 & 0/1 \\ 0/1 & 0.5/0 & 1/-0.5 \end{bmatrix}. \tag{368}$$

Prisoner's Dilemma:

$$\begin{bmatrix} -1/-1 & -3/0 \\ 0/-3 & -2/-2 \end{bmatrix}. \tag{369}$$

### K.1.1 NashConv is Biased

We use Chicken to demonstrate the effect of sampled play on the bias of the popular NashConv loss. NashConv is unable to capture the interior Nash equilibrium because of its high bias. In contrast, our proposed loss $\mathcal{L}^\tau$ is guaranteed to capture all equilibria at low temperature $\tau$.

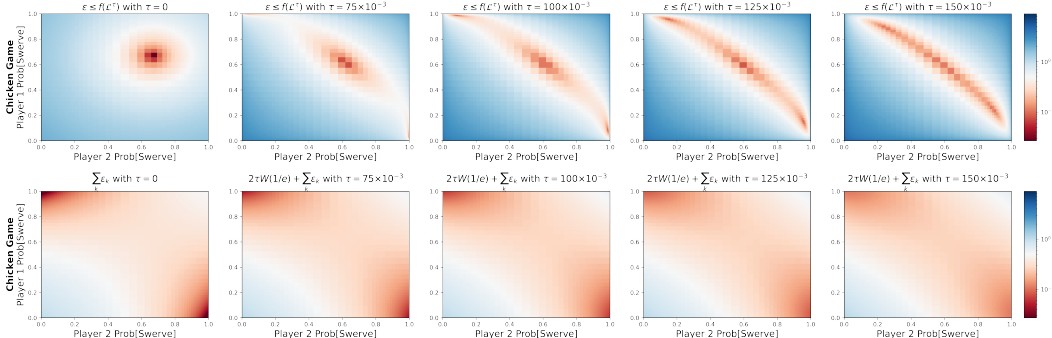

Figure 5: Effect of Sampled Play on a Biased Loss. The first row displays the expected upper bound guaranteed by our proposed loss $\mathcal{L}^\tau$ (also displayed in Figure 1). The second row displays the expectation of NashConv under sampled play, i.e., $\sum_k \epsilon_k$ where $\epsilon_k = \mathbb{E}_{a_{-k} \sim x_{-k}}[\max_{a_k} u_k^\tau(\boldsymbol{a})] - \mathbb{E}_{\boldsymbol{a} \sim \boldsymbol{x}}[u_k^\tau(\boldsymbol{a})]$. To be consistent, we add the offset $n\tau W(1/e) + \sum_k \epsilon_k$ to NashConv per Lemma 14, which relates the exploitability at positive temperature to that at zero temperature. The resulting loss surface clearly shows NashConv fails to recognize the interior Nash equilibrium due to its inherent bias. NashConv succeeds in finding pure equilibria because sampling from a pure joint equilibrium is a deterministic process (no noise means no bias).

### K.2 Saddle Point Analysis

To generate Figure 2, we follow a procedure similar to the study of MNIST in [12] (Section 3 of Supp.). Their recommended procedure searches for critical points in two ways. The first repeats a randomized, iterative optimization process 20 times. They then sample one these 20 trials at random, select a random point along the descent trajectory, and search for a critical point (using Newton's method) nearby. They repeat this sampling process 100 times. The second approach randomly selects

a feasible point in the decision set and searches for a critical point nearby (again using Newton's method). They also perform this 100 times.

Our protocol differs from theirs slightly in a few respects. One, we use SGD, rather than the saddle-free Newton algorithm to trace out an initial descent trajectory. Two, we do not add noise to strategies along the descent trajectory prior to looking for critical points. Lastly, we use different experimental hyperparameters. We run SGD for 1000 iterations rather than 20 epochs and rerun SGD 100 times rather than 20. We sample 1000 points for each of the two approaches for finding critical points.

### K.3 SGD on Classical Games

The games examined in Figure 3 were all taken from [15]. Each is available via open source implementations in OpenSpiel [22] or GAMUT [33].

We compare against several other baselines, replicating the experiments in [15]. RM indicates regret-matching and FTRL indicates follow-the-regularized-leader. These are, arguably, the two most popular scalable stochastic algorithms for approximating Nash equilibria. $^y\text{QRE}^{auto}$ is a stochastic algorithm developed in [15].

For each of the experiments, we sweep over learning rates in log-space from $10^{-3}$ to $10^2$ in increments of $1$. We also consider whether to run SGD with the projected-gradient and whether to constrain iterates to the simplex via Euclidean projection or entropic mirror descent [6]. We then presented the results of the best performing hyperparameters. This was the same approach taken in [15].

**Saddle Points in Blotto**  To confirm the existence of saddle points, we computed the Hessian of $\mathcal{L}(\boldsymbol{x_{10k}})$ for SGD ($s = \infty$), deflated the matrix by removing from its eigenvectors all directions orthogonal to the simplex, and then computed its top-$(n\bar{m} - n)$ eigenvalues. We do this because there always exists a $n$-dimensional nullspace of the Hessian at zero temperature that lies outside the tangent space of the simplex, and we only care about curvature within the tangent space. Specificaly, at an equilibrium $\boldsymbol{x}$, if we compute $z^\top Hess(\mathcal{L})z$ where $z$ is formed as a linear combination of the vectors $\{[x_1, 0, \ldots, 0]^\top, \ldots, [0, \ldots, x_n]^\top\}$, then each block $\tilde{B}_{kl}$ is identically zero at an equilibrium: $\tilde{B}_{kl}x_l = \sqrt{\eta_k}[I - \frac{1}{m_k}\mathbf{1}\mathbf{1}^\top]H_{kl}^k x_l = \sqrt{\eta_k}\Pi_{T\Delta}(\nabla_{x_k}^k) = 0$. By Lemma 17, this implies there is zero curvature of the loss in the direction $z$: $z^\top Hess(\mathcal{L})z = 0$.

### K.4 BLiN on Artificial Game

To construct the 7-player, 2-action, symmetric, artifical game in Figure 4, we used the following coefficients (discovered by trial-and-error):

$$
\begin{bmatrix}
0.09906873 & 0 & 0.23116037 & 0 & 0.62743528 & 0 & 0.19813746 \\
0 & 0.33022909 & 0 & 0.03302291 & 0 & 0.62743528 & 0
\end{bmatrix}. \tag{370}
$$

The first row indicates the payoffs received when player $i$ plays action $0$ and the background population plays any of the possible joint actions (number of combinations with replacement). For example, the first column indicates the payoff when all background players play action $0$. The second column indicates all background players play action $0$ except for one which plays action $1$, and so on. The last column indicates all background players play action $1$. These $2n$ scalars uniquely define the payoffs of a symmetric game.

Given that this game only has two actions, we represent a mixed strategy by a single scalar $p \in [0, 1]$, i.e., the probability of the first action. Furthermore, this game is symmetric and we seek a symmetric equilibrium, so we can represent a full Nash equilibrium by this single scalar $p$. This reduces our search space from $7 \times 2 = 14$ variables to $1$ variable (and obviates any need for a map $s$ from the unit hypercube to the simplex—see Lemma 24).