# OpenReview forum: "Approximating Nash Equilibria in Normal-Form Games via Unbiased Stochastic Optimization"
_NeurIPS.cc/2023/Conference — Submitted to NeurIPS 2023_

### Official Review · Reviewer_4TEh · 2023-06-19

**Soundness:** 2 fair
**Presentation:** 3 good
**Contribution:** 2 fair
**Rating:** 4
**Confidence:** 4

**Summary:**

This paper formulates a Lipschitz loss function that makes computing approximate interior Nash equilibria in normal-form games amenable to unbiased Monte Carlo estimation, opening the door to using a number of scalable stochastic optimization techniques. They also provide a loss function with similar properties but under the notion of quantal-response equilibrium (QRE). The authors also provide certain illustrative experiments to support their claims.

**Strengths:**

This paper provides a novel approach to computing equilibria in multi-player games. In particular, the authors derive loss functions that make equilibrium computation amenable to scalable methods from stochastic optimization. Given the lack of scalable algorithms for computing solutions concepts such as the Nash equilibrium, this is a promising approach, and has the potential to bring many new insights to equilibrium computation. In particular, the idea proposed for deriving an unbiased estimator (Section 4.4) is interesting, and addresses many of the pitfalls of other commonly used loss functions in the more challenging constrained setting. Furthermore, the presentation and the writing are overall clear, and the authors accurately place their results into the existing literature.

**Weaknesses:**

There are a number of issues that weaken the contribution of the paper. First, the underlying assumption that there is an interior Nash equilibrium is very strong. For one, if there is an interior NE it is known that it can be computed in polynomial time via linear programming, which significantly weakens the motivation regarding the hardness of NE. There appears to be some confusion regarding this point. For example, Corollary 3 in the Appendix claims a new FPTAS for computing interior NE in polymatrix games, which I believe is known (beyond polymatrix games); there is perhaps still some benefit in using the proposed methodology in practice, but no evidence of that is provided in the paper. (As an aside, it would be helpful to clarify in the prelimaries that by interior you mean relative interior.) Beyond the very restrictive assumption of having an interior Nash equilibrium, the authors provide similar results for QRE, but that is a significantly weaker equilibrium concept. I would also strongly recommend clarifying in the abstract that your results apply for interior NE; as it is written currently it is very misleading.

Besides the issue mentioned above, there is an underlying premise in the proposed methodology which I find unconvincing: Why should we expect local optima in the formulated loss functions to give meaningful guarantees? The fact that this turned out to be the case in many ML applications is not enough to justify this proposition. It is a significant weakness that the proposed method has no theoretical finite-time guarantees of reaching a Nash equilibrium.

And unfortunately the experiments do not offer enough evidence to support this approach. Indeed, there are many issues in the experiments that can be significantly improved. First, the games experimented on are overly small; for example, Shapley's game is completely toy, no meaningful conclusions can be drawn from it. Since the main message of this paper is about scalability, I expected to see experiments on much bigger games. It would be helpful if the proposed theory applied in extensive-form games for which there are many large benchmark games in the literature, but the current method is tailored to normal-form games. Besides this issue, I am very confused regarding the compared benchmarks. It is claimed in the last sentence of the abstract that the method often outperforms prior state of the art, but the main algorithms compared against are RM and FTRL. These algorithms will not even find an NE in finite-time, how can you claim that those are the state of the art? In particular, in Lines 966-969 it is claimed that those are the two most popular scalable stochastic algorithms for approximating NE; I strongly disagree with this claim. I would suggest trying some other benchmarks, such as the Lemke-Howson algorithm; a mixed-integer programming approach; or the algorithm presented in "Exclusion Method for Finding Nash Equilibrium in Multiplayer Games."

Another issue is on the proof of Corollary 1. For a constant $\epsilon$, it is claimed that you have a poly-time algorithm (since it is a PRAS), but you also say that the temperature parameter has to be exponentially small to achieve that. So it seems that even for a constant $\epsilon$ you need an exponential number of iterations to converge.

Overall, although the approach proposed is promising, there are a number of issues that have to be addressed before the paper is ready for publication.


**Questions:**

Some minor issues:

1. There are many missing punctuation marks in the equations throughout the paper
2. There are many overfull equations in the Appendix; I would recommend fixing those in the revised version

**Limitations:**

The authors have addressed the limitations.

---

> ### Author Rebuttal · Authors · 2023-08-09
>
> Thank you for your review and your encouraging statements. We believe we can address your concerns by clearing up a few misunderstandings and reporting the results of some additional experiments in accordance with your feedback. We hope you will consider increasing your score in light of these updates.
>
> **Interior (Fully-mixed) Assumption**: We do **not** assume there exists a Nash equilibrium in the interior of the simplex. That assumption only exists for the “warm-up” to aid the reader in our derivation of an appropriate loss function. Starting in Section 4.5, this assumption is removed and Lemma 17 expresses a key result that connects our loss function back to the standard $\epsilon$-Nash definition for any approximate equilibrium (including pure equilibria).
>
> **Local Optima**: We **agree with you** that local (suboptimal) optima could be a problem, and this is why we directly explore this point empirically. Our critical point experiments as showcased in Figure 2 are meant to study the frequency of encountering these saddle points / local minima in practice. Interestingly enough, although suboptimal saddle points are prevalent, the more critical case of local minima seems rarer.
>
> The critical part of Figure 2 is the top left corner. This corresponds to local minima of our loss function (no descent directions exist) that have relatively high levels of exploitability. This part of the figure shows that only one of our games (2p Sheriff) has several such bad local optima. But even for this game, as the first column shows, the vast majority of local optima have very low exploitability. In all other games, high exploitability is strongly correlated with large index and thus SGD should perform well.
>
> **Theoretical Finite Time Guarantees**: The primary theorem (Theorem 1) in our paper concerns a globally convergent stochastic non-convex optimization algorithm (BLiN) applied to our loss. This theorem expresses a finite time convergence rate to a **global, not local** optimum.
>
> **Insufficient Experiments (Toy Games and Missing Baselines)**: We experiment on games much larger than Shapley’s. Our largest Blotto game examined in Figure 3 contains $4 \cdot 66^4 > 75,000,000$ payoff entries. As far as NFGs go, these are quite large, and as we’ll demonstrate below, present a challenge for classical solvers. We also agree that extending our approach to EFGs is quite interesting, but is out of scope for the current paper. Note that in our experiments, we measure exploitability of an approximate equilibrium exactly. If the game is too large, this measurement becomes intractable by nature of a sum over an exponential $nm^n$ number of payoff entries. In this case, to our knowledge, our loss (which requires only $2nm$ lookups to evaluate) presents the only option for estimating an unbiased upper bound on $\epsilon$.
>
> Lemke-Howson only applies to 2-player games, however, Govinda and Wilson developed a method that is now recognized as its counterpart for 3+ player games. We ran this algorithm (*gambit-gnm*) and several others from the gambit library [1] (listed below) on both Blotto games. Only *gambit-enumpoly* and *gambit-enumpure* are able to return any NE for 3-player Blotto within a 1 hour time limit (and only pure equilibria). And only *gambit-enumpure* returns any NE for the 4-player game. Note we also test on the D7-Covariant game from the GAMUT benchmark set (Figure 3, second plot), which was revealed to be a particularly challenging game to solve for a set of classical methods (including Govinda Wilson, simplicial subdivision, and CSP-style approaches) in the paper by Porter, Nudelman, and Shoham [2].
>
> Thank you for sharing the paper “*Exclusion Method for Finding Nash Equilibrium in Multiplayer Games*”! In fact, BLiN applied to our loss can be thought of as a stochastic generalization of this method (see Figure 1 of X-armed Bandits [8] and Figure 1 of BLiN [14] for nice visuals of the X-armed bandit family approach that parallel the *exclusion method*). In the paper you cite, the regret calculated in their definition (1) is exactly the $\epsilon$ that we have developed an unbiased estimate for.
>
> Lastly, we consider the previous state-of-the-art to be ADIDAS [15] (denoted by $^yQRE^{auto}$ in the legend); note that [15] similarly includes negative results from running gambit on Blotto (see appendix H.2). Again, thank you for reading the appendix. FTRL and RM are popular methods that researchers often resort to [3, 4] because so few algorithms currently scale to large many player games.
>
> **PRAS Designation**:
> Thank you for catching this! The phrasing in the appendix is not perfectly accurate as the relevant temperature parameter is $\tau$ which is equal to $1/\ln(1/p)$. In this case, any exponential decrease for $p$ manifests as a linear decrease for $\tau$ due to the natural logarithm. We will make sure to resolve this subtlety in the appendix and make this clearer in the paper.
>
> Thank you again for the careful reading of our paper. **Proposed edits will follow in another post (space limits).**
>
> [1] McKelvey, Richard D., McLennan, Andrew M., and Turocy, Theodore L. (2016). Gambit: Software Tools for Game Theory, Version 16.0.1. http://www.gambit-project.org.
>
> https://gambitproject.readthedocs.io/en/latest/tools.html
>
> - gambit-enumpoly [73s 3-player, timeout 4-player]
> - gambit-enumpure [72s 3-player, 45s 4-player]
> - gambit-gnm
> - gambit-ipa
> - gambit-liap
> - gambit-logit
> - gambit-simpdiv
>
> [2] Porter, Ryan, Eugene Nudelman, and Yoav Shoham. "Simple search methods for finding a Nash equilibrium." Games and Economic Behavior 63.2 (2008): 642-662.
>
> [3] Bakhtin, Anton, et al. "Mastering the Game of No-Press Diplomacy via Human-Regularized Reinforcement Learning and Planning." The Eleventh International Conference on Learning Representations. 2022.
>
> [4] Gray, Jonathan, et al. "Human-Level Performance in No-Press Diplomacy via Equilibrium Search." International Conference on Learning Representations. 2020.

---

> > ### Comment · Reviewer_4TEh · 2023-08-11
> >
> > I thank the authors for the detailed response. I am satisfied with your clarifications regarding the experimental evaluation.
> >
> > The assumption of having fully mixed Nash equilibria still puzzles me; starting from Section 4.5 you are focusing on QRE not Nash equilibria. So based on your response it is fair to say that the main focus of this paper is on QRE and not Nash equilibria. If that is the case, then at the very least this has to be highlighted (for example, in the title, the abstract, and the introduction). This has a bearing on the evaluation of the paper since QRE is arguably a much less attractive solution concept, and is not what the paper promises based on the title and the abstract. Let me know if I am misunderstanding some point.

---

> > > ### Author Response · Authors · 2023-08-14
> > > **Solution Concept (Nash) vs Algorithm (Regularization and Homotopies)**
> > >
> > > Thank you for your quick response. We believe our discussion below can help clarify your concerns about the paper’s framing. To explain, we want to differentiate between the solution concept we are studying (Nash) and the algorithmic approach we take to approximate it (QREs at low temperature).
> > >
> > > **QRE: An Algorithmic Approach To Computing Nash**: Even in much more restrictive classes of games (e.g., monotone games), introducing and then annealing strong regularizers is a traditional and practical approach to approximating Nash equilibria: Tikhonov regularization [1] (monotone), Friction FoReL [2] (monotone), ADIDAS [3] (non-monotone). In addition, McKelvey and Palfrey introduced the QRE solution concept and immediately used it to compute a Nash equilibrium by annealing the temperature of a QRE to zero [4]. Our point is that each of the approaches above regularize the game, thereby solving for an intermediate yet transient solution concept, in order to solve for the ultimately desired solution of Nash equilibrium. This is a well-established technique in the literature for designing algorithms with convergence to Nash.
> > >
> > > **Our Focus: Nash Equilibrium**: Our paper focuses on constructing an algorithm with convergence guarantees to approximate Nash equilibria (as measured by exploitability $\epsilon$) in general-sum games. We demonstrate this focus both theoretically and empirically.
> > > 1) Lemma 17 establishes a result that allows us to upper bound the exploitability $\epsilon$ of a strategy profile $\boldsymbol{x}$ as a function of our loss (norms of entropy-regularized gradients). Lemma 17 explains how to approximate Nash equilibria given QRE as a stepping stone. QRE is not the final end goal.
> > > 2) Theorem 1 uses Lemma 17 in conjunction with non-convex optimization guarantees to provide a convergence rate to approximate Nash equilibria -- note we still measure our approximation error by $\epsilon$.
> > > 3) Lastly, experimental performance in Figure 3 is measured without entropy regularization. We’ll make sure to emphasize this in the updated version.
> > >
> > > In summary, our theory and evaluation metrics use Nash exploitability as the yardstick (and not e.g., distance to a QRE).
> > >
> > > We appreciate, and with hindsight, **agree** with your concern that readers might miss this difference (solution vs algorithm). We will make sure to emphasize this difference throughout the text explaining we approximate Nash equilibria (our solution concept of focus) by way of approximating QREs at vanishing temperature (our algorithmic approach).
> > >
> > > Thank you again for bringing this to our attention.
> > >
> > > [1] Facchinei, Francisco, and Jong-Shi Pang, eds. Finite-dimensional variational inequalities and complementarity problems. New York, NY: Springer New York, 2003. Page 1125.
> > >
> > > [2] Perolat, Julien, et al. "From Poincaré recurrence to convergence in imperfect information games: Finding equilibrium via regularization." International Conference on Machine Learning. PMLR, 2021.
> > >
> > > [3] Gemp, Ian, et al. "Sample-based Approximation of Nash in Large Many-Player Games via Gradient Descent." Proceedings of the 21st International Conference on Autonomous Agents and Multiagent Systems. 2022.
> > >
> > > [4] McKelvey, Richard D., and Thomas R. Palfrey. "Quantal response equilibria for normal form games." Games and economic behavior 10.1 (1995): 6-38.

---

> > > > ### Comment · Reviewer_4TEh · 2023-08-16
> > > >
> > > > Thank you for the response.
> > > >
> > > > I need some further clarifications regarding Theorem 1. I don't see how the assumption of having a well-isolated equilibrium affects Theorem 1; is it through the zooming dimension? Also, if there is a unique NE, it is presumably well-isolated; are you claiming that your algorithm finds it in polynomial time?

---

> > > > > ### Author Response · Authors · 2023-08-16
> > > > >
> > > > > Correct, it is through the zooming dimension. An equilibrium with a positive definite Hessian (i.e., *well* isolated equilibrium) implies $d_z = 0$. We mention this briefly on lines 264-265, but do not explicitly use the term *well* isolated. We will add that to the text for clarity.
> > > > >
> > > > > It's possible for the loss $\mathcal{L}^{\tau}$ at a unique NE to have a positive semi-definite Hessian. For example, the loss landscape $\mathcal{L}^{\tau}$ could be lower bounded by a polynomial of higher degree along some dimensions around the equilibrium (e.g., qualitatively similar to the function $x^2 + y^4$). In this case, the zooming dimension would be $d_z = n\bar{m} \frac{4 - 2}{4 \cdot 2} = \frac{n\bar{m}}{4}$. In this case, the resulting convergence rate implied by Theorem 1 would not be polynomial.
> > > > >
> > > > > Our approach of building on the non-convex optimization theory of BLiN requires stating our convergence result in terms of zooming dimension. This is inherent in the approach of theorem 1 and thus it cannot imply polynomial rates for all games with unique equilibrium.

---

> > > > > > ### Comment · Reviewer_4TEh · 2023-08-16
> > > > > >
> > > > > > Thank you for the quick reply. It seems that I am missing something: isn't a unique NE necessarily well-isolated? Could you provide the definition of a well-isolated equilibrium?

---

> > > > > > > ### Author Response · Authors · 2023-08-16
> > > > > > >
> > > > > > > We introduce the definition of *well* isolated on lines 226-228. In short, if the Hessian of $\mathcal{L}^{\tau}$ at an equilibrium is positive definite, then we call the equilibrium *well* isolated.
> > > > > > >
> > > > > > > In general, we would agree with you that a unique NE is isolated. By our definition, it is not necessarily *well* isolated per the example we gave above (e.g., $x^2 + y^4$).

---

> > > > > > > > ### Comment · Reviewer_4TEh · 2023-08-16
> > > > > > > >
> > > > > > > > Thanks for the clarification. Has this terminology ("well-isolated") being used in prior work? I don't see how the term "well-isolated" reflects the actual definition. Can the authors elaborate on this point?

---

> > > > > > > > > ### Author Response · Authors · 2023-08-16
> > > > > > > > >
> > > > > > > > > As far as we know, the term *well*-isolated is new. We agree it is a bit vague and are happy to consider suggestions. Maybe *strongly*-isolated to borrow from convex optimization theory? Or *polymatrix*-isolated given that the Hessian depends only on the polymatrix approximation to the game at an equilibrium (i.e., the equilibrium is *polymatrix*-isolated if its equilibrium is isolated according to its local polymatrix approximation)?

---

> > > > > > > > > > ### Author Response · Authors · 2023-08-19
> > > > > > > > > >
> > > > > > > > > > Given our lengthy and productive discussion, we want to take the opportunity to summarize our mutual understandings as we see them.
> > > > > > > > > >
> > > > > > > > > > i) You are now satisfied with our clarifications regarding the experimental evaluation. We have reported additional results already previously cited in the literature, that acknowledge the inability of classical solvers (e.g., gambit) to scale to large game instances. We have also compared and contrasted our approach to the nearest in the literature, which you kindly identified, the *exclusion method*.
> > > > > > > > > >
> > > > > > > > > > ii) We have clarified a distinction between the algorithmic approach we use and the solution concept we judge performance of our algorithm by.
> > > > > > > > > >
> > > > > > > > > > ii) Our convergence guarantees rely on a notion of *well*-isolated equilibria, a refinement of isolated equilibria (i.e., every *well*-isolated equilibrium is an isolated equilibrium). We are happy to change the name of this definition.
> > > > > > > > > >
> > > > > > > > > > We would ask you to kindly consider raising your score if you agree with this summary in which we have satisfied your concerns.

---

> > > > > > > > > > > ### Comment · Reviewer_4TEh · 2023-08-19
> > > > > > > > > > >
> > > > > > > > > > > I thank the authors again for their time. I will consider increasing my score as the discussions with the fellow reviewers progress. I currently maintain most of my original concerns. On the experimental side, the bulk of the literature on computational game theory focuses on extensive-form games, so I am not convinced that this paper will have a significant impact on that front since the techniques are tailored to normal-form games. On the theoretical side, much of the results (like Theorem 1) depend on notions of dimension that appear to be artificial from a game-theoretic standpoint, and are disconnected with the practical applications targeted in this paper.
> > > > > > > > > > >
> > > > > > > > > > > That being said, there are certainly merits in the paper as I acknowledged in my original review, so I might reconsider my evaluation as the discussions go forward.

---

> > > > > > > > > > > > ### Author Response · Authors · 2023-08-21
> > > > > > > > > > > > **Immediate Impact Beyond NFG (1/2)**
> > > > > > > > > > > >
> > > > > > > > > > > > Thank you for your response and for your continued engagement. We appreciate your willingness to continue weighing the paper’s merits.
> > > > > > > > > > > >
> > > > > > > > > > > > First, looking back at your initial review, we would like to acknowledge that you raised the issue of EFGs, *“It would be helpful if the proposed theory applied in extensive-form games for which there are many large benchmark games in the literature, but the current method is tailored to normal-form games.”*, and we did not adequately address your concern.
> > > > > > > > > > > >
> > > > > > > > > > > > We would like to rectify that here. There are two ways our technique extends to EFGs. One **a)** is immediate and standard in the literature, the other **b)** is future work.
> > > > > > > > > > > >
> > > > > > > > > > > > **a)** Policy-Space Response Oracles (PSRO) [1] generalizes a technique Double-Oracle [2] in a way that allows NFG techniques to be applied to a variety of settings including EFGs and Markov games. The original PSRO paper showed how to solve EFGs using Leduc poker as an example and has sparked a rich line of follow-up research. PSRO was even used to develop alphaStar (see AlphaStar Nash League [3]), a superhuman AI that defeated humans in real time StarCraft [4].
> > > > > > > > > > > >
> > > > > > > > > > > > [1] Lanctot, Marc, et al. "A unified game-theoretic approach to multiagent reinforcement learning." Advances in neural information processing systems 30 (2017).
> > > > > > > > > > > >
> > > > > > > > > > > > [2] McMahan, H. Brendan, Geoffrey J. Gordon, and Avrim Blum. "Planning in the presence of cost functions controlled by an adversary." Proceedings of the 20th International Conference on Machine Learning (ICML-03). 2003.
> > > > > > > > > > > >
> > > > > > > > > > > > [3] Vinyals, Oriol et al. Alphastar: Mastering the real-time strategy game starcraft ii. DeepMind blog (2019), 2.
> > > > > > > > > > > >
> > > > > > > > > > > > [4] Vinyals, Oriol et al. Grandmaster level in StarCraft II using multi-agent reinforcement learning. Nature 575, 7782 (2019), 350–354.
> > > > > > > > > > > >
> > > > > > > > > > > > **b)** In our original response to your concern, we stated *“We also agree that extending our approach to EFGs is quite interesting, but is out of scope for the current paper.”* We did not mean to imply *“the techniques are tailored to normal-form games”* in a way that prevents these techniques from being re-applied to EFGs. It would be premature to answer this question here affirmatively, but it is clear to us how one would extend our approach to EFGs via the sequence form which, similarly to the simplex, defines EFG strategies via a set of linear constraints. In fact, several NFG techniques have been directly adapted to EFGs through this connection [5, 6] including one of this year’s most recent Outstanding papers at ICML 2023 [7]. Our gradient projection operator can be straightforwardly generalized to this setting, but working out the mathematical details is 1) beyond the scope of this work and 2) would elongate and complicate an already long and dense paper.
> > > > > > > > > > > >
> > > > > > > > > > > > [5] Kroer, Christian, et al. "Faster first-order methods for extensive-form game solving." Proceedings of the Sixteenth ACM Conference on Economics and Computation. 2015.
> > > > > > > > > > > >
> > > > > > > > > > > > [6] Bai, Yu, et al. "Near-optimal learning of extensive-form games with imperfect information." International Conference on Machine Learning. PMLR, 2022.
> > > > > > > > > > > >
> > > > > > > > > > > > [7] Fiegel, Côme, et al. "Adapting to game trees in zero-sum imperfect information games." International Conference on Machine Learning. PMLR, 2023.

---

> > > > > > > > > > > > > ### Author Response · Authors · 2023-08-21
> > > > > > > > > > > > > **Immediate Impact Beyond NFG (2/2)**
> > > > > > > > > > > > >
> > > > > > > > > > > > > Setting EFGs aside, NFG research is thriving with numerous papers just presented at last year’s NeurIPS 2022 (e.g., [8, 9, 10] among many). The computational game theory community generally recognizes the value of NFG research and its importance as a keystone of algorithm development. We have included additional recent citations at the end of our response for completeness. If we don’t understand $n$-player, general-sum NFGs, we certainly do not understand the more complex $n$-player, general-sum EFGs or MARL.
> > > > > > > > > > > > >
> > > > > > > > > > > > > [8] Anagnostides, Ioannis, et al. "Optimistic Mirror Descent Either Converges to Nash or to Strong Coarse Correlated Equilibria in Bimatrix Games." Advances in Neural Information Processing Systems 35 (2022): 16439-16454.
> > > > > > > > > > > > >
> > > > > > > > > > > > > [9] Wibisono, Andre, Molei Tao, and Georgios Piliouras. "Alternating mirror descent for constrained min-max games." Advances in Neural Information Processing Systems 35 (2022): 35201-35212.
> > > > > > > > > > > > >
> > > > > > > > > > > > > [10] Marris, Luke, et al. "Turbocharging solution concepts: Solving NEs, CEs and CCEs with neural equilibrium solvers." Advances in Neural Information Processing Systems 35 (2022): 5586-5600.
> > > > > > > > > > > > >
> > > > > > > > > > > > > We understand your concern that Theorem 1’s assumptions on characterizations of equilibria appear abstruse, but providing a new view on a problem ($>2$-player NFGs) that researchers have struggled to make progress on is an important and critical contribution to the field. New headway often requires new techniques, and while these approaches are unfamiliar to the computational game theory community they are clearly familiar to the optimization community who can be recruited to help translate and collaborate on this new direction.
> > > > > > > > > > > > >
> > > > > > > > > > > > > Finally, here is a continued, partial list of very recent experimental papers on solving NFGs showing the intense interest in the area.
> > > > > > > > > > > > >
> > > > > > > > > > > > > [11] Duan, Zhijian, et al. "Is Nash Equilibrium Approximator Learnable?." Proceedings of the 2023 International Conference on Autonomous Agents and Multiagent Systems. 2023.
> > > > > > > > > > > > >
> > > > > > > > > > > > > [12] Fischer, Miriam, and Akshay Gupte. "Multilinear Formulations for Computing a Nash Equilibrium of Multi-Player Games." 21st International Symposium on Experimental Algorithms (SEA 2023). Schloss Dagstuhl-Leibniz-Zentrum für Informatik, 2023.
> > > > > > > > > > > > >
> > > > > > > > > > > > > [13] Gemp, Ian, et al. "Sample-based Approximation of Nash in Large Many-Player Games via Gradient Descent." Proceedings of the 21st International Conference on Autonomous Agents and Multiagent Systems. 2022.
> > > > > > > > > > > > >
> > > > > > > > > > > > > [14] Goktas, Denizalp, and Amy Greenwald. "Exploitability minimization in games and beyond." Advances in Neural Information Processing Systems 35 (2022): 4857-4873.
> > > > > > > > > > > > >
> > > > > > > > > > > > > [15] Marris, Luke, et al. "Multi-agent training beyond zero-sum with correlated equilibrium meta-solvers." International Conference on Machine Learning. PMLR, 2021.
> > > > > > > > > > > > >
> > > > > > > > > > > > > [16] Omidshafiei, Shayegan, et al. "α-rank: Multi-agent evaluation by evolution." Scientific reports 9.1 (2019): 9937.

---

### Official Review · Reviewer_rzbR · 2023-06-25

**Soundness:** 3 good
**Presentation:** 3 good
**Contribution:** 3 good
**Rating:** 6
**Confidence:** 2

**Summary:**

This paper presents a novel approach for determining the Nash equilibrium of normal form games, utilizing a solution to a non-convex stochastic optimization problem. It defines the Nash equilibria in normal form games as the global minima of a specifically cunstructed loss function. Moreover, a randomized algorithm is developed to resolve this newly proposed loss function. Finally, empirical results further verify the theoretical analysis.

**Strengths:**

Though the idea of loss function has been proposed before, this paper contributes to the discourse with several innovative insights that enhance the understanding and applicability of loss functions. For example, this paper restricts the parameter to the simplex, which is the key of making stochastic gradient unbiased.

Regarding the quality and clarity, this paper is sufficiently complete. It also provides clear backgrounds, which make it easy to understand how this loss function comes from. It is not completely new but it has something new.

**Weaknesses:**

The motivation of this work is not sufficiently clear. I could understand solving a Nash equilibrium may not be efficient but I don't think proposing a NE solver via unbiased stochastic optimization will make it better.

It is unclear how this method is better than some existing NE solver such as Lemke–Howson algorithm.

**Questions:**

1. I agree that there is a gap between the success of using SGD solving non-convex optimization problem and the failure of efficiently computing Nash equilibria. Why does this motivate the goal: "Can we solve for Nash equilibria via unbiased stochastic optimization"? To my understanding, solving a non-convex optimization problem is still very hard.

2.  Solving a non-convex optimization problem may lead to a stationary point instead of the global minima. Why does this proposed method is better than using an existing NE solver such as Lemke–Howson algorithm to ensure obtaining a NE?

**Limitations:**

This is a theoretical work so there is no negative impact.

---

> ### Author Rebuttal · Authors · 2023-08-09
>
> Thank you for your review and your encouraging statements. We have answered both your questions below. We hope you will consider increasing your score in light of these updates.
>
> **Why Stochastic Non-Convex Opt? Isn’t that hard?**: You are correct. Solving a stochastic non-convex optimization problem is hard. However, it has been well studied and several algorithms exist with global convergence guarantees. We employ one such algorithm, BLiN, from the X-armed bandits family. In contrast, very few stochastic algorithms exist (to our knowledge, none with guarantees) from the game theory literature for directly approximating Nash equilibria of $n$-player, general-sum normal-form games. Therefore, while stochastic non-convex optimization is hard, reformulating the problem of approximating Nash in that framework, opens up the possibility of applying a much larger class of algorithms than what is currently available.
>
> **SGD Lacks Global Guarantees**: It is correct that stochastic gradient descent may converge to a local instead of a global minimum, and that is problematic. This is why we explore using a non-gradient method like BLiN which enjoys global convergence guarantees. Regarding the classical Lemke-Howson (LH) algorithm, it is designed specifically for 2-player games. For 2+ player games, Govinda-Wilson (*gambit-gnm*) is its closest counterpart. We have run *gambit-gnm* as well as many other classical algorithms from the gambit library [1] (listed below) on the two Blotto games we examine in Figure 3. Only *gambit-enumpoly* and *gambit-enumpure* are able to return any NE within a 1 hour time limit (and only pure equilibria) and *gambit-enumpoly* times out on the larger 4-player Blotto game.  In addition, all of these algorithms require storing the entire payoff tensor in memory. For larger games, this is prohibitive whereas our stochastic sample-based algorithm can still run in these cases.
>
> *Proposed Edits*:
> - We will add text to the introduction explaining that while non-convex optimization is hard, 1) much progress has been made 2) solving games is arguably harder, and 3) stochastic techniques have yet to be thoroughly explored in the game setting.
> - We will add the above gambit results to the paper. We will also explain that we do not expect robust guarantees from running SGD, but believe it is worth investigating empirically.
>
> [1] McKelvey, Richard D., McLennan, Andrew M., and Turocy, Theodore L. (2016). Gambit: Software Tools for Game Theory, Version 16.0.1. http://www.gambit-project.org.
>
> Algorithm descriptions: https://gambitproject.readthedocs.io/en/latest/tools.html
>
> - gambit-enumpoly [73 sec 3-player, timeout 4-player]
> - gambit-enumpure [72 sec 3-player, 45 sec 4-player]
> - gambit-gnm
> - gambit-ipa
> - gambit-liap
> - gambit-logit
> - gambit-simpdiv

---

> > ### Comment · Reviewer_rzbR · 2023-08-16
> > **Continuing questions**
> >
> > Thanks for the clarification! My concerns are mostly addressed and I believe the construction of objective function that has many desired properties is an interesting contribution. I will keep my positive rating to support this work.
> >
> > However, I may have other concerns after I read others' review, and I would like to confirm it a little bit. If all NE are interior of the probability simplex, then we can directly solve it by solving the minimum point of either (3) or (6) given in this paper. If some NE are not interior, we need to add a small entropy to the original utility $u_k(x)$.
> >
> > 1. It seems that all NE of this new game will be interior NE of the new game (and QRE of the original game). Then we solve one QRE by minimizing the corresponding objective function (7) of this new game. Is my understanding correct?
> > 2. How do you connect the QRE solution and the original NE? Should there always be a non-vanishing gap?
> > 3. How large is the first term in (13), $\frac{n}{\ln(1/p)}(W(1/e)+(\bar{m}-2)/e)$? Will this term be vanishing for sufficient long training ($T\to \infty$)?

---

> > > ### Author Response · Authors · 2023-08-16
> > > **Answers to Follow-up Questions**
> > >
> > > Dear reviewer, thank you for your positive feedback and for your continued engagement. We have answered your questions below.
> > >
> > > 1. Yes, your understanding is correct. All NEs in the new game will lie in the interior (see Figure 1 for visual examples), and these will be QREs of the original game. We solve for these equilibria by minimizing (7) as you say.
> > > 2. Lemma 14 connects the QRE solution with the original NE. It shows that QREs well approximate NEs at low temperature. And yes, there is always a non-vanishing gap ($n\tau(W(1/e) + \frac{\bar{m} - 2}{e})$) that depends on the temperature $\tau$. In order to shrink this term, one must reduce the temperature $\tau$.
> > > 3. No, the first term in (13) does not vanish as $T \rightarrow \infty$, but it can be set arbitrarily small by choosing a low temperature (note $\tau = \frac{1}{\ln(1/p)}$ and the relation to Lemma 14 ). Note that decreasing the temperature increases the number of iterations required for the second term to vanish. Hence, it is left up to the user to decide how close an approximation they want.

---

> > > > ### Comment · Reviewer_rzbR · 2023-08-16
> > > >
> > > > Really appreciate the quick response!
> > > >
> > > > It seems that by following this approach, if I set a sufficiently small $\tau$ (for example, to make the first term in (13) less than $\epsilon$) then we could find an $\epsilon$-approximate Nash equilibrium of the original game in polynomial time. However, it is known that finding an $\epsilon$-approximation of Nash equilibrium of a general-sum multi-player game is PPAD-complete. Do I misunderstand something?

---

> > > > > ### Author Response · Authors · 2023-08-16
> > > > >
> > > > > Thank you for the follow up question. Indeed, we agree that this hardness result exists for the class of $n$-player, general-sum NFGs. However, our theorem statement relies on several technical conditions which further restrict our problem setting, e.g., the existence of a *well*-isolated equilibrium (defined in line 228, restated on lines 264-265) and the construction of a randomized (rather than deterministic) algorithm. Hence, we do not provide any shortcuts around PPAD-hardness results and do not see a simple modification that would enable us to make such a strong claim. Instead this theorem complements our experimental results by quantifying the performance of known global non-convex optimization techniques in regards to our system parameters. We will be happy to expand on these issues in an updated version.

---

> > > > > > ### Comment · Reviewer_rzbR · 2023-08-17
> > > > > >
> > > > > > I agree with your explanation. It seems the well-isolated equilibrium is crucial for the result. If Nash equilibria are special, obtaining the polynomial rate for general-sum multi-player games is possible. So I won't have further concerns on this point.  Still, appreciate your active responses!

---

### Official Review · Reviewer_Deih · 2023-07-03

**Soundness:** 3 good
**Presentation:** 2 fair
**Contribution:** 3 good
**Rating:** 6
**Confidence:** 3

**Summary:**

This work studies the computation of Nash equilibria (NE) of normal-form games and proposes a new loss function: the (weighted) sum of the squared norms of the projections of each player gradient onto the tangent space of the unit simplex. The authors show that this loss function is a meaningful surrogate of exploitability when the game has an interior equilibria. Then, the authors provide methods to efficiently construct unbiased estimators of the loss function via unbiased estimation of each player gradients. To extend these results to handle games with only pure equilibria, the authors propose surrogate player utility functions via entropy regularization (with coefficient $\tau$, the "temperature") and show how the modified loss function (based on the modified game with surrogate player utility functions) captures the exploitability of the original game. Next, the authors derive gradient and Hessian expressions for the modified loss function. Leveraging a recent bandit optimization method BLiN, and assuming a sufficiently large temperature (which degrades the convergence rate), the authors provide a high-probability convergence guarantee for computing NE using this approach (loss function + BLiN). Experiments on SGD and BLiN show the effectiveness of the proposed approaches.

**Strengths:**

- Novel observation of the connection between projection of player gradient to simplex tangent space and best response, which lead to the loss function proposed in this paper.
- Extensive studies of the newly proposed loss function in terms of its gradient, Hessian, and other properties.

**Weaknesses:**

Some technical details seem to require further clarification. See **Questions**.

**Questions:**

- Since BLiN is technically a zeroth order method (pulling an arm <==> sampling a function value), can you elaborate/repeat, somewhere around Theorem 1, what is the oracle passed into BLiN? I believe it should be a Monte-Carlo approximation through (6) but with the player gradients being the ones with temperatures (entropy regularization). In other words, please point out what needs to be computed in each step of BLiN.
- As stated in 229-231, if a NFG has a unique equilibrium which is also mixed, then $\mathcal{L}$ is strongly convex. Based on earlier results in this paper, are there other conditions that ensure strong or non-strong convexity of $\mathcal{L}$ (or $\mathcal{L}^\tau$)? It would be helpful to state them explicitly, as many stochastic optimization methods can exploit (strong) convexity.

**Limitations:**

This is a methodological work that does not have immediate or potential negative societal impact. The limitations are on the technical contributions and are discussed above.

---

> ### Author Rebuttal · Authors · 2023-08-09
>
> Thank you for your review and your encouraging statements. Your summary was spot on and your intuition regarding your first question is exactly correct. We hope you will consider increasing your score in light of these updates.
>
> **BLiN Steps and Oracles**: We pass an oracle that is able to produce unbiased estimates of equation (7) in exactly the same way as equation (6) (just replace all gradients with entropy-regularized ones as you said). Every subsequent step of BLiN makes a call to (6) with an increasing batch size. That batch is split in half to generate estimates of each of the gradients in the squared norm separately.
>
> **Conditions for Strong Convexity**: Excellent question. We would love to be able to say more here, but at the very least, we cannot expect strong convexity if the game has multiple disconnected equilibrium points. You can see from our Figure 1 that even small 2-player games can induce non-convex landscapes. We are able to state conditions for strong-convexity in the zero temperature setting because it avoids a complicated analysis of the third-order tensor in the second term of the Hessian. We must study the non-zero temperature setting to understand conditions for strong-convexity in the partially-mixed and pure equilibria, but so far that analysis evades us.
>
> *Proposed Edit*: As you suggested, we will add a statement similar to above that explains the BLiN procedure when applied to our setting.

---

### Official Review · Reviewer_JGDw · 2023-07-07

**Soundness:** 3 good
**Presentation:** 3 good
**Contribution:** 3 good
**Rating:** 6
**Confidence:** 2

**Summary:**

This work studies solving Nash Equilibria (NE) by stochastic unbiased optimization. The main contribution is providing a new loss function based on the gradient norm of the utility function, and finding the NE by using standard stochastic optimization methods (like Lipschitz bandit algorithms and stochastic gradient descent (SGD)). The authors also carried our experiment results on several games to show the scalability of their proposed methods.

**Strengths:**

The presentation of this work is very clear. The experiment results are comprehensive and back up the main claims of this work. The results are also significant as they point out a new way to solve the NE problem in general.

**Weaknesses:**

More remarks are supposed to be added to the main text. For example,

- What does 's' mean in the legend of Figure 3?

- In Table 1, why the obstacle of NI method is 'max of random variable'? I did not see any max operator in the definition of the loss function of NI.

- In Table 1, for the unconstrained method, can the authors show one concrete example to show why it 'lose the ability to sample from strategies when iterates are no longer proper distributions', as stated in line 113?


**Questions:**

The same to the 'Weaknesses' section.

**Limitations:**

This work aims to solve an open problem about the algorithmic game theory, thus it does not need to address the potential negative societal impacts of their work.

---

> ### Author Rebuttal · Authors · 2023-08-09
>
> Thank you for your review and your support! Indeed, we see this as a completely novel and scalable approach to solve games and we hope others can build and improve on this work. We believe we can easily answer each of your questions as follows:
> - Thank you for pointing out our omission of any description of “s”! It indicates the number of Monte Carlo samples used to estimate a gradient (i.e., the “batch size”, equiv. the number of joint actions sampled).
> - In Table 1, NI is defined as a sum over $\epsilon_k$’s. If you look further down the page, you’ll see $\epsilon_k$ is defined via a “best-response” $BR_k$ which contains an $\arg\max$. More directly, we can equivalently define $\epsilon_k = \max_z u_k(z, x_{-k}) - u_k(\boldsymbol{x})$ which makes the appearance of the max operator obvious.
> - Thank you for raising this. “No longer proper distributions” was probably poor word choice. We mean to say “no longer a vector of probabilities”. For example, it's clear how to sample 1 of 2 pure strategies from a probability vector that looks like [0.3, 0.7]. But how do you sample 1 of 2 pure strategies from a vector that looks like [-0.2, 1.6]? Do you softmax it first? Do you shift and normalize it? Any of these operations is nonlinear and would be problematic for the same reasons as the others in Table 1.
>
> We hope you will consider increasing your score in light of these updates.
>
> *Proposed Edits*:
> - We will add a description of “s” to the figure caption in addition to describing the baselines in the legend.
> - We will add a note to the table caption with the definition of $\epsilon_k$ so it is clear this term hides a max operator.
> - We will add text to the appendix elaborating on the issue of sampling strategies when they do not lie on the simplex, including a concrete example like the one above.

---

> > ### Comment · Reviewer_JGDw · 2023-08-21
> > **Reply to the authors**
> >
> > Thanks for your reply. I will keep my score as it is.
> >
> > Best,

---

### Official Review · Reviewer_ksS3 · 2023-07-08

**Soundness:** 3 good
**Presentation:** 4 excellent
**Contribution:** 3 good
**Rating:** 5
**Confidence:** 3

**Summary:**

This paper proposes a loss function (optimization problem) for normal-form games to estimate Nash equilibria which can be solved via unbiased stochastic optimization. They do this by relating their proposed loss function with exploitability. They also provide theoretical guarantees (under some technical conditions) of using bandit stochastic gradient algorithms to solve their proposed problem. They show the applicability of their method by conducting some numerical experiments.

**Strengths:**

The paper tackles an important problem in the game theory of estimating Nash equilibria using optimization. It proposes a potentially scalable solution and provides theoretical guarantees for the same. The paper is well-written (albeit a bit notation-heavy) and the content is easy to follow.

**Weaknesses:**

The authors propose an optimization problem with possible unbiased estimators. However, the proposed problem is still non-convex and it is not clear to me whether it can be solved efficiently with SGD with a potentially large number of saddle points. I understand the analogy to deep learning problems but recent work ([1] and related papers) have shown that those problems carry some interesting structure. Similar properties are unknown (and are perhaps more difficult to establish) for the proposed function.

Du, S., Lee, J., Li, H., Wang, L., & Zhai, X. (2019, May). Gradient descent finds global minima of deep neural networks. In International conference on machine learning (pp. 1675-1685).

**Questions:**

Could authors comment on the applicability of their methods on real problems in the context of my comments in the "Weaknesses" section?

**Limitations:**

The limitations are addressed adequately by the authors.

---

> ### Author Rebuttal · Authors · 2023-08-09
>
> Thank you for your review and your intriguing question! We have answered below. We hope you will consider increasing your score in light of these updates.
>
> **SGD Lacks Global Guarantees**: We agree that it remains unknown whether SGD and/or other gradient-based methods can cope with the potentially many saddle points and local minima of our proposed loss function. This is what motivated us to reproduce the experiment of [12] in Figure 2, which reveals an interesting story on “real” normal-form games. For some games (e.g., 2-player Sheriff), the loss has many local minima (every circle plotted @ $\alpha=0$ indicates a local minimum) which could be problematic for a gradient-based solver. However, for other games (e.g., 3-player Leduc poker), the loss only has a few suboptimal local minima, but has many saddle points. In that case, there exist gradient-based solvers that are specifically designed to circumvent saddle points [12]. Moreover, we analyze this property of our loss in Figure 3 in the Blotto game. In both Blotto games, SGD asymptotes to a positive (suboptimal) level of epsilon. We are able to analyze the Hessian at the end of 10k iterations and measure its spectrum to determine that SGD is not converging to a local minimum, but rather being temporarily slowed by a saddle point! We only pointed this out in the last sentence of Figure 3’s caption due to space constraints, but we should emphasize this.
>
> Hence, to answer your question, we believe our preliminary analysis shows that while the loss is non-convex in general, it looks like SGD can in fact successfully minimize this loss in some cases. And in other cases, where saddle points are a problem, we should be able to employ more sophisticated gradient-based methods as in [1] capable of circumventing these saddle points.
>
> Furthermore, note that the analysis of loss landscapes for deep networks requires considering different choices of architectures, nonlinearities, drop-out, and other network design options. In contrast, our loss is polynomial for any normal-form game. In that sense, we already have a much better understanding of the landscapes induced by our loss than researchers initially had of neural networks. We hope we can similarly make quick progress in the game setting.
>
> *Proposed Edit*: We will try to incorporate parts of this discussion into the main body and a longer version into the appendix.

---

### Author Rebuttal · Authors · 2023-08-09

Thank you for your constructive feedback! We really appreciate the interesting and constructive questions that were raised, and believe they will help us improve the exposition of the paper. In each of our responses, we answer your questions and propose edits to address them in the paper. Please let us know if the proposed edits satisfy your concerns and/or whether you have further suggestions.
Overall, it seems the reviewers found the proposed approach to be innovative and the presentation to be very clear with comprehensive experiments and analysis. We have pulled and grouped a few of the reviewers' own quotes below for convenience.

**Innovative**:
- **ksS3**: N/A
- **JGDw**: “The results are also significant as they point out a new way to solve the NE problem in general”
- **Deih**: “Novel observation of the connection between projection of player gradient to simplex tangent space and best response”
- **rzbR**: N/A
- **4TEh**: “promising approach, and has the potential to bring many new insights to equilibrium computation”

**Clear**:
- **ksS3**: presentation rating: excellent, “The paper is well-written”
- **JGDw**: “The presentation of this work is very clear”
- **Deih**: N/A
- **rzbR**: “Regarding the quality and clarity, this paper is sufficiently complete. It also provides clear backgrounds”
- **4TEh**: “Furthermore, the presentation and the writing are overall clear, and the authors accurately place their results into the existing literature”

**Comprehensive Experiments & Analysis**:
- **ksS3**: N/A
- **JGDw**: “The experiment results are comprehensive and back up the main claims of this work.”
- **Deih**: “Extensive studies of the newly proposed loss function in terms of its gradient, Hessian, and other properties.”
- **rzbR**: “enhance the understanding and applicability of loss functions…Finally, empirical results further verify the theoretical analysis.”
- **4TEh**: N/A

---

### Decision · Program_Chairs · 2023-09-21

**Decision:**

Reject

**Comment:**

The paper has received mildly positive reviews. While your reviews appreciate the innovative insights, clear presentation, and potential scalability of the method, concerns are raised about the strong assumption of an interior Nash equilibrium, clarity on the paper's motivation, comparison with existing algorithms like Lemke–Howson, and the need for clearer experimental benchmarks and real-world applicability.

Based on my reading of your reviews, here is the summary of positive and negative aspects of the paper:

Positive Aspects:

- The paper tackles an important problem in game theory regarding estimating Nash equilibria using optimization.
- The paper proposes a potentially scalable solution for which the authors provide theoretical guarantees. The approach appears to be novel.

Negative Aspects:

- There are questions about the practical applicability of the proposed method and how it compares with existing methods.
- There is a concern regarding the underlying assumptions or conditions of the paper's methods, particularly regarding the existence of an interior Nash equilibrium.
- The proposed optimization problem is non-convex, and as a result it is not clear if it can be solved efficiently with SGD. The proposed method has no theoretical finite-time guarantees of reaching a Nash equilibrium.
- There are some questions about the motivation for the work.
- The assumption of an interior Nash equilibrium is very strong and can be computed in polynomial time via linear programming, weakening the motivation.
- There are issues about the experiment.

After the discussion a number of issues remain. The first main issue is the theoretical soundness of the approach --- there are no finite-time guarantees of convergence to Nash equilibria, and there are considerable gaps between the theory provided and the experimental evaluation. The theoretical results are not convincing. Main results, such as Theorem 1, crucially depend on notions of dimension not justified at all from a game-theoretic standpoint. The reviewers has issues with the notion of "well-isolated equilibria," which is overly restrictive and artificial. The empirical success of the proposed method also relies on whether algorithms such as SGD find "good" local optima; the fact that this is the case is many single-agent deep learning applications does not mean that the same should hold for the problem of computing Nash equilibria. It is not clear that the paper currently provides enough evidence to support this claim. Finally, it is also unclear if the paper will have practical impact because it applies solely on normal-form games, which is unlike most literature on computational game solving that focuses on extensive-form games.